# Single-cell analysis of lizard blastema fibroblasts reveals phagocyte-dependent activation of Hedgehog-responsive chondrogenesis

Ariel C. Vonk [1,2], Xiaofan Zhao[3], Zheyu Pan [1,2], Megan L. Hudnall[2], Conrad G. Oakes [2], Gabriela A. Lopez[1], Sarah C. Hasel-Kolossa [1], Alexander W. C. Kuncz [2], Sasha B. Sengelmann[2], Darian J. Gamble[1,2] & Thomas P. Lozito [1,2] ✉

Lizards cannot naturally regenerate limbs but are the closest known relatives of mammals capable of epimorphic tail regrowth. However, the mechanisms regulating lizard blastema formation and chondrogenesis remain unclear. Here, single-cell RNA sequencing analysis of regenerating lizard tails identifies fibroblast and phagocyte populations linked to cartilage formation. Pseudo-time trajectory analyses suggest *spp1*[+]-activated fibroblasts as blastema cell sources, with subsets exhibiting *sulf1* expression and chondrogenic potential. Tail blastema, but not limb, fibroblasts express *sulf1* and form cartilage under Hedgehog signaling regulation. Depletion of phagocytes inhibits blastema formation, but treatment with pericytic phagocyte-conditioned media rescues blastema chondrogenesis and cartilage formation in amputated limbs. The results indicate a hierarchy of phagocyte-induced fibroblast gene activations during lizard blastema formation, culminating in *sulf1*[+] pro-chondrogenic populations singularly responsive to Hedgehog signaling. These properties distinguish lizard blastema cells from homeostatic and injury-stimulated fibroblasts and indicate potential actionable targets for inducing regeneration in other species, including humans.

Appendage regeneration remains a lofty goal in mammalian stem cell research, with few examples of native large-scale tissue repair/replacement in humans and traditional mammalian model organisms. With complex immune systems leading to extensive inflammatory responses to tissue injury, mammal appendage loss, with few exceptions, tends to result in fibrosis and scar formation rather than a regenerative response[1, 2]. Research in non-mammalian vertebrates may provide clues as to the molecular mechanisms and pathways responsible for a regenerative outcome following limb and tail loss.

Regeneration research often highlights the axolotl salamander (*Ambystoma mexicanum*) due to its remarkable ability to regenerate perfectly patterned copies of lost limbs or tails. However, these amphibian organisms are distantly related to humans evolutionarily, do not undergo the same life cycle stages as mammals, and exhibit

[1]Department of Stem Cell Biology and Regenerative Medicine, Keck School of Medicine, University of Southern California, 1425 San Pablo St, Los Angeles, CA 90033, USA. [2]Department of Orthopaedic Surgery, Keck School of Medicine, University of Southern California, 1540 Alcazar St, Los Angeles, CA 90033, USA. [3]Molecular Genomics Core, USC Norris Comprehensive Cancer Center, Keck School of Medicine, University of Southern California, 1441 Eastlake Ave, Los Angeles, CA 90033, USA. ✉e-mail: lozito@usc.edu

neoteny, meaning they maintain juvenile features through adulthood[3]. Evolutionarily closer related mouse digit tip studies involve a model system more akin to humans and other mammals, but can only achieve limited regrowth, requiring nail bed presence and restricting amputation to the most distal phalangies[4]. Further, mouse digit tip healing is lineage-restricted, and blastema cells in adult mice are unable to differentiate into cartilage[5,6]. Deer regenerate antlers, another example of mammalian regeneration, but do not develop blastemas during the process of regeneration[7], highlighting a significant divergence between their regrowth process and other blastema-based appendage regeneration models.

Lizards, amniotes that are more closely related evolutionarily to humans than amphibian models, provide an intermediary for appendage regeneration research due to their natural epimorphic tail regeneration capacity throughout their lifespan[8]. In addition, lizards have more complex and adaptive immune systems[9], more like that of mammals, contrasting more rudimentary immune systems exhibited in amphibians, lacking sophisticated adaptive immunity[10,11]. Thus, lizards emerge as an excellent model to investigate the role of the inflammatory response in large-scale appendage replacement. Interestingly, lizards regenerate distinctly different copies of their tails following loss, rather than recapitulating the original tail tissues and patterning. Regenerated tails are fully functional and contain similar muscle, epithelial, endothelial, adipose, and nervous tissues[12]. However, original tails are structured by an ossified vertebrate skeleton, while the regenerated lizard tail skeleton consists of a single cartilage tube surrounding the regenerated spinal cord. In addition, regenerated tails appear to be innervated only by the peripheral nervous system and lack true tendons[9,12-18]. Despite these differences, lizards provide a promising model for appendage regeneration in amniotes, as well as a model for large-scale chondrogenesis.

The immune response to appendage loss plays a massive role in the outcome of repair and regeneration. Macrophages act in the early innate immune response to injury, traditionally referred to as M1 macrophages, removing debris and necrotic tissue via phagocytosis, and secreting chemokines, cytokines, and matrix-degrading enzymes, aiding in the coordination and recruitment of other inflammatory response cells. Later in the immune response, M2 macrophages aid in tissue growth and repair, secreting pro-angiogenic and proliferative factors to the regenerating tissue environment[19-22].

The recruitment of macrophages and phagocytic cells to injury sites has been shown to be crucial for appendage regeneration in several model organisms. For example, upon macrophage depletion and amputation, mouse digit tip cells fail to accumulate at the wound site, the wound does not re-epithelialize, and the digit does not regenerate[23]. Similar depletion treatments in axolotls following limb amputation result in successful wound closure but lack subsequent limb regeneration[24]. Ablation of macrophages during adult zebrafish caudal fin amputation resulted in lack of blastema formation and fin regeneration[25], and macrophage loss prevents epimorphic ear pinna regeneration in African spiny mice[26], underscoring the importance of macrophage interaction during regrowth. Phagocytic osteoclasts have also been shown to play a critical role in bone resorption and patterning in axolotl skeleton regrowth during limb regeneration[27-29]. Macrophages have been identified in amputated lizard tail and limbs injuries, including within regenerating tail blastema[30-32], while systemic depletion of phagocytic cells in lizards has been shown to prevent tail stump tissue ablation and subsequent tail regeneration[9].

Among the cells recruited by macrophages during the immune response, fibroblasts have been shown to activate and migrate to areas of tissue loss in a coordinated manner[33]. Fibroblasts localize to wounded tissue and respond to signals from the injured tissue environment in a similar manner to innate immune cells, reacting to chemokines from damaged platelets and damage-associated molecular patterns released by apoptotic cells[20]. In later stages of immune responses, when inflammation is resolving, fibroblasts proliferate and deposit extracellular matrices and collagens, which are critical for successful tissue remodeling and regeneration[22,34]. Reciprocally, fibroblasts have also been shown to recruit macrophages in some instances of tissue repair through chemokine secretion[35]. Dysregulation of macrophage-fibroblast crosstalk in pro-inflammatory stages can lead to fibrosis, scarring, and aberrant or incomplete tissue repair[2].

Fibroblastic connective tissue cells (FCTCs) have been previously identified as the most abundant cell type found in blastemas of regenerating axolotl limbs[28,36-39] Lineage tracing of axolotl blastema FCTCs reveal derivation from connective tissues of remaining limb stump skeleton, cartilage, tendons, dermis, pericytes, and interstitial fibroblasts, following limb loss. These blastema FCTCs mimic embryonic limb bud signatures before redifferentiating into the patterned limb skeleton and connective tissues[36,40]. The cell types and signatures that make up lizard blastemas and lead to chondrogenesis are largely unknown.

Here, we present single-cell RNA sequencing performed in the green anole lizard, *Anolis carolinensis*, characterizing regeneration of the tail and an investigation of the role of FCTCs and phagocytes in regeneration and chondrogenesis.

## Results
### Heterogeneous fibroblasts contribute to lizard tail regrowth
To investigate the complexity and heterogeneity of each regeneration state during tail regrowth, we performed single-cell RNA sequencing (scRNAseq) on staged tail samples of the green anole lizard, *Anolis carolinensis*. Utilizing the 10x Genomics scRNAseq platform, tail samples were divided into one of the four following sample groups: original tail (0 day post-amputation, DPA), inflammatory stage (1, 3, and 7 DPA), blastema stage (14 and 21 DPA), or regenerated homeostasis (28 DPA). Inflammatory and blastema stage samples included multiple sample time points to ensure consistency among sample groups considering inherent variability in tail regeneration between individuals[41]. In addition to tracking sample DPA, samples were assessed for morphology and characteristic regeneration stage phenotypes[42], resulting in multiple time points utilized in inflammatory and blastema stages (detailed phenotypes/morphologies for each tail stage are described in Methods). Sequencing results were analyzed using the 10x Genomics Cell Ranger[43] pipeline and R packages Seurat[44] and Harmony[45].

Unsupervised UMAP clustering of the regeneration time course revealed 14 distinct cell clusters (Fig. 1a). Clusters were analyzed for top differentially expressed genes, and key cell types were validated via in situ hybridization (ISH) and histology for corresponding tissue gene expression in regenerating tail samples (Supplementary Fig. 1). Several clusters of immune and blood cells were identified including cathepsin B (*ctsb*+) macrophages[9], clusters delineated by high levels of CD8 subunit A (*cd8a*+) and CD8 subunit B (*cd8b*+) expression and nucleated red blood cells. Keratin type II cytoskeletal 5 (*krt5*+) epithelial cells, von Willebrand factor (*vwf*+) endothelial cells, fatty acid binding protein 7 (*fabp7*+) ependymal cells[46], cycling cells characterized by high marker of proliferation Ki-67 (*mki67*) expression, creatine kinase M-type (*ckm*+) muscle-related cells[15], premelanosome protein (*pmel*+) melanocytes, SRY-box transcription factor 9 (*sox9*+) chondrocytes[47], and collagen type I alpha 1 chain (*col1a1*+) fibroblastic connective tissue cell (FCTC) clusters were also identified through differential gene expression analysis and mammalian ortholog identification.

When analyzed by regeneration stage, proportional contributions of each cell type compared to the total cells in the sample (Fig. 1b) revealed a large expansion in the FCTCs population at the blastema stage, similar to the previously mentioned axolotl scRNAseq analysis[36]. This FCTC cluster was closely associated with the chondrocyte cell cluster, the skeletal precursors for regenerating lizard tail. Proportional cell type contributions were validated via fluorescent ISH (FISH)

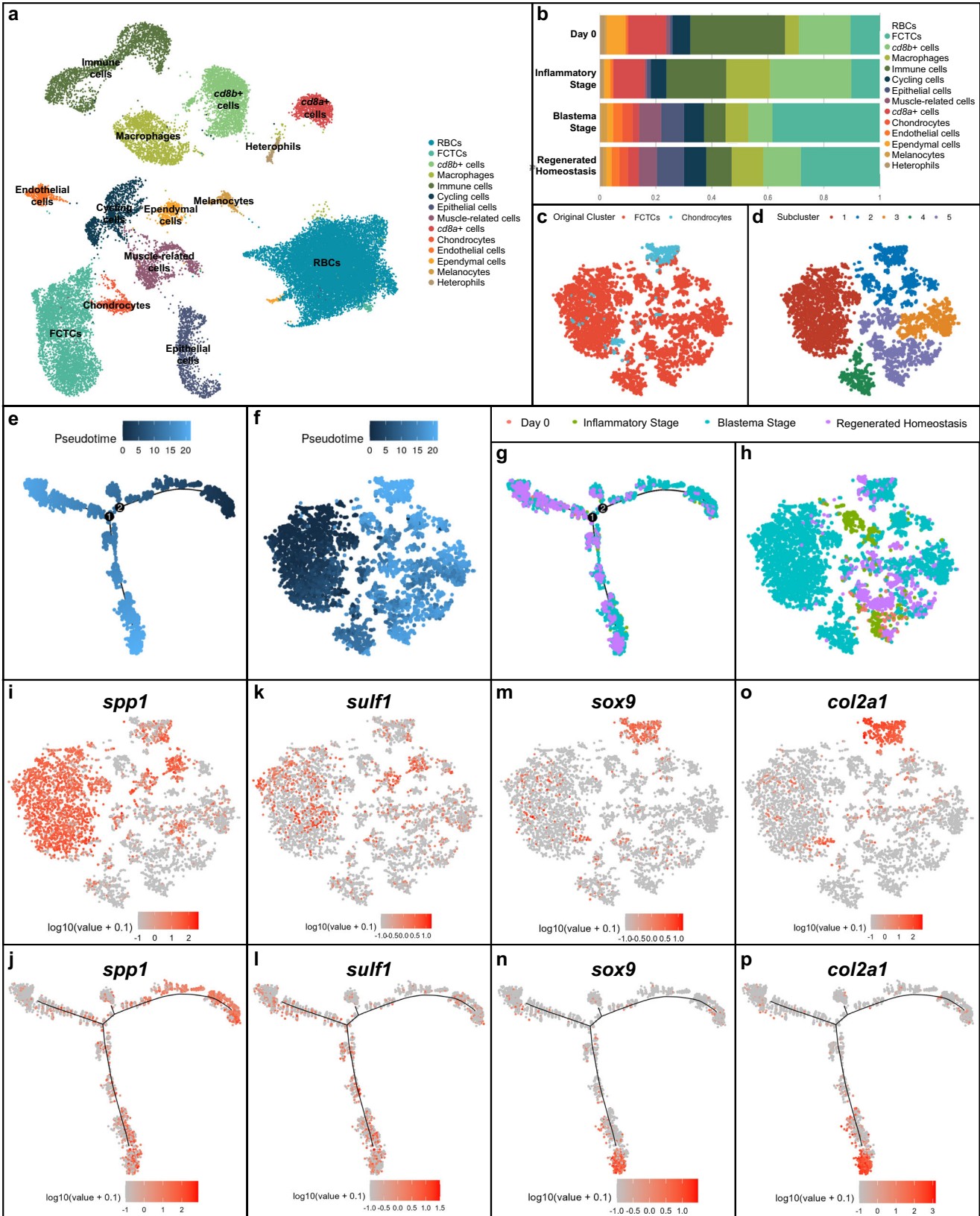

using cluster-defining marker genes (Supplementary Fig. 2), depicting a significant increase in FCTCs 14 DPA compared to 0 and 7 DPA. Given the expansion of FCTCs in the blastema stage and their close clustering with chondrocytes, the FCTC cluster and chondrocyte cluster were isolated, computationally integrated, and subclustered (Fig. 1c, d) to determine if distinct fibroblast populations existed within the FCTC

cluster and to reveal any possible gene signatures leading to FCTC chondrogenic potential.

Pseudotime trajectory analysis, a computational method used to infer biological transitions or cell lineage relationships based on scRNAseq gene expression profiles[48], was performed on the FCTC/chondrocyte subcluster (Fig. 1e, f) using R package Monocle2[49–51] and

**Fig. 1 | Single-cell RNA sequencing of regenerating lizard tails with pseudotime trajectory analysis of fibroblastic connective tissue cells and chondrocytes throughout tail regeneration. a** UMAP of unsupervised clustering of single-cell RNA-sequencing (scRNAseq) results for lizard (*Anolis carolinensis*) tail regeneration time course comprised of uninjured day 0 lizard tail (0 days post-amputation or DPA), inflammatory stage (1, 3, and 7 DPA), blastema stage (14 and 21 DPA) and regenerated homeostasis (28 DPA) samples. **b** Quantification of relative cell type composition proportions per tail sample stage, excluding red blood cells. **c** TSNE of fibroblastic connective tissue cells (FCTCs) and chondrocytes isolated, integrated, and subclustered from scRNAseq dataset in (**a**). **d** TSNE of unsupervised clustering of FCTCs and chondrocyte cell subset. **e** Monocle2 pseudotime trajectory analysis for FCTCs and chondrocyte scRNAseq cell subset from (**c**). **f** Pseudotime overlayed onto TSNE FCTC/chondrocyte cell subcluster. **g** Regeneration time point overlayed on pseudotime trajectory for FCTC/chondrocyte subset. **h** FCTC/chondrocyte TSNE analyzed by regeneration time point. **i, j** *Spp1*, **k, l** *sulf1*, **m, n** *sox9*, and **o, p** *col2a1* gene expression in FCTC/chondrocyte TSNE subcluster and FCTC/chondrocyte subcluster pseudotime trajectory, respectively.

revealed one minor and two major branch points of potential cell fate trajectory change over the course of regeneration. When analyzing by regeneration time point (Fig. 1g, h and Supplementary Fig. 3), earlier pseudotime cell populations corresponded mostly with blastema stage cell populations, while later pseudotime branch points were dominated by regenerated homeostasis and original tail cells, suggesting the blastema stage and early pseudotime represent a less terminally differentiated, more plastic cell type, while regenerated homeostatic cells in late pseudotime may have a more restricted/defined cell fate.

Further analysis of differentially expressed genes (DEGs) in the FCTC/chondrocyte subcluster revealed high levels of osteopontin, also referred to as secreted phosphoprotein 1 (*spp1*), expression (Fig. 1i, j), particularly in subcluster 1 of the FCTC/chondrocyte dataset, within the blastema stage sample and early pseudotime. *Spp1* was first described for its role in bone mineralization and extracellular matrix deposition[52], but more recent work suggests *spp1* can act as a cytokine and play a role in injury response[53-56]. For example, *spp1* has been implicated in Wnt and Hedgehog (Hh) pathway signaling modulation[57,58] and may help critical cell populations survive wound environments due to anti-apoptotic activities[59].

Sulfatase 1 (*sulf1*) expression (Fig. 1k, l) spanned blastema and regenerated time points, as well as most of pseudotime, with expression ranging early, middle, and late pseudotime. As a heparin sulfate 6-O-endosulfatase that selectively removes 6-O-sulfate groups from heparin sulfate proteoglycans (HSPGs), *sulf1* enzymatic activity modulates the binding and downstream signaling of many HSPG receptors for heparin-binding growth factors and cytokines[60,61]. Endosulfatases have been shown to be necessary for activating Wnt and BMP signaling during mammalian and avian skeletalgenesis and to dampen FGF signaling[61-63], while *sulf1* specifically has also been shown to modulate Hh signaling by enhancing local sonic hedgehog (*shh*) concentrations and availability[64].

Chondrocyte markers *sox9* (Fig. 1m, n) and collagen type II alpha 1 chain (*col2a1*)[15] (Fig. 1o, p) were heavily focused in subcluster 2, in regenerated homeostatic cells, and were specifically expressed in the bottom branch of late pseudotime. Thus, the right branch of the early pseudotime trajectory was dominated by *spp1*+ blastema stage fibroblasts and the bottom branch of late pseudotime represented more differentiated *sox9*+ regenerating chondrocytes and terminally differentiated *col2a1*+ cartilage in regenerated homeostasis. DEGs phospholipid transfer protein (*pltp*) and spalt like transcription factor 1 (*sall1*) were also analyzed via pseudotime trajectory analysis (Supplementary Fig. 4) but did not reveal distinct expression patterns correlating pseudotime with specific regeneration stages.

Taken together, pseudotime trajectory analysis suggested fibroblasts gained FCTC marker gene expression over the course of regeneration, eventually leading to potential chondrogenic capacity and chondrocyte cell fate. Many blastema FCTCs express *spp1*. Some, but not all, of those fibroblasts may go on to express *sulf1* as they continue along the tail regeneration process, and later, can become *sox9*+ chondrocytes and form *col2a1*+ cartilage. This scRNAseq study represents a critical time course analysis of lizard tail regeneration, and through pseudotime trajectory, proposes a potential relationship between FCTC marker gene expression and blastema FCTC chondrogenesis, investigated further below.

## Lizard tail blastemas consist of FCTCs

Using single-cell data analysis, several differentially upregulated fibroblast genes were identified within the FCTCs cluster, many with implications in other organisms for condensing mesenchyme, cartilage formation and bone remodeling and deposition functions[52,65,66]. These FCTC marker genes were stained for expression throughout tail regeneration, revealing changes in fibroblast gene expression patterns in a spatiotemporal manner (Fig. 2 and Supplementary Fig. 5). In uninjured original tails, collagen type III alpha 1 chain (*col3a1*+) homeostatic FCTCs lined periosteum, perichondrium, epidermis, and other connective tissues (Fig. 2a–f), with many FCTCs also expressing low levels of cadherin 11 (*cdh11*), (Supplementary Fig. 5a–e), as previously reported[67]. By 7 DPA, several additional fibroblast genes activated compared to 0 DPA and relocalized from terminal vertebrae to distal wound sites. These injury-state FCTCs exhibited elevated *spp1* (Fig. 2k), as well as *col3a1* (Fig. 2g), *cdh11*, collagen type XII alpha 1 chain (*col12a1*), midkine (*mdk*), secreted protein acidic and cysteine rich (*sparc*), and tenascin-like (*tnl*) expression (Supplementary Fig. 5f–j). *Sulf1* began to localize to regions adjacent to amputated spinal cords (Fig. 2l), while phospholipid transfer protein (*pltp*), spalt like transcription factor 1 (*sall1*), and *sox9* (Fig. 2h–j) exhibited little-to-no expression 7 DPA.

During blastema formation 14 DPA, FCTCs aggregated at distal tail tips and increased expression of several marker genes. Extensive *sparc* expression was observed throughout blastemas, including regenerating muscle bundles (Supplementary Fig. 5n), while *col3a1, spp1, col12a1, mdk*, and *tnl* labeled all newly formed blastemal tissue, but with markedly lower expression in regenerating muscle bundles (Fig. 2m, q and Supplementary Fig. 5l, m, o). Several FCTC markers exhibited more localized expression patterns within tail blastemas. *Sox9* expression, the conserved marker gene of chondrogenic potential and cartilage regeneration in lizards[47], labeled pro-chondrogenic mesenchyme condensing around central regenerated spinal cords and exhibited high medial expression to low lateral expression and proximal (high) to distal (low) organizations (Fig. 2p). Condensing chondrogenic mesenchyme was also labeled by *pltp, sall1, sulf1*, and *cdh11* expression (Fig. 2n, o, r and Supplementary Fig. 5k). *Sulf1* also labeled FCTC populations at distal blastema tips, exhibiting medial to lateral and proximodistal organizations, inverse to those of *sox9* (Fig. 2r).

At 28 DPA, regenerated connective tissue maintained high expression of *spp1, cdh11, mdk*, and *tnl*, but these markers were largely excluded from differentiated muscle and cartilage elements (Fig. 2w and Supplementary Fig. 5p, r, t). *Col12a1* and *sparc* were highly expressed in connective tissues and regenerated cartilage, but specifically excluded from regenerated muscle bundles (Supplementary Fig. 5q, s). Only *col3a1* maintained high expression in FCTC populations in every stage of regeneration from 0 to 28 DPA in connective tissues within the epidermis, muscle bundles, cartilage elements and interstitium (Fig. 2s). *Sox9* expression specifically labeled cartilage tubes and decreased proximodistally (Fig. 2v). *Pltp* and *sall1* were most highly expressed medially, surrounding cartilage tubes, and distally at tail tips, but were largely lost in other connective tissues (Fig. 2t, u). *Sulf1* expression was notably absent 28 DPA, with only minimal expression remaining in cartilage tubes (Fig. 2x). Overall, regenerated tails 14 and 28 DPA were dominated by FCTC populations despite their

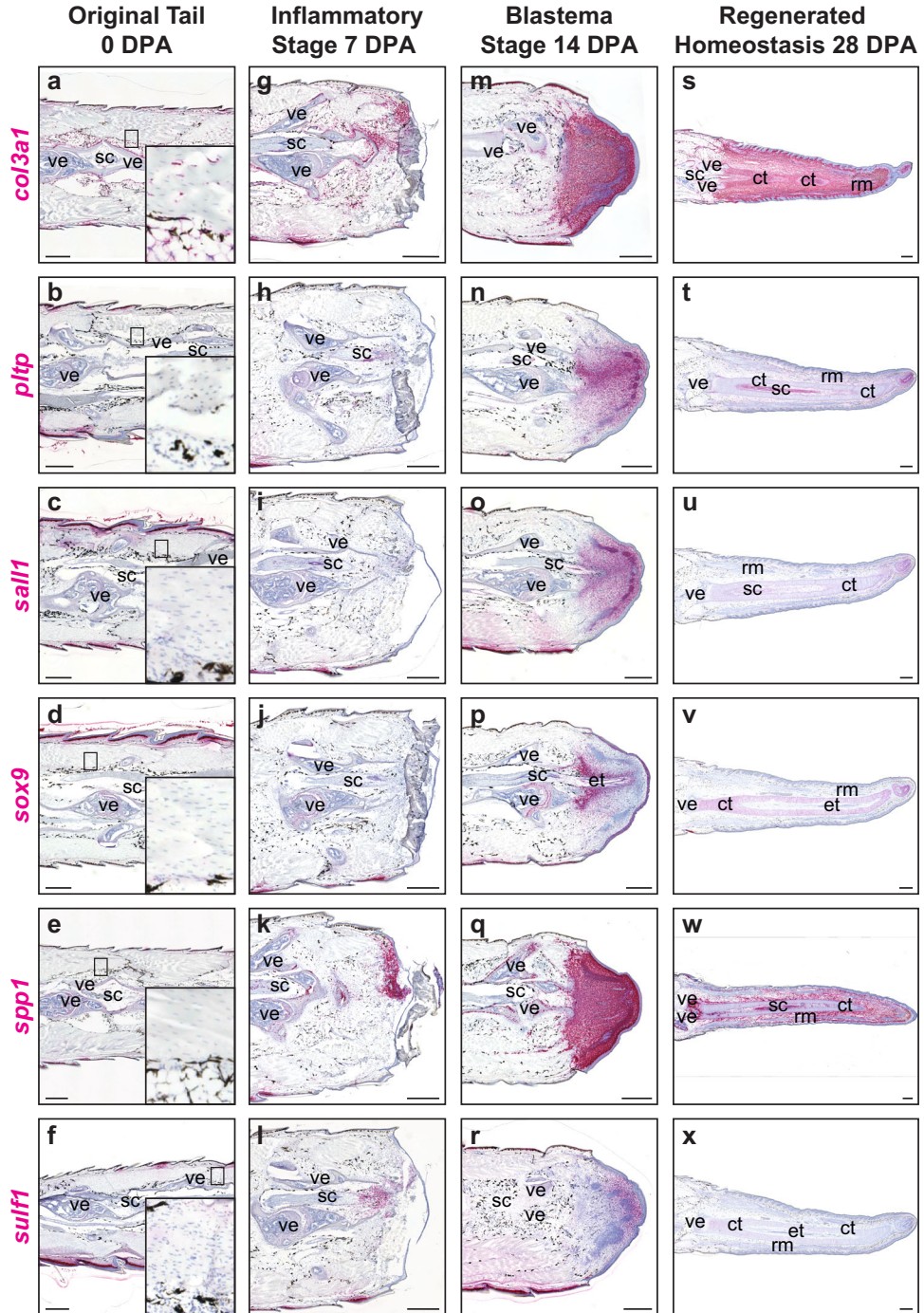

**Fig. 2 | Lizard tail fibroblasts exhibit spatiotemporal marker gene activation during regeneration.** Representative sagittal sections of original and regenerating lizard (*A. carolinensis*) tail samples analyzed via RNAscope™ in situ hybridization. **a**–**f** Original tails (0 DPA) with magnified inset of homeostatic fibroblasts, **g**–**l** inflammatory stage tails 7 DPA, **m**–**r** blastema stage tails 14 DPA, and **s**–**x** regenerated homeostasis stage tails 28 DPA analyzed for *col3a1, pltp, sall1, sox9, spp1*, and *sulf1*. *n* = 5 animals/samples per time point. ct cartilage tube, et ependymal tube, rm regenerated muscle, sc spinal cord, ve vertebra. Bar = 500 µm.

relatively small and restricted niches within original tails (0 DPA), while several FCTC genes were turned on during the injury-state of tail regeneration 7 DPA.

### *Sulf1*+ blastema FCTCs stimulated by Hh form cartilage

ScRNAseq results described above indicated distinct cartilage and blastema FCTC clusters, and previous studies from our lab have identified ependyma-contributed Hh signaling as the critical signal for inducing blastema cartilage formation[14]. Pharmacological agents were used to test the effects of Hh inhibition and activation

on blastema cell chondrogenesis (Supplementary Fig. 6). Lizards were treated with the Hh inhibitor cyclopamine, Hh smoothened agonist (SAG), or vehicle control for 28 days. Tails were then collected and analyzed by gross morphology (Supplementary Fig. 6a–c) and histology/FISH for expression of *col2a1*, a marker of mature cartilage differentiation, *shh*, the predominant Hh signal within regenerated tails, and *fabp7*, an ependymal cell marker (Supplementary Fig. 6d–l).

Lizards treated with vehicle control developed typical regenerated tails with cylindrical *col2a1*+ cartilage tubes

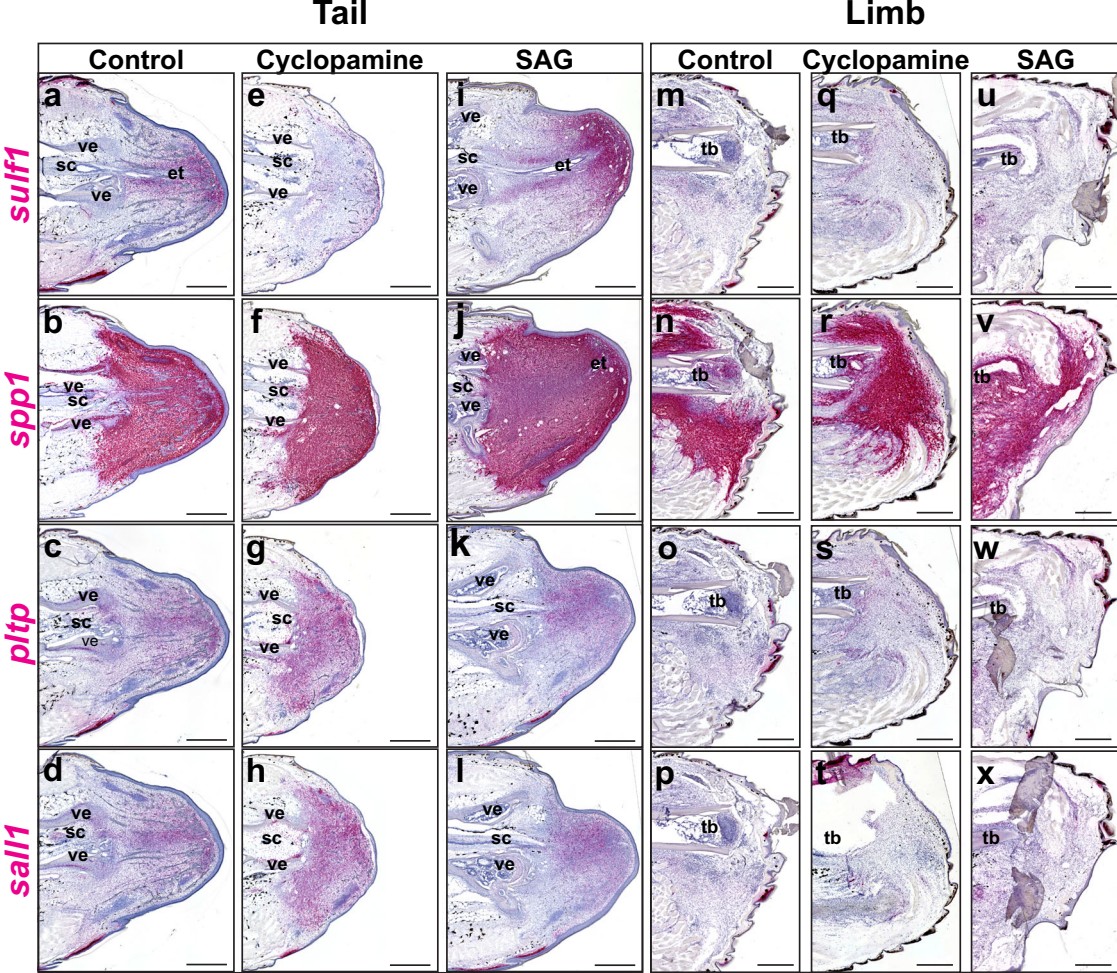

**Fig. 3 | *Sulf1* expression in lizard tail blastema, but not limb, is regulated by Hedgehog stimulation.** Representative sagittal sections of **a–l** tail and **m–x** limb samples collected from lizards (*A. carolinensis*) 14 DPA treated with **a–d, m–p** vehicle control, **e–h, q–t** Hedgehog inhibitor cyclopamine and **i–l, u–x** Hedgehog smoothened agonist, SAG, and analyzed by histology/ISH for *sulf1, spp1, pltp,* and *sall1*. *n* = 5 lizards per treatment condition. et ependymal tube, sc spinal cord, tb tibia, ve vertebra. Bar = 500 μm.

surrounding *shh⁺ fabp7⁺* ependymal tubes (Supplementary Fig. 6a, d, g, h). Lizards treated with cyclopamine regrew tails of normal length, but completely lacked cartilage despite maintenance of *shh* expression by ependymal tubes (Supplementary Fig. 6b, e, i, j). Conversely, treatment with SAG resulted in stunted, bulbous tails filled with abundant *col2a1⁺* ectopic cartilage regions in addition to endogenous cartilage tubes (Supplementary Fig. 6c, f, k, l). Neither Hh inhibition nor activation effected *shh* expression by *fabp7⁺* ependymal cells (Supplementary Fig. 6h, j, l), indicating that changes in blastema FCTC chondrogenesis resulted directly from drug treatments. SAG-induced cartilage was not observed anywhere else in the lizard and was specific to blastema-derived tail regions. These results suggested that Hh signaling is necessary and sufficient for inducing chondrogenesis in blastema FCTCs and that a large portion of blastema cells are capable of cartilage differentiation. Exogenous Hh signals from SAG treatment extend pro-chondrogenic areas beyond regions that typically form cartilage in response to endogenous signaling.

Next, the effects of Hh activators/inhibitors on lizard limb and tail FCTCs gene expression and chondrogenic potential were compared. Lizard tails and limbs were collected 28 DPA from animals treated with cyclopamine, SAG, or vehicle control and analyzed by histology/ISH for *col2a1, spp1,* and GLI family zinc finger 1 (*gli1*) expression (Supplementary Fig. 7). As described above, control tails developed *col2a1⁺* cartilage tubes. Cartilage tube development was inhibited by

cyclopamine treatment and expanded by SAG treatment, resulting in extensive ectopic cartilage formation (Supplementary Fig. 7a, g, m). Tail FCTCs maintained *spp1* expression 28 DPA, and expression was unaffected by cyclopamine or SAG treatment (Supplementary Fig. 7b, h, n).

In direct contrast to tail blastema cells, limb fibroblasts did not express *col2a1* or undergo chondrogenesis in any of the conditions tested (Supplementary Fig. 7d–f, j–l, p–r). Specifically, ectopic cartilage did not form in SAG-treated groups (Supplementary Fig. 7p). Furthermore, limb FCTC *spp1* expression was not maintained 28 DPA and was not sensitive to Hh signaling (Supplementary Fig. 7e, k, q). *Gli1*, a downstream reporter of the Hh signaling pathway and an established readout of Hh pathway activation[68], was expressed natively in control tails (Supplementary Fig. 7c), while expression was absent in control limbs (Supplementary Fig. 7f) and in both limb and tail Hh inhibitor cyclopamine treatment samples (Supplementary Fig. 7i, l). SAG-treated tail and limb both exhibited high levels of *gli1* activation (Supplementary Fig. 7o, r), validating SAG treatment as a Hh pathway activator in both tail and limb. These results indicated that, unlike tail blastemal FCTCs, amputated limb fibroblasts lacked Hh-responsive chondrogenic potential, despite evidence of sufficient Hh pathway activation with SAG treatment.

Single-cell sequencing results described above identified *sulf1*, *pltp*, *sall1*, and *spp1* as lizard tail FCTC blastema markers. *Sulf1* is

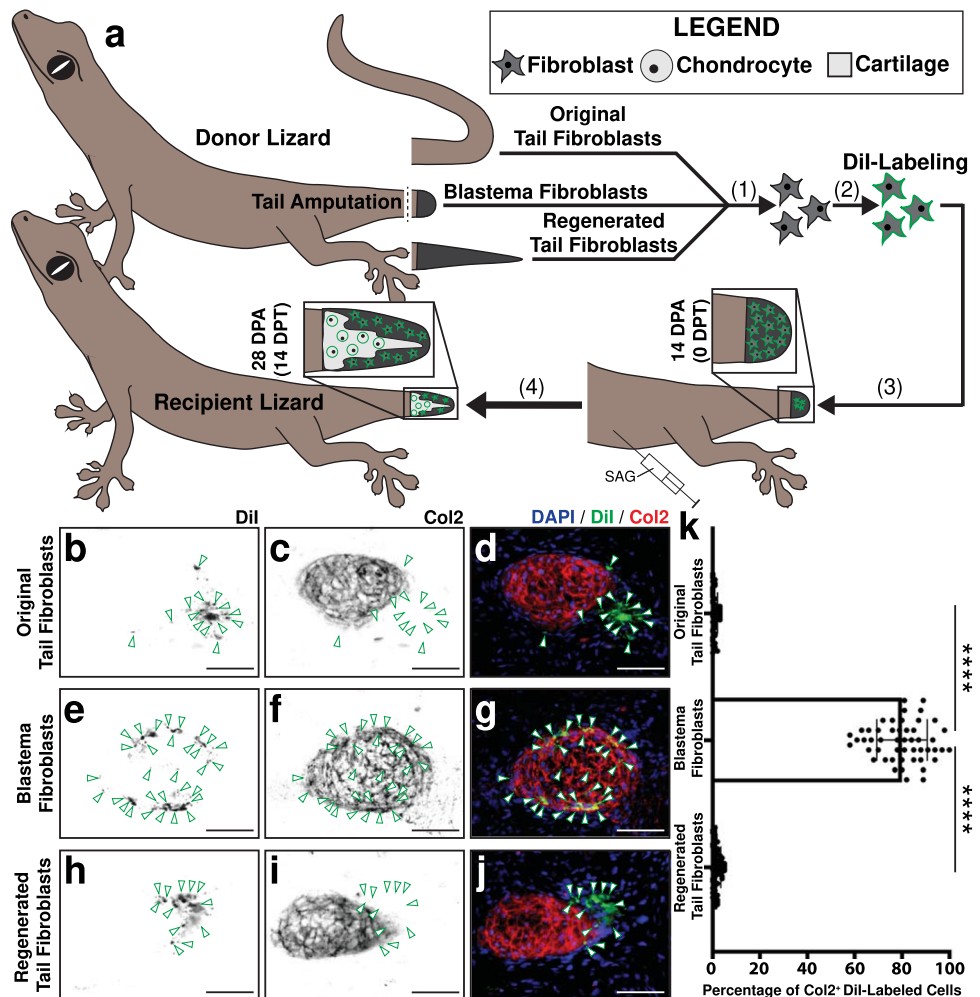

**Fig. 4 | Chondrogenic abilities of fibroblasts derived from original tail, blastema, and regenerated tails in vivo. a** Experimental scheme for *Lepidodactylus lugubris* tail fibroblast transplantations. (1) Donor lizard tails are amputated. FCTCs are isolated from original tails (0 DPA), blastema tails (14 DPA), and regenerated tails (28 DPA). (2) Each tail FCTC pool is labeled with fluorescent DiI, separately. (3) Labeled FCTC pools are transplanted into tail blastemas (14 DPA) of SAG-treated recipient lizards, separately. (4) Following 14 days of SAG treatment post-transplantation (14 DPT), regenerated tails are analyzed via Col2 immunofluorescence staining (IF) and fluorescence microscopy.
**b–j** Representative histological and fluorescent analysis of tails regenerated by SAG-treated lizards pre-injected with DiI-labeled tail FCTCs derived from (**b–d**)

original tail, **e–g** blastema tails, or **h–j** regenerated tails, analyzed by Col2 IF 14 DPT/28 DPA. DiI and Col2 signals are presented separately and together to highlight co-localization or lack thereof. Green arrowheads denote DiI⁺ cells. Bar = 50 μm. **k** Quantification of DiI-labeled cells incorporated within Col2⁺ cartilage regions 14 DPT. *n* = 50 cell counts measured from five images among 10 different animals/tail samples for each condition. Data are presented as mean values +/− standard deviation. One-way Welch's ANOVA for unequal variances and Dunnett's T3 multiple comparisons tests was used. ****adjusted *P* < 0.0001 (Dunnett's T3 multiple comparisons tests). Source data are provided as a Source Data file.

reported to be regulated by Hh stimulation in other systems[60,64,69], and here, we tested the effects of Hh inhibition and activation on tail blastema marker expression (Fig. 3). Amputated lizard limbs, which do not naturally form blastemas, were included in analyses to distinguish blastema-specific markers from nonspecific healing responses[8,70–72] (Fig. 3m–x). Tail blastema and limb samples were collected 14 DPA from the same lizards treated with cyclopamine, SAG, or vehicle control and analyzed by histology/ISH for *sulf1*, *pltp*, *sall1*, and *spp1* to provide context for any Hh signaling dependencies observed among marker expression patterns.

Control tail blastemas expressed *sulf1*, *spp1*, *pltp*, and *sall1* (Fig. 3a–d). *spp1* was expressed at high levels throughout blastemas, while *pltp* and *sall1* were expressed at relatively lower levels (Fig. 3b–d). *Sulf1* exhibited the most defined expression pattern, being localized in areas surrounding ependymal tubes, especially apical blastemas and pre-cartilage tube condensations (Fig. 3a). Cyclopamine treatment significantly reduced tail blastema *sulf1* expression (Fig. 3e)

but did not affect other markers tested (Fig. 3f–h). Conversely, SAG treatment resulted in increased and expanded *sulf1*⁺ blastema areas (Fig. 3i), especially in dorsal and proximal blastema areas, but did not affect *spp1*, *pltp*, or *sall1* expression (Fig. 3j–l).

Amputated limb FCTCs expressed high levels of *spp1* (Fig. 3n), but *sulf1*, *pltp*, and *sall1* were absent from limbs under control conditions (Fig. 3m, o, p). Expression of gene markers assayed was not affected by either cyclopamine or SAG treatments in limbs (Fig. 3q–x), and, unlike tail blastema cells, amputated limb FCTCs did not increase *sulf1* expression in response to SAG treatment (Fig. 3u). These results were confirmed quantitatively via real-time polymerase chain reaction (RT-PCR) analysis of tail samples 14 DPA for *sulf1*, *spp1*, *pltp* and *sall1* (Supplementary Fig. 8). Taken together, these results identified *sulf1*, *pltp*, and *sall1* as specific blastema markers. *Sulf1* was the only blastema marker tested that was particularly responsive to Hh stimulation and inhibition. *Spp1* was further confirmed as a general marker of injury-state FCTCs stimulated by wound healing.

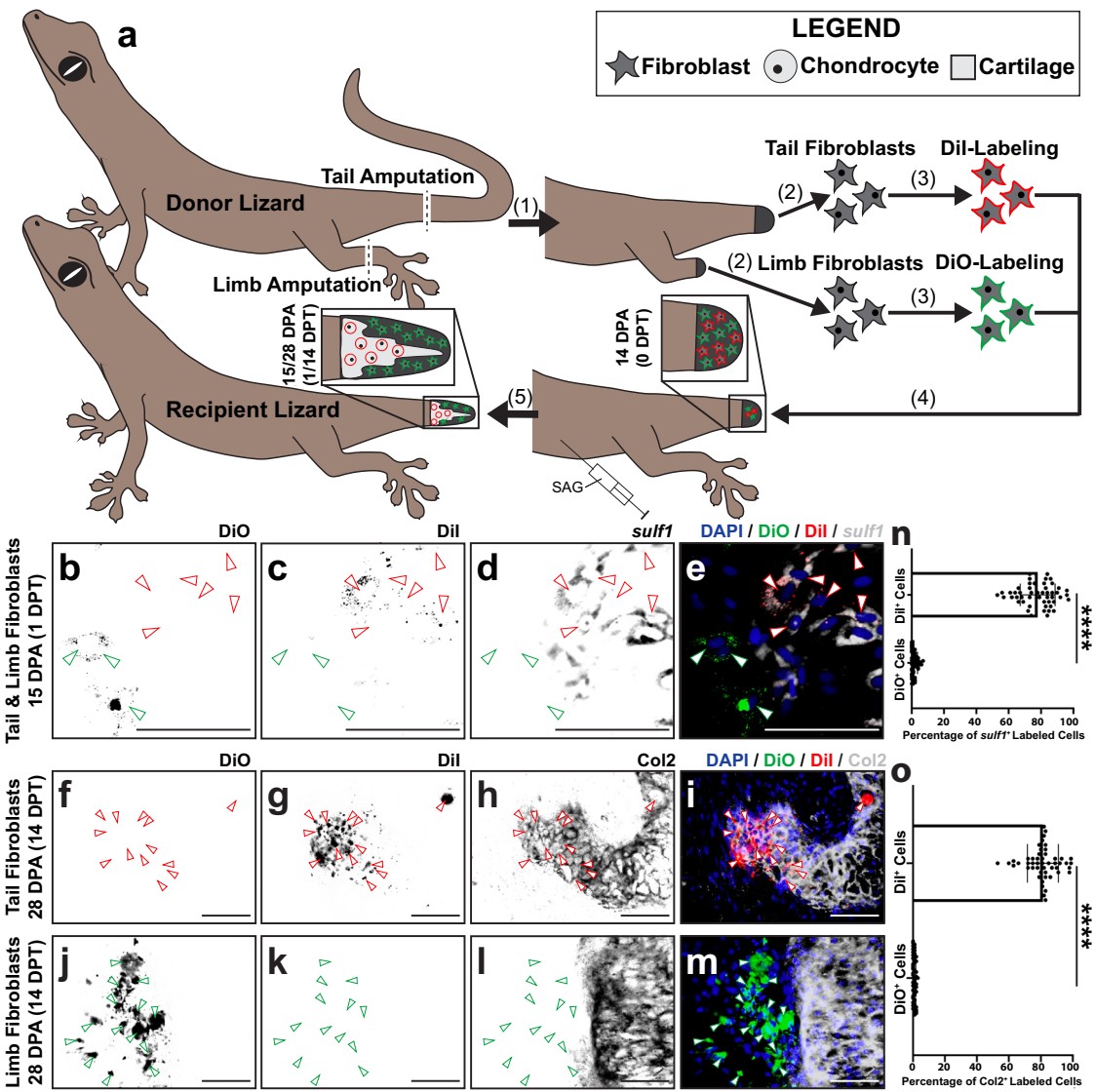

**Fig. 5 | Chondrogenic abilities of *sulf1*⁺ tail blastema cells and *sulf1*⁻ tail fibro-blasts in vivo. a** Experimental scheme for *L. lugubris* tail blastema and amputated limb fibroblast transplantations. (1) Donor lizard tails and left hind limbs are amputated. (2) FCTCs are isolated from tail blastemas and limb stumps 14 DPA. (3) Tail FCTCs are labeled with DiI, while limb FCTCs are labeled with DiO. (4) Labeled FCTCs are co-transplanted into tail blastemas (14 DPA) of SAG-treated recipient lizards. (5) Following 1 and 14 days of SAG treatment post-transplantation, regen-erated tails are analyzed via *sulf1* FISH, Col2 IF, and fluorescence microscopy. **b–m** Representative histological and fluorescent analysis of tails regenerated by SAG-treated lizards pre-injected with DiI-labeled tail FCTCs and DiO-labeled limb

FCTCs, analyzed by **b–e** *sulf1* FISH 1-day post-transplantation (DPT)/15 DPA and **f–m** Col2 IF 14 DPT/28 DPA. DiI, DiO, and *sulf1* or Col2 signals are presented separately and together to highlight co-localization or lack thereof. Red arrow-heads mark DiI⁺ cells, and green arrowheads denote DiO⁺ cells. Bar = 50 μm. **n, o** Quantification of DiI- and DiO-labeled cells (**n**) co-expressing *sulf*1 DPT and **o** incorporated within Col2⁺ cartilage regions 14 DPT. *n* = 50 cell counts measured from five images among ten different animals/tail samples for each condition. Data are presented as mean values +/− standard deviation. Unpaired two-way *t* tests with Welch's correction for unequal variances were used. ****P < 0.0001. Source data are provided as a Source Data file.

### *Sulf1* marks blastema FCTCs with chondrogenic potential

Next, we assessed the chondrogenic capacity of FCTCs from specific tail regeneration stages, to determine which FCTCs were competent to form cartilage, regardless of local tail environmental signaling in vivo, using a transplantation model previously established to trace cell fates during lizard tail regeneration[73]. Cells collected from the parthenoge-netic lizard *Lepidodactylus lugubris* reconstitute regenerated struc-tures following transplantation into amputated tail stumps of lizards belonging to the same clonal population without the need for immu-nosuppressant drug treatments[74], previously shown to negatively impact regeneration[24,75,76] (Fig. 4). Fibroblasts were isolated from donor *L. lugubris* original tails, blastema tails 14 DPA, and regenerated tails 28 DPA, and resulting isolated cell pools were enriched for FCTC populations using physical and enzymatic cell digestion, as well as

MACS® bead treatments (Supplementary Fig. 9). Each FCTC pool was labeled with DiI and injected, separately, into SAG-treated recipient blastema tails 14 DPA. 14 days post-transplant (14 DPT, recipient lizards 28 DPA) recipient tails were analyzed for Col2 via immunofluorescence staining (IF) (Fig. 4a). Neither original nor regenerated tail fibroblasts incorporated into Col2⁺ cartilage by 28 DPA (Fig. 4b–d, h–k), while the majority of transplanted blastema fibroblasts co-stained DiI label and Col2 expression, incorporating into cartilage elements at a sig-nificantly higher rate than original and regenerated tail fibroblasts (Fig. 4e–g, k). This suggested that blastema fibroblasts were uniquely competent to form cartilage in response to Hh stimulation, compared to homeostatic original and regenerated tail fibroblasts.

Previous work has indicated Hh signaling as regulating lizard blastema chondrogenesis[14], and *sulf1* is reported to be expressed in

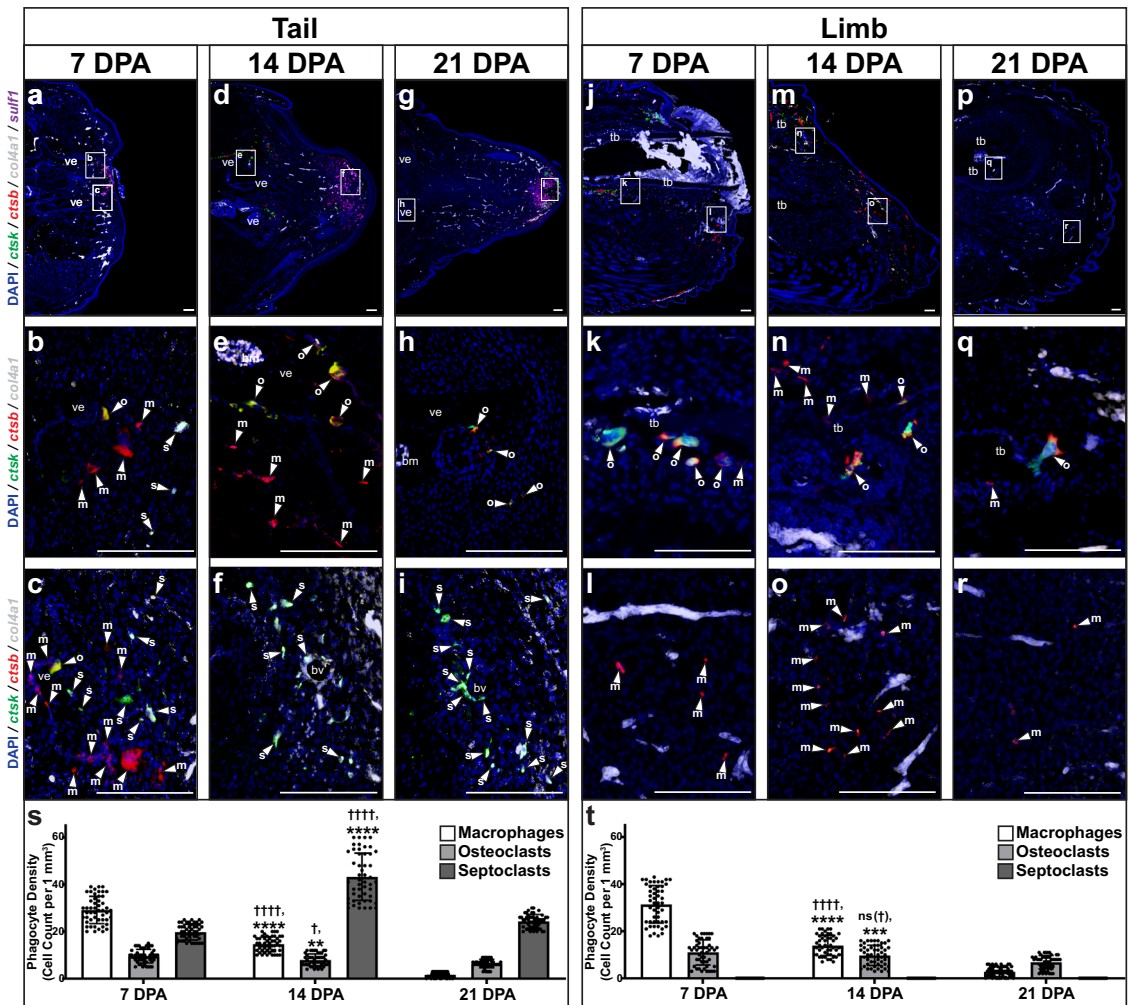

**Fig. 6 | Spatiotemporal localizations of macrophage, osteoclast, and septoclast populations during lizard tail regeneration and limb amputation.** Representative sagittal sections of lizard (*A. carolinensis*) **a**–**i** tails and **j**–**r** limbs collected **a**–**c**, **j**–**l** 7 DPA, **d**–**f**, **m**–**o** 14 DPA, and **g**–**i**, **p**–**r** 21 DPA analyzed by histology/FISH for *ctsb, ctsk, col4a1,* and *sulf1* expression. Higher magnification views of corresponding regions in (**a, d, g, j, m, p**) highlighting *ctsb⁺ ctsk⁻ col4a1⁻* macrophages (m, arrowhead), *ctsb⁺ ctsk⁺ col4a1⁻* osteoclasts (o, arrowhead), and *ctsb⁻ ctsk⁺ col4a1⁺* septoclasts (s, arrowhead) in regions proximal to **b, e, h** vertebra, **k, n, q** tibia, **c, f, i** distal tail and **l, o, r** distal limb. bm bone marrow, bv blood vessel, tb tibia, ve vertebra. Bar = 100 μm. **s, t** Quantification of macrophage, osteoclast, and septoclast densities measured in lizard **s** tails and **t** limbs collected 7, 14, and 21 DPA. *n* = 50 cell densities measured from five images among 10 different animals/samples for each time point. Data are presented as mean values +/− standard deviation. One-way Welch's ANOVA for unequal variances and Dunnett's T3 multiple comparisons tests were used. †adjusted *P* = 0.0139; ††††adjusted *P* < 0.0001; ns (†), not significant (adjusted *P* = 0.4663), compared to corresponding cell type 7 DPA (Dunnett's T3 multiple comparisons tests). **adjusted *P* = 0.0024; ***adjusted *P* = 0.0005; ****adjusted *P* < 0.0001, compared to corresponding cell type 21 DPA (Dunnett's T3 multiple comparisons tests). Source data are provided as a Source Data file.

pre-condensing mesenchyme during chondrogenesis[77, 78]. Here, we tested co-localization of *sulf1* with *sox9*, the transcription factor regulating chondrogenesis, within lizard tail blastemas 14 DPA in response to treatment with cyclopamine, SAG, and vehicle control. (Supplementary Fig. 10). Control tail blastemas exhibited proximodistal gradients of *sulf1* and *sox9* expression in FCTCs surrounding ependymal tubes (Supplementary Fig. 10a, b). *Sulf1* expression localized in distal apical blastema regions and reduced proximally as it was replaced by *sox9*. *Sox9* exhibited its strongest expression in proximal skeletal elements adjacent to original tail vertebrae at amputation planes. Cyclopamine treatment reduced both *sulf1* and *sox9* expression and interrupted proximodistal marker localizations (Supplementary Fig. 10c, d). In SAG-treated tails, both *sulf1⁺* and *sox9⁺* areas expanded peripherally into regions removed from ependymal tubes, but proximodistal expression relationships were maintained (Supplementary Fig. 10e, f).

The above results suggested a relationship between Hh signaling, *sulf1* expression, and chondrogenesis. Here, we tested this relationship by comparing the abilities of *sulf1⁺* and *sulf1⁻* lizard fibroblasts to undergo chondrogenesis in vivo using the *L. lugubris* transplantation model (Supplementary Fig. 11). First, donor original (0 DPA) tail and limb FCTCs were pre-labeled with the fluorescent dyes DiI and DiO, respectively. Labeled tail and limb FCTCs were mixed, and co-transplanted into recipient lizard tails at the time of recipient lizard tail amputation (0 DPA). Recipient lizards were treated with SAG, and tails were collected 14 and 28 DPT for integration assessment (Supplementary Fig. 11a). FISH and histological analysis of 14 DPT samples showed DiI⁺ tail and DiO⁺ limb-derived fibroblasts expressed *sulf1* (Supplementary Fig. 11b–e, j). Similarly, both DiI⁺ and DiO⁺ fibroblasts formed Col2⁺ cartilage in 28 DPT samples assessed by IF (Supplementary Fig. 11f–i, k). These results suggested that both tail and limb fibroblasts possess the capacity for *sulf1* expression and chondrogenesis when exposed to the blastema formation process and signaling niche.

Finally, we compared the above results with those observed when exogenous tail blastema and limb FCTCs were transplanted into

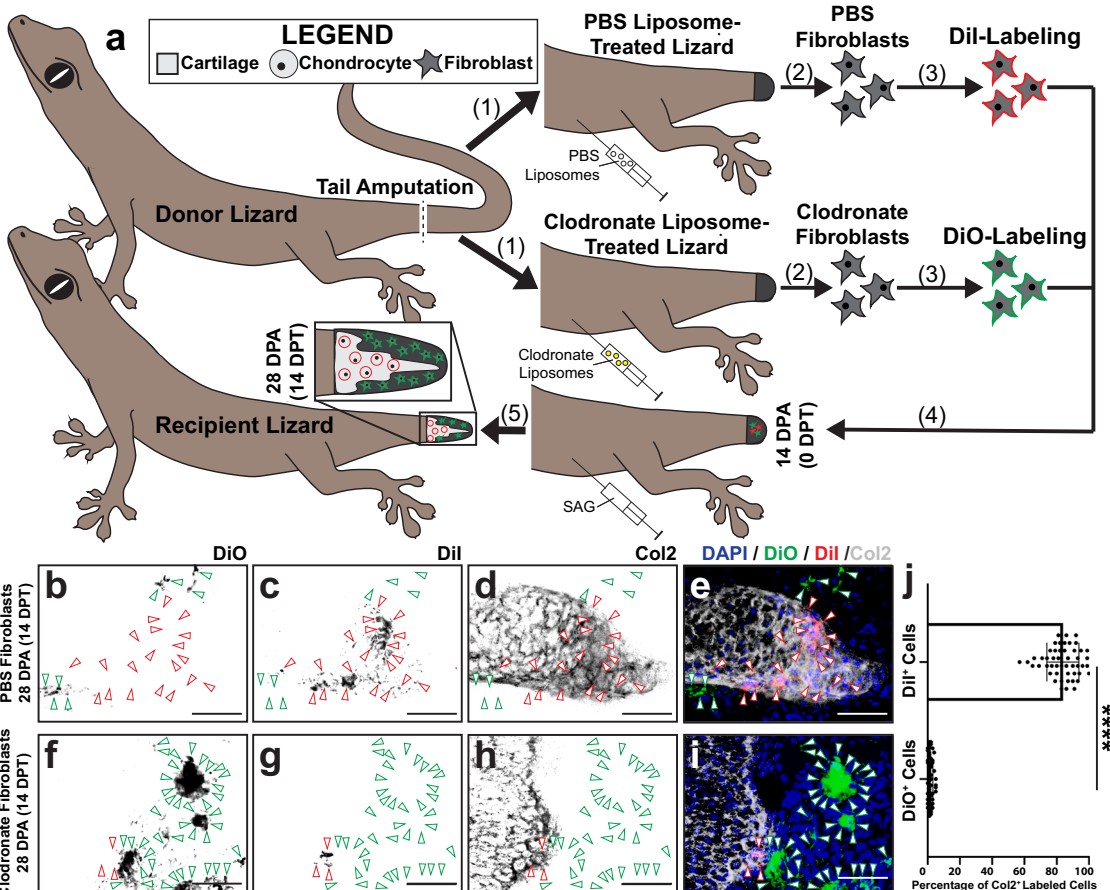

**Fig. 7 | Pre-exposure to phagocytic cells is required for activation of Hedgehog-responsive chondrogenesis in lizard fibroblasts. a** Experimental scheme for applying clodronate liposome treatments and fibroblast transplantations in lizards (*L. lugubris*) toward investigating dependencies of blastema cell chondrogenesis on phagocyte populations. (1) Tails of PBS liposome- and clodronate liposome-treated donor lizards are amputated. (2) Tail fibroblasts are isolated from PBS liposome- and clodronate liposome-treated lizard tails 14 DPA. (3) PBS lizard-derived fibroblasts are labeled with DiI, while clodronate fibroblasts are labeled with DiO. (4) Labeled cells are co-transplanted into 14 DPA tail blastema of recipient SAG-treated lizards. (5) Following 14 days of SAG treatment post-transplantation, regenerated tails are analyzed via Col2 IF and fluorescence microscopy. **b–i** Representative histological analysis of tails regenerated by SAG-treated lizards pre-injected with DiI-labeled PBS liposome-treated FCTCs and DiO-labeled clodronate liposome-treated FCTCs and analyzed by histology, DiI and DiO fluorescence, and Col2 IF 14 DPT/28 DPA. Bar = 50 μm. **j** Quantification of DiI- and DiO-labeled cells co-expressing Col2 14 DPT. $n = 50$ cell counts measured from five images among ten different animals/tails for each condition. Data are presented as mean values +/− standard deviation. Unpaired two-way *t* test with Welch's correction for unequal variances was used. ****$P < 0.0001$. Source data are provided as a Source Data file.

recipient tails that had already formed blastemas. Fibroblasts isolated from donor *L. lugubris* tail blastemas and limbs 14 DPA, pre-labeled with the DiI and DiO, respectively, were co-transplanted into SAG-treated recipient lizard tail blastemas (Fig. 5). Tails were collected 1 and 14 DPT and analyzed as described above to compare percentages of exogenous tail blastema and limb fibroblasts that expressed *sulf1* via FISH or Col2 via IF (Fig. 5a).

In 1 DPT samples, DiI⁺ tail blastema-derived fibroblasts colocalized with *sulf1* expression, while DiO⁺ limb-derived fibroblasts did not (Fig. 5b–e, n). 14 DPT samples analyzed via IF for Col2 expression revealed significantly higher percentages of tail blastema FCTCs forming Col2⁺ cartilage than limb fibroblasts (Fig. 5f–o). Tail and limb fibroblasts were also transplanted separately into vehicle control- or SAG-treated recipient blastemas as controls (Supplementary Fig. 12). Only tail fibroblasts stimulated by SAG co-expressed DiI and Col2 (Supplementary Fig. 12d–f, m), while tail and limb FCTCs transplanted into vehicle control recipients and limb fibroblasts transplanted into SAG-treated recipient blastemas did not express Col2 or form cartilage (Supplementary Fig. 12a–c, g–m). Taken together, these results confirmed the abilities of *sulf1*⁺ blastema FCTCs, but not *sulf1*⁻ limb FCTCs, to undergo chondrogenesis in response to Hh stimulation.

## Tail regrowth involves distinct localizations of phagocytes

Further single-cell sequencing analyses of lizard tail regeneration investigated macrophages and other phagocytic cells, revealing heterogeneity and distinct cell populations delineated by differential marker gene expression and nearest-neighbor clustering (Supplementary Fig. 13a–c). *Ctsb*⁺ cathepsin K-negative (*ctsk*) macrophages and *ctsb*⁺ *ctsk*⁺ osteoclasts clustered with other immune cells, as expected (Supplementary Fig. 13a, b). However, a distinct *ctsb*⁻ *ctsk*⁺ population clustered with collagen type IV alpha 1 chain (*col4a1*⁺) pericytes (Supplementary Fig. 13a–d). Septoclast populations, phagocytic cells derived from a pericytic rather than myeloid lineage, have previously been shown to regulate skeletal development and healing in mammals[79], and we hypothesized that *ctsb*⁻ *ctsk*⁺ *col4a1*⁺ cells represented lizard septoclast-like phagocytic cells (herein, referred to as septoclasts).

Single-cell sequencing results were validated via histology/FISH in lizard tails and limbs collected 7, 14, and 21 DPA. (Fig. 6). *Ctsb*⁺ *ctsk*⁻ *col4a1*⁻ macrophage levels peaked in tails at 7 DPA (Fig. 6a–c, s) before diminishing by 21 DPA (Fig. 6g–i, s). *Ctsb*⁺ *ctsk*⁺ *col4a1*⁻ osteoclast population numbers were highest in tails at 7 and 14 DPA before decreasing by 21 DPA (Fig. 6a–i, s). Tail osteoclasts were exclusively

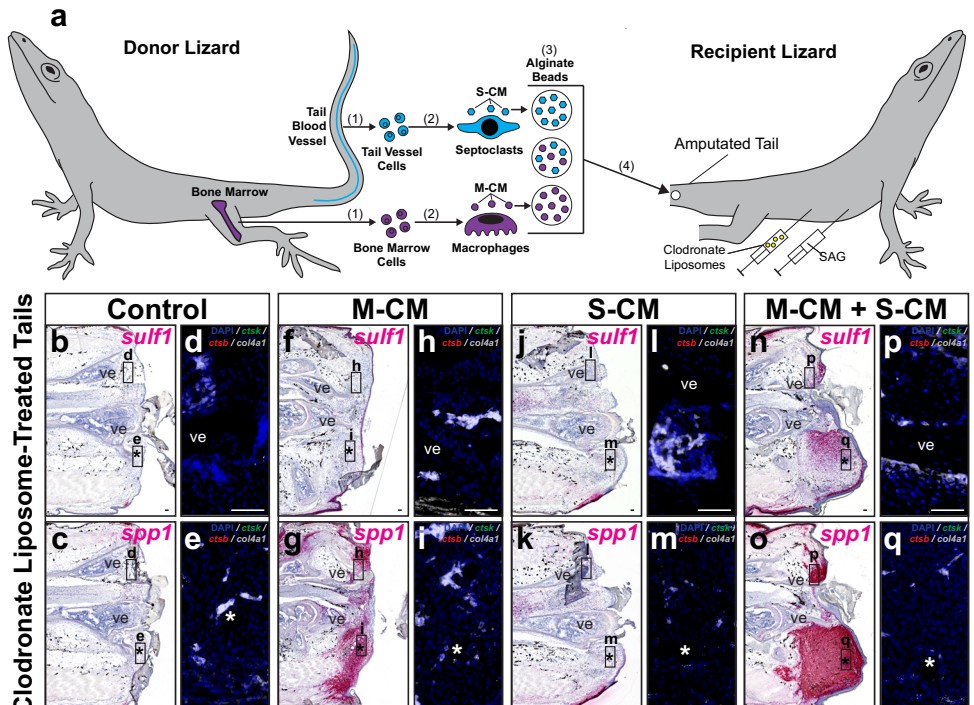

**Fig. 8 | Factors secreted by lizard macrophage and septoclast populations rescue tail FCTC marker gene expression and chondrogenic potential following endogenous phagocyte depletion. a** Experimental scheme for testing effects of biomolecules secreted by macrophage and septoclast populations on FCTC gene expression in clodronate liposome-treated lizard (*A. carolinensis*) tails. (1) Bone marrow and tail blood vessel cells are isolated from donor lizards and (2) differentiated into macrophages and septoclasts in vitro, respectively. (3) Macrophage- and septoclast-conditioned media (M-CM and S-CM, respectively) are collected, concentrated, and absorbed by alginate beads, both separately and together. (4) Alginate beads are implanted into amputated tails of lizards (0 DPA) co-treated with clodronate liposomes and SAG. **b–q** Representative sagittal sections of lizard tails 14 DPA co-treated with SAG and clodronate liposomes, implanted with M-CM and/or S-CM beads, and analyzed via histology/ISH/FISH for *sulf1, spp1, ctsk, ctsb,* and *col4a1* expression. Higher magnification fluorescent views of (**b, c, f, g, j, k, n, o**), respectively, depict indicated regions around **d, h, l, p** vertebra and **e, i, m, q** implanted beads. *n* = 8 lizards/samples per treatment condition tested. *, location of implanted bead; ve, vertebra. Bar = 50 μm.

associated with vertebrae, occupying classic crypts along periosteal surfaces (Fig. 6a–i). *Ctsb⁻ ctsk⁺ col4a1⁺* septoclast levels peaked in tails during blastema formation at 14 DPA (Fig. 6a–i, s) and were detected among *sulf1⁺* blastema fibroblast populations (Fig. 6a–i). In addition, tails collected 0, 1, 3, 7, and 14 DPA confirmed *ctsb⁺* macrophage and osteoclast populations peaked at 7 DPA within the span of immune cell-active inflammatory stage time points 1–7 DPA (Supplementary Fig. 14). Unlike tail macrophage and osteoclasts, however, elevated septoclast numbers in tails persisted beyond blastema stages and, at 21 DPA, occupied *sulf1⁺* apical regions at regenerating tail tips (Fig. 6g–i). Taken together, these results suggested a relationship between septoclasts and blastema cell state.

Limbs exhibited macrophage and osteoclast populations that followed similar spatiotemporal trends to time-matched tail cell populations (Fig. 6). Limb macrophages were associated with wound epidermis and, along with bone-associated osteoclasts, peaked prior to 14 DPA before diminishing further at 21 DPA (Fig. 6j–r, t). However, limbs did not exhibit identifiable septoclasts at any time points tested and did not express *sulf1* (Fig. 6j–r, t). Given the lack of Hh-responsive chondrogenesis exhibited by amputated lizard limbs described above, these results further supported the links among lizard septoclasts, blastema formation, *sulf1* expression, Hh sensitivity, and cartilage formation, which are investigated further below.

**Blastema formation and chondrogenesis depend on phagocytes**
We previously validated clodronate liposome treatments as effective methods for depleting lizard phagocyte/macrophage populations and inhibiting blastema formation in *A. carolinensis*[9]. Here, we tested the effect of clodronate liposome treatment on macrophage, osteoclast, and septoclast population levels and fibroblast marker expression during lizard tail blastema formation (Supplementary Fig. 15). Lizards were pre-treated with clodronate or vehicle control phosphate-buffered saline (PBS) liposomes 72 and 48 h prior to tail amputation, and tails were collected at the blastema stage 14 DPA for analysis.

Histology revealed clodronate liposome treatment inhibited blastema formation compared to controls (Supplementary Fig. 15a, b, f, g) and resulted in depletion of *ctsb⁺ ctsk⁻* macrophages, *ctsb⁺ ctsk⁺* osteoclasts and *ctsb⁻ ctsk⁺ col4a1⁺* septoclasts (Supplementary Fig. 15c–e, h–j), as expected. Clodronate-treated tails also lost *spp1* and *sulf1* expression in FCTCs (Supplementary Fig. 15a, b, f, g). Control blastema revealed macrophages and osteoclasts localized to *sulf1⁻* areas of the blastema, while septoclasts were only detected distal to *sulf1⁺* areas of the tail (Supplementary Fig. 15c–e). Co-staining of *spp1* and *sulf1* at 7 and 14 DPA in untreated blastema revealed *spp1* expression at both 7 and 14 DPA, while large areas of *sulf1⁺* blastema cells co-expressed *spp1*, confirming both markers are active in a subpopulation of blastema FCTC (Supplementary Fig. 16). Taken together, these results established a link between lizard phagocytic lineages and FCTC marker gene acquisition.

Next, we tested the effects of clodronate liposome pre-treatment on lizard FCTC responsiveness to Hh signaling in *L. lugubris* blastema fibroblast transplantation studies (Fig. 7). This model was found to be particularly applicable for testing exogenous FCTC chondrogenic potential in situ when endogenous conditions are not conducive to blastema/cartilage formation. *L. lugubris* lizards were pre-treated with clodronate liposomes (validated in Supplementary Fig. 17), or control

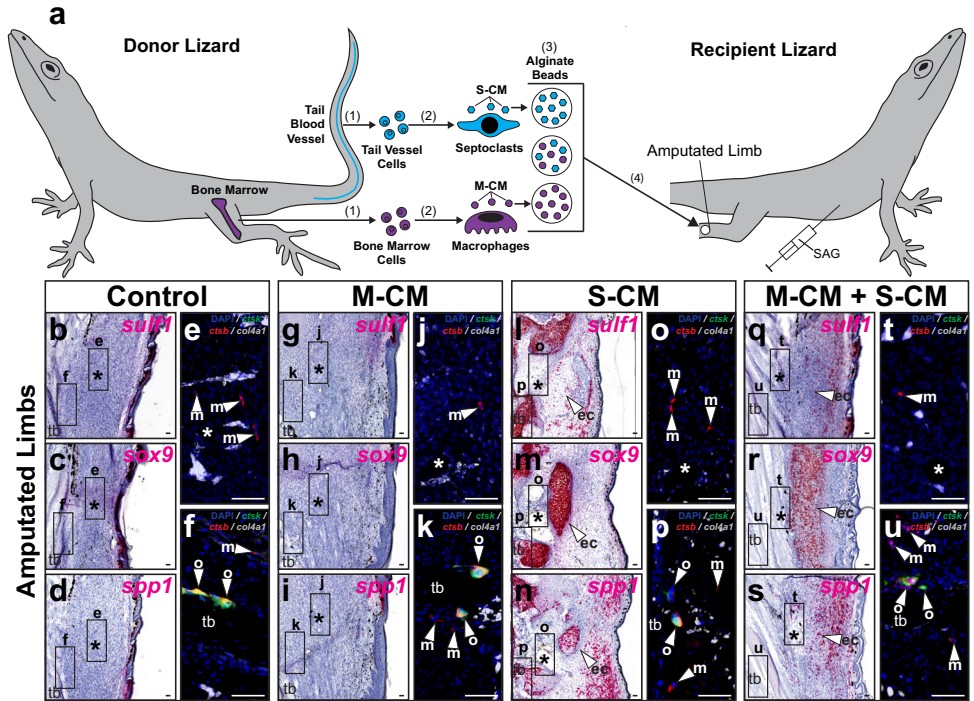

**Fig. 9 | Lizard macrophage- and septoclast-secreted factors stimulate FCTCs and promote chondrogenesis in amputated lizard limbs. a** Experimental scheme for testing effects of macrophage- and septoclast-conditioned media on native fibroblasts in lizard (*A. carolinensis*) limbs. (1) Bone marrow and tail blood vessel cells are isolated and (2) used to derive macrophages and septoclasts in vitro, respectively. (3) Media conditioned by macrophages and septoclasts (M-CM and S-CM) are collected, concentrated, and absorbed by alginate beads, both separately and together. (4) Alginate beads are implanted into amputated limbs of SAG- treated lizards 7 DPA. **b–u** Representative sagittal sections of lizard limbs 28 DPA treated with SAG, implanted with M-CM and/or S-CM beads, and analyzed via histology/ISH/FISH for *sulf1, sox9, spp1, ctsk, ctsb,* and *col4a1* expression. Higher magnification fluorescent views of (**b–d, g–i, l–n, q–s**), respectively, depict indicated regions around **e, j, o, t** implanted beads and **f, k, p, u** tibia. *n* = 8 lizards/ samples per treatment condition tested. *, location of implanted bead; ec (arrowhead), ectopic cartilage; m (arrowhead), macrophage; o (arrowhead), osteoclast; tb, tibia. Bar = 50 μm.

PBS liposomes prior to amputation, and fibroblasts were isolated from tails 14 DPA. Fibroblasts collected from PBS and clodronate liposome-treated lizards were pre-labeled with DiI or DiO, respectively, before co-transplanting into a separate cohort of SAG-treated recipient lizard blastemas (Fig. 7a).

Tails were collected 14 DPT and analyzed for the contribution of DiO⁺ and DiI⁺ cells to cartilage formation (Fig. 7b–j). Significantly higher levels of DiI⁺ control tail FCTCs underwent chondrogenesis, incorporating into Col2⁺ areas of condensing cartilage, compared to DiO⁺ clodronate-treated FCTCs, suggesting that blastema fibroblasts contributed to cartilage regions while fibroblasts derived from clodronate-treated tails, lacking blastemas, did not (Fig. 7b–j). Clodronate and PBS liposome-treated fibroblasts transplanted separately into SAG- or vehicle control-treated recipient blastemas revealed similar results (Supplementary Fig. 18), with only PBS liposome-treated fibroblasts incorporating into Col2⁺ cartilage (Supplementary Fig. 18d–f, m), while clodronate liposome fibroblasts in SAG-treated recipients and PBS or clodronate liposome-treated fibroblasts transplanted into vehicle control blastema did not express Col2 (Supplementary Fig. 18a–c, g–m). Taken together, these results established a dependency of lizard blastema cell Hh-responsive chondrogenesis on pre-conditioning by phagocyte populations.

**Phagocyte-conditioned media rescues blastema formation**

We have previously established protocols for isolating and differentiating phagocyte populations from multiple lizard tissues[9]. Given the evidence described above linking septoclasts with pericytes, we hypothesized that pericyte-rich tail blood vessels represented effective sources of septoclasts. Caudal blood vessels were isolated from lizard tails and subjected to phagocyte isolation protocols. Lizard bone

marrow were utilized as a source of macrophages, as previously validated[9]. Phagocytes differentiated from caudal blood vessels and bone marrow cells were analyzed by phagocytosis assays, flow cytometry and by IF/FISH for Ctsk, *ctsb*, integrin subunit alpha M (*itgam/cd11b*), and *col4a1* expression (Supplementary Fig. 19).

Both caudal vessel- and bone marrow-derived cells exhibited similarly high levels of phagocytosis (Supplementary Fig. 19a–c, g) and expressed *itgam/cd11b* (Supplementary Fig. 19d, h), validating their identities as phagocytes. However, blood vessel and bone marrow phagocytes exhibited differential marker expressions that mirrored differences observed in vivo (Supplementary Fig. 19d–f, h–j); bone marrow phagocytes expressed macrophage marker *ctsb* (Supplementary Fig. 19d–f), while caudal vessel phagocytes expressed septoclast markers Ctsk and *col4a1* (Supplementary Fig. 19h–j). Flow cytometry confirmed the purity of respective macrophage and septoclast populations, with an average of 94.5% of cells within the bone marrow phagocyte pools expressing Ctsb⁺ Ctsk⁻ macrophage signatures and more than 98% of caudal vessel phagocyte cells expressed Ctsb⁻ Ctsk⁺ septoclast signatures (Supplementary Fig. 19k–p). Taken together, these results validated *ctsb⁻ ctsk⁺ col4a1⁺* caudal blood vessel-derived phagocyte cultures as septoclasts and *ctsb⁺ ctsk⁻ col4a1⁻* bone marrow-derived phagocyte cultures as macrophages.

Next, we tested the abilities of septoclast populations to rescue the lizard tail blastema cell chondrogenic-potential state, defined by Hh-responsive *sulf1* expression, following clodronate liposome treatment (Fig. 8 and Supplementary Fig. 20). Macrophage-conditioned media (M-CM) and septoclast-conditioned media (S-CM) were collected from bone marrow- and caudal vessel-derived phagocytes, respectively, and concentrated. Alginate beads soaked in concentrated M-CM and/or S-CM were implanted into amputated tails of lizards co-

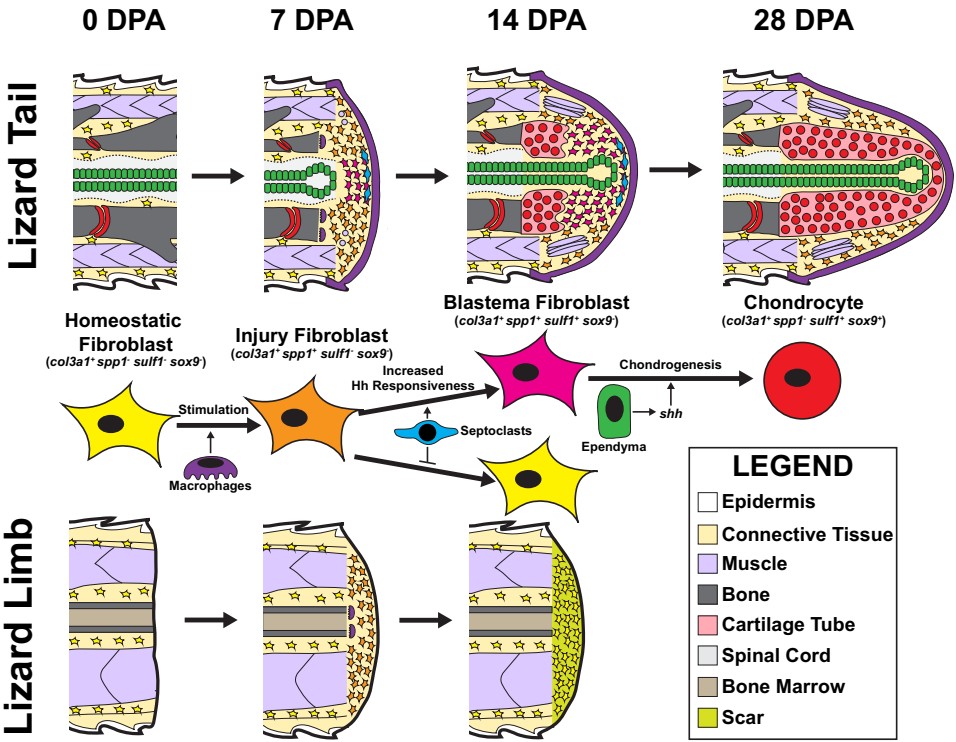

**Fig. 10 | Proposed mechanism of FCTC gene expression acquisition and chondrogenesis during lizard tail regeneration.** *Col3a1*⁺ homeostatic original tail FCTCs (0 DPA) are stimulated by macrophage paracrine signals and express *spp1* in response to injury by 7 DPA. Factors secreted by tail septoclasts maintain *spp1* expression in FCTCs and enhance FCTC sensitivity to Hedgehog (Hh) signals contributed by blastema ependymal tubes. Hh stimulation induces *sulf1* expression in septoclast-stimulated blastema FCTCs by 14 DPA. *Sulf1*⁺ FCTCs enter a chondrogenic program in response to Hh signals and express *sox9* and *col2a1*, differentiating into chondrocytes by 28 DPA. Amputated lizard limbs provide context for

tail blastema formation and regrowth as non-regenerative appendages that exhibit FCTC mobilization and state changes, but not chondrogenesis. *Col3a1*⁺ homeostatic limb fibroblasts express *spp1* at 7 DPA in response to macrophage infiltration following amputation. However, *spp1* expression is not maintained without stimulation from septoclasts, which are absent in amputated lizard limbs. Without exposure to septoclast-secreted factors, limb FCTCs lose *spp1* expression by 14 DPA, remain unresponsive to Hh signaling, and do not express *sulf1* or undergo chondrogenesis. Instead, amputated lizard limbs form scars.

treated with clodronate liposomes and either SAG (Fig. 8a) or vehicle control (Supplementary Fig. 20a). Tails were collected 14 days post-implantation and analyzed by histology/ISH/FISH for *sulf1* and *spp1* expression and phagocytic cell markers (Fig. 8b–q and Supplementary Fig. 20b–q).

*Spp1* signal was detected in M-CM-treated tails regardless of SAG/vehicle control treatment (Fig. 8g, o and Supplementary Fig. 20g, o), but was absent in S-CM-only treated conditions (Fig. 8k and Supplementary Fig. 20k). *Sulf1* expression was only detected in tails co-treated with SAG and both M-CM and S-CM (Fig. 8n and Supplementary Fig. 20n). *Spp1*⁺ and *sulf1*⁺ tail regions concentrated around implanted beads in M-CM and S-CM-treated samples (Fig. 8g, n, o). Endogenous *ctsb*⁺ *ctsk*⁻ *col4a1*⁻ macrophages, *ctsb*⁺ *ctsk*⁺ *col4a1*⁻ osteoclasts, and *ctsb*⁻ *ctsk*⁻ *col4a1*⁺ septoclasts were not detected in any of the conditions tested, as expected due to clodronate liposome treatment (Fig. 8 and Supplementary Fig. 20).

Unconditioned culture media-soaked beads were utilized as a control and implanted into clodronate liposome-treated tails (Supplementary Fig. 21), revealing no changes in *spp1* or *sulf1* expression in vehicle control or SAG-treated recipient tails (Supplementary Fig. 21a–d), signifying phagocyte-conditioned media treatments were responsible for changes in gene expression rather than bead implantation or culture media alone. Taken together, these results suggested that biomolecules secreted by lizard septoclasts were necessary for rescuing hallmarks of blastema formation including *sulf1* in response to Hh stimulation even when endogenous septoclast populations had been depleted. While macrophage-secreted factors were sufficient for inducing FCTC *spp1* expression, *sulf1* was only detected in

tails treated with both macrophage- and septoclast-conditioned media, indicating a sequential addition of FCTC marker gene expression during blastema establishment.

**Septoclast-CM induces cartilage formation in amputated limbs**
Above comparisons of amputated lizard tail and limb healing suggested a link between FCTC chondrogenesis and septoclast populations, and, here, the effects of S-CM on lizard limb *sulf1*, *spp1*, and *sox9* expression were tested (Fig. 9 and Supplementary Fig. 22). Beads soaked in S-CM and/or M-CM were implanted in lizard limbs 7 DPA. After 21 days of treatment with SAG (Fig. 9) or vehicle control (Supplementary Fig. 22), limbs were collected and analyzed via histology/ISH for FCTC and chondrogenesis marker expression and via FISH for phagocyte markers (Fig. 9a and Supplementary Fig. 22a).

FCTCs in control limbs and limbs implanted with M-CM beads without S-CM did not express *sulf1*, *sox9*, or *spp1* regardless of SAG/vehicle control treatments (Fig. 9b–d, g–i and Supplementary Fig. 22b–d, g–i). Implanted S-CM beads, with and without M-CM, induced FCTC *sulf1* and *sox9* expression around bead implantation sites, but only in response to co-treatment with SAG (Fig. 9l, m, q, r and Supplementary Fig. 22l, m, q, r). S-CM beads, again, regardless of M-CM addition, induced *spp1* expression in limbs of both SAG- and vehicle control-treated lizards (Fig. 9n, s and Supplementary Fig. 22n, s). Endogenous macrophages and osteoclasts, but not septoclasts, were detected in all conditions tested and levels were unaffected by drug or bead treatment regardless of conditioned media type (Fig. 9 and Supplementary Fig. 22).

Unconditioned media treatment did not induce gene expression changes or chondrogenesis in vehicle control or SAG-treated recipient limbs (Supplementary Fig. 21e–j). These results suggested that exogenous septoclast-derived signals were necessary and sufficient for supplementing the naturally septoclast-deficient amputated limb environment, introducing Hh-responsive chondrogenesis to amputated lizard limbs. Lizard septoclast cell factors also maintained *spp1* expression in lizard limbs until at least 28 DPA. (Fig. 9n, s and Supplementary Fig. 22n, s). Since loss of *spp1* expression by lizard limb FCTCs 28 DPA coincides with scar formation by 28 DPA (Supplementary Fig. 7k), these results suggest a role for lizard septoclast populations in the inhibition of fibrosis.

In summary, the results described above suggest the following mechanisms of sequential fibroblast marker gene acquisition during lizard tail blastemal formation (Fig. 10). *Col3a1*+ resting fibroblasts in both tail and limb respond to amputation injury with expression of *spp1*, along with other injury-state FCTC marker genes such as *col12a1* and *mdk*. Injury-state FCTCs migrate to amputation sites following infiltration of macrophage populations and signaling from macrophage-secreted factors. Lizard tails, but not limbs, exhibit septoclast cell populations following amputation injuries that induce increased FCTC Hh sensitivity. FCTCs exposed to septoclast-secreted factors maintain *spp1* and express *sulf1* and *sox9* following stimulation by *shh*, produced by ependymal cells. This spatial patterning results in cartilage forming around blastema ependymal tubes. Without septoclasts, amputated limb FCTCs do not maintain *spp1* expression and remain unresponsive to Hh signaling. Limb FCTCs do not express *sulf1* or undergo chondrogenesis, even when treated with exogenous Hh signals, ultimately forming scars instead of cartilage.

## Discussion

As reptiles, lizards occupy a unique, intermediate position between amphibians and mammals that is reflected in their regenerative biology, and this study identifies potential cellular and molecular mechanisms underlying several of these distinctions. Amphibian urodeles exhibit the most complete and drastic regenerative capabilities among tetrapods, with most species retaining the ability to regrow near-perfect copies of both limbs and tails following amputation[80–82]. Conversely, lizards are the only known amniotes and the closest relatives of mammals capable of multi-tissue, blastema-based appendage regeneration as adults. Some lizard species can regrow amputated tails, but never limbs, and regenerated lizard tails are referred to as "imperfect regenerates" due, in part, to a bias for forming cartilage over osseous skeletal tissues[8, 9,12–14,16–18]. The goal of this study was to determine the cell types and pathways involved in lizard blastema formation and subsequent cartilage differentiation.

This study leveraged the power of single-cell sequencing methodologies and pseudotime trajectory analysis to perform a comprehensive molecular interrogation into the heterogeneity of lizard tail blastema cell populations. Our results, identifying FCTCs as the primary contributors to lizard blastemas and regenerated tail cartilage, align with classical and modern single-cell studies analyzing appendage regeneration in salamanders[28,36–40,83]. However, our lizard results suggested a sequential addition of FCTC markers that diverge from the de-differentiation processes reported in amphibian studies. Specifically, single-cell and ISH results confirmed that *col3a1*+ fibroblasts increase expression of *spp1, mdk, tnl, sparc*, and *col12a1* upon amputation injury, regardless of ultimate regenerate outcome. Only FCTCs that subsequently contribute to tail blastema formation add expression of *sulf1, pltp*, and *sall1*. Despite computational limitations of pseudotime trajectory analysis without additional cell lineage information[84], the suggestive pseudotime model of sequential addition of FCTC marker expression was tested experimentally and supported with fibroblast transplantation studies.

As previously mentioned, *sulf1* specifically marks blastema cells capable of entering the cartilage differentiation and formation program. This hierarchical addition of mature connective tissue markers throughout the injury, blastema, and chondrogenic processes is distinct from salamander blastema formation, wherein FCTCs revert to embryonic mesenchymal states that are transcriptionally distinct from resting populations, utilizing genes previously activated in tail embryogenesis[36,85]. Previous studies have also identified structural differences in urodele and lizard blastema cells that may reflect differences in de-differentiation and cell states, which could influence plasticity during regeneration[86].

Furthermore, our results suggest that lizard tail regrowth is distinguished from amphibian appendage regeneration by the retention of injury markers in mature regenerates. While blastema cells undergoing chondrogenesis eventually lose injury markers *spp1, mdk, tnl, sparc*, and *col12a1*, these markers are retained in non-cartilage connective tissue in mature regenerates. Conversely, amphibian blastema FCTCs are reported to revert to cell states nearly transcriptionally identical to uninjured conditions following differentiation into replacement connective tissues. These differences between lizard and amphibian blastema FCTC populations may have repercussions for the regenerative outcomes and fidelities of blastema-derived skeleton. Direct unbiased comparison of regenerating lizard and axolotl tail scRNAseq datasets could provide more comprehensive distinctions between respective blastema fibroblast transcriptional profiles and identify additional differential gene expression changes between the two species' cell populations.

Differences between lizard and salamander blastema cell states may account for the inability of lizard blastema FCTCs to differentiate into bone. There is precedence for such a hypothesis, as we have reported on similar potency deficiencies in lizard versus salamander neural progenitor cells (NPCs)[87]. Adult lizard NPCs are unable to differentiate into roof plate identities and undergo neurogenesis, resulting in a lack of dorsoventral patterning and new neurons in regenerated lizard tails. Future work will be aimed at determining whether similar limitations in lizard blastema FCTC differentiation capacities underlie the lack of osteogenesis in regenerated lizard tails.

Lizards are one of the only adult vertebrates that combine regenerative appendages (i.e., tails), and non-regenerative appendages (i.e., limbs) in the same animal, affording the lizard model with unique opportunities for study. For example, comparing tail blastema versus amputated limb wound healing, similar to previous studies[88–90], allowed for the identification of blastema-specific markers by eliminating limb transcripts associated with general healing mechanisms. Here, we were able to leverage this strategy to classify several fibroblast injury markers, such as *spp1, mdk, col12a1*, etc., as general markers of amputation wound healing. Conversely, we identified a subset of injury markers, including *sulf1, pltp*, and *sall1*, as specific for blastema formation.

*Sulf1* was found to be particularly sensitive to Hh signaling and associated with blastema fibroblasts possessing chondrogenic potential. However, this Hh responsiveness and chondrogenic capacity appears to be highly specific for regenerated tissues derived from blastemal cells. FCTCs in non-injured lizards; non-blastema FCTCs in lizards with amputated tails; FCTCs involved in the healing of other, non-tail injuries such as skin biopsies and limb amputations; tail stump FCTCs in clodronate-treated lizards: None of these populations express *sulf1* or undergo chondrogenesis following stimulation with SAG. Taken together, these results show that, within adult lizards, only blastema FCTCs respond to Hh stimulation with *sulf1* expression and chondrogenesis.

Using the parthenogenetic lizard *L. lugubris* as a platform for cell transplantation experiments, we took advantage of the differences in *sulf1* expression between tail blastema and limb FCTCs to specifically demonstrate the abilities of *sulf1*+, but not *sulf1*–, fibroblasts to undergo

Hh-responsive chondrogenesis in vivo. Furthermore, *sulf1⁻* limb FCTCs transplanted into tail blastemas did not form cartilage in response to Hh stimulation despite the presence of endogenous tail FCTC *sulf1* expression, suggesting that *sulf1* transcription is indicative of larger cell state changes in FCTCs affecting Hh responsiveness and chondrogenesis. Indeed, exogenous sulfatase produced by *sulf1⁺*-transplanted FCTCs failed to stimulate chondrogenesis in limb fibroblasts, suggesting that *sulf1* alone is not sufficient for cartilage formation in adult lizard fibroblasts, and that concurrent epigenetic reprogramming may be required to transition adult fibroblasts to Hh-responsive cell states.

Given the publication record showing *sulf1* alters local signaling environments by remodeling heparin sulfate proteoglycans and growth factor interactions[60–64], these results may indicate *sulf1* as a critical regulator of blastema formation that, through a positive feedback scenario, modulates its own Hh-regulated transcription and transforms the blastema signaling environment through varied effects on other critical signaling cascades. In addition, as displayed in pseudotime trajectory analysis, not all *sulf1⁺* fibroblasts activate chondrogenic programming in the regenerating tail. Some *sulf1*-expressing FCTCs, particularly those distal to Hh signaling sources, could be playing a role in regrowth and differentiation of other regenerating tissues via signal/pathway modulation. Future work will be aimed at determining the specific roles of *sulf1* in modulating Hh, Wnt, BMP, and FGF signaling and resultant effects on blastema cell derivation and behavior.

The divergent regenerative potentials of lizard tails and limbs within the same model organism also facilitate identification and testing of pro-regenerative tail-specific molecules and cells to enhance blastema formation in naturally non-regenerative limbs. Here, comparisons of phagocyte populations between lizard tails and limbs lead to the observations that only tail blastemas exhibit pro-regenerative septoclast populations, and that treatment with septoclast-secreted factors enhances *sulf1* expression and cartilage formation in amputated limbs. Thus, this study has led to the intriguing question: Why does the amputated lizard tail, but not limb, wound healing environment support septoclast differentiation, survival, and/or persistence? Previous reports in zebrafish identified vascularized hypertrophic regions of bone growth plate as sources of septoclast-like cell populations[91]. Interestingly, adult lizard tail vertebrae, but not limb bones, exhibit prominent growth plates associated with hypertrophic cartilaginous intervertebral pads[92]. Perhaps blastema septoclasts originate from these tail-specific structures. Future work will aim to identify the origins of blastema septoclasts and if other mechanisms account for phagocyte population differences between amputated lizard tails and limbs.

Prior investigations into blastema formation and immune cell regulation of wound environments have focused on cytokine stimulation of intracellular signaling cascades in resident FCTCs[9,20,93,94]. However, we have focused on functional changes to FCTCs following exposure to septoclasts and found that lizard blastema FCTCs are uniquely responsive to Hh signaling. Another question considers how septoclasts alter lizard FCTC Hh responsiveness. Canonical Hh signaling is highly conserved across the animal kingdom and targets activation of the Gli family of transcription factors[95]. Activated Gli transcription factors bind accessible Hh response elements (HHREs) within promoters/enhancers containing the recognition sequence GACCACCCA, resulting in expression of corresponding genes[96]. Many genes, including *sulf1* and *sox9*, contain HHREs within their promoters/enhancers[69,97], but epigenetic regulation via DNA methylation, histone binding, and other chromatin state changes can restrict accessibility to Gli binding[98]. Future work will investigate possible roles of epigenetic reprogramming in supporting septoclast-induced changes in FCTC Hh signaling sensitivities and responses.

Our studies demonstrated that lizard limbs, which do not naturally form blastemas, treated with septoclast-secreted factors can form cartilage in response to Hh stimulation. While previous studies have used other growth factors and treatments to produce cell proliferation and cartilage development within amputated lizard limbs[99], our results are distinguished by several important points. For example, treatment with FGF beads is reported to increase cartilage callus formation along periosteum of amputated limb long bones. However, we have previously shown that callus cartilage originates from periosteum of long bones[14], not from blastema fibroblasts.

Periosteum-derived calluses are also distinguished from blastema fibroblast-derived cartilage in that callus cartilage undergoes hypertrophy and endochondral ossification. Blastema FCTC-derived cartilage, as well as FCTC-derived cartilage formed in limb amputation experiments co-treated with septoclast-conditioned media and SAG, do not undergo hypertrophy or ossification. Taken together, these results suggest that lizard limbs do contain FCTC populations with the potential to form blastemas and cartilage, but limbs lack the necessary environmental signals required to enhance Hh-responsive FCTC chondrogenic potential. This study has highlighted septoclast-like phagocyte populations as a critical source of signals regulating FCTC blastema derivation. Single-cell subclustering and pseudotime trajectory analysis of immune and phagocytic cells over the course of regeneration may provide interesting insight to evolving transcriptional profiles and signaling roles of phagocytes in tail regrowth.

In summary, lizards represent an underused, but powerful model for studying the biology of blastema formation and regeneration. While models of mammalian blastema-based regeneration exist, such as rodent digit tip, the periosteal fibroblasts of adult mammalian blastemas exhibit very limited differentiation potentials and can only form bone[4–6]. Lizards are the only known amniotes, and closest relatives to mammals, that retain the regenerative abilities of amphibian salamanders, forming blastema rich with fibroblasts possessing multilineage potential, including cartilage, adipose, muscle tissues, etc. This study revealed that the unique evolutionary position occupied by lizards affects their blastema cell identity, which takes the form of transcriptionally distinct adult FCTCs rather than re-created embryonic populations. These cells demonstrate unique, embryonic-like responses to Hh signaling that functionally distinguish lizard blastema cells from other adult fibroblast populations. Thus, this study highlights interesting ways that a fully developed, adult amniote can build on nonspecific wound healing responses to create a blastema, inhibit scar formation and promote appendage regeneration.

## Methods

Additional materials and methods can be found in Supplementary Information. All experiments complied with relevant ethical regulations for animal testing and research.

### Lizard husbandry and species selection

Green anole lizards (*Anolis carolinensis*) were housed on a 12-h light/ 12-h dark schedule with 50 W spot basking heat lamps and UVB lamps during light hours. Anoles were maintained at 65% humidity, at 24–26 °C during light hours and 18.5–21 °C during dark hours in metal mesh cages. Cages were misted with water five times per week, and lizards were fed a diet of an excess of ½-inch crickets dusted in calcium supplement three times per week. Male and female anoles, ages 9–12 months old, were tested equally in all experiments.

Cell transplantation experiments were carried out using mourning geckos (*Lepidodactylus lugubris*). This all-female lizard species is parthenogenetic and reproduces asexually, yielding clonally identical offspring[74]. Cells and tissues can be transplanted between colony members in the absence of immunosuppressants and anti-rejection therapeutics[87], which have been shown to impact tail regeneration[24,75,76]. Mourning geckos were maintained at 24–26 °C

during light hours and 18.5–21 °C during dark hours in plastic cages, with 65% humidity, water misting three times per week, and were fed a diet of fruit meal replacement powder 3 times per week. All mourning geckos utilized in experiments were ages 9–12 months old. Husbandry and experimental use of lizards were conducted per guidelines of the Institutional Animal Care and Use Committees at the University of Pittsburgh (protocols 15114947, 16128889, and 18011476) and the University of Southern California (protocol 20992).

## Lizard amputations and staging

Lizard tails were anesthetized by exposing tails to a spray of ethyl chloride for 10 s and amputated with a sterile scalpel blade to begin regeneration. Tail samples were collected in Hank's balanced salt solution (HBSS) supplemented with 100 units/mL penicillin and 100 μg/mL streptomycin (HBSS with P/S). Samples were collected at 0, 1, 3, 7, 14, 15, 21, and 28 days after initial amputation or days post-amputation (DPA) for histology or cell dissociation.

Although lizard tail regeneration follows a reliable pattern, inherent variability in tail growth and regeneration rates between individual, field-collected lizards can lead to inconsistent timelines for regeneration[41]. To ensure consistency among sample groups, in addition to tracking sample DPA, tails were assessed for standardized morphological and phenotypical characteristics for each regeneration stage[42]. Inflammatory stage tails ranged from the closing of the wound epidermis to scab accumulation, blastema stage tails ranged from the loss of the amputation site scab to tail cone shape formation, and homeostatic regeneration stage began after tail cone formation through elongation and full regrowth. Original tail samples were isolated during initial amputations and were consistently 0 DPA. 1, 3, and 7 DPA tails were classified as the inflammatory stage, and 14 and 21 DPA tails were classified as blastema stage samples for single-cell RNA-sequencing (scRNAseq) analysis.

Phagocytic macrophages and osteoclasts populations peaked at 7 DPA (Supplementary Fig. 14) and thus, were utilized as the inflammatory stage time point throughout the manuscript. In total, 28 DPA samples were consistently utilized as regenerated homeostasis stage tails. Despite continued tail growth and elongation after 28 DPA[16], this time point represents the earliest reliable point at which terminally differentiated cartilage and regenerated muscle, blood vessels, peripheral nerves, spinal cord, and mature connective tissues can be identified in *A. carolinensis*[12].

## Histology and imaging

Lizard tail and limb samples were fixed overnight in 10% neutral-buffered formalin (NBF) and then decalcified for 1 week in 14–20% ethylenediaminetetraacetic acid (EDTA) solution in PBS, pH 7.2. Samples were then subjected to a sucrose gradient before snap freezing in Optimal Cutting Temperature Compound (OCT) with 2-methylbutane and dry ice. Cryosamples were sectioned at 16 μm via Leica CM1860 cryostat. All images of sagittal sections are presented in figures with dorsal tail toward the top, ventral toward the bottom, distal toward the right, and proximal toward the left, unless otherwise noted.

Processed samples were imaged with Keyence BZ-X810 Microscope with 2X, 10X, and 40X Nikon CFI Plan Apochromat Lambda D objective lenses. BZX DAPI, GFP, TRITC, and Cy5 filter cubes (Keyence) were utilized for corresponding fluorescence staining. Z-stacks and stitched images were processed using BZ-X800 Analyzer software (Keyence, v1.1.1.8), according to the manufacturer's instructions. Adobe Photoshop 2021 (v22.5.3.561) and Illustrator 2022 (v26.0.1) were used for preparing images and figures for publication.

## Drug treatments

For Hedgehog pathway signaling modulation, lizards were weighed and treated with cyclopamine (50 μg/g), or smoothened agonist (SAG, 40 μg/g) dosed per gram weight of the animal and administered via intraperitoneal (IP) injections every 24 h. Control animals were treated with phosphate-buffered saline (PBS) in place of drug treatments.

For phagocyte depletion, lizards were weighed and received IP injections of ʟ-α-phosphatidylcholine/cholesterol liposomes containing clodronate (0.125 mg/g animal weight) 72 and 48 h prior to amputation, dosed per gram weight of the animal. Control animals were treated with liposomes containing PBS instead of clodronate.

## Single-cell RNA-sequencing cell dissociation

Three tail samples per time point (0, 1, 3, 7, 14, 21, and 28 DPA) were cut to 3–5-mm pieces in length and each piece was cut into 1/8ths. Tail pieces were added to gentleMACS™ C tubes (Miltenyi Biotec, PN: 130-093-237) containing 2.5 mL Dulbecco's modified Eagle media (DMEM), 100 μL proprietary Enzyme D, 50 μL Enzyme R and 12.5 μL Enzyme A from Multi Tissue Dissociation Kit 1 (Miltenyi Biotec, PN: 130-110-201). Tubes were inverted and placed onto gentleMACS™ OctoDissociator with sleeve (Miltenyi Biotec, PN: 130-096-427) and a fibroblast dissociation protocol was run for 1 h. Enzymatic activity was inactivated with DMEM supplemented with 10% fetal bovine serum (FBS). Cells were gently resuspended via pipetting and run through MACS® 70 μm SmartStrainer (Miltenyi Biotec, PN: 130-098-462), followed by filtration via Scienceware FlowMi® 40 μM Cell Strainers for p1000 pipettes (Sigma Aldrich, PN: H13680-0040). Cells were pelleted and washed with DMEM with 10% FBS. Cells were resuspended in HBSS with 0.04% bovine serum albumin (BSA) prior to library preparation.

## Single-cell RNA-sequencing library preparation and next-generation sequencing

Isolated cells were counted on a hemocytometer and prepared using the 10x Genomics Chromium Single-Cell Gene Expression kit (V2, PN: 120267) according to the manufacturer's recommendations. Cells were encapsulated into droplets via gel bead-in emulsion (GEM) method for barcoding using the 10x Genomics Chromium controller (PN: 1000202). GEMs were incubated for cDNA synthesis, followed by amplification and library construction, according to the manufacturer's instructions. Pooled libraries were sequenced on the MiSeq platform (Illumina). Single-cell libraries were run with MiSeq Reagent Kit v3, 150 cycles (Illumina, PN: MS-102-3001) and paired-end sequenced (2 × 150 base reads) at the USC Molecular Genomics Core. 10,000 cells per sample were sequenced to an estimated 50,000 reads per cell. Per-sample BCL file outputs were converted to FASTQ files using the Cell Ranger mkfastq pipeline with the single-cell RNA-sequencing analysis software Cell Ranger[43] (v3.1.2, 10x Genomics).

## Single-cell RNA-sequencing data analysis

Green anole (*A. carolinensis*) genome FASTA and GTF files (Ensembl, v2.105) were used to generate a reference genome for indexing by Cell Ranger mkref function (v3.1.2, 10x Genomics). Sample read FASTQ files were aligned to the generated reference genome and counted with Cell Ranger count pipeline to generate gene-barcode expression matrices. For all scRNAseq sample datasets, Seurat[44] (v4.1.1) was used to perform standard quality control/pre-processing, including normalization of libraries and removal of cells with greater than 5% mitochondrial gene expression. To integrate the samples, RunHarmony() from R package Harmony[45] (v1.0) was used to iteratively correct PCA embeddings.

The filtered, normalized, and integrated scRNAseq data from original tail (0 DPA samples), inflammatory (1, 3, and 7 DPA samples), blastema (14 and 21 DPA samples), and regeneration stage (28 DPA) tails were used for unsupervised clustering using FindClusters (resolution = 0.3), then identification and terming metaclusters using Seurat. To investigate regeneration-related metaclusters, we used multiple methodologies, including dimensional reduction and trajectories. The first 20 dimensions from Harmony embedding were used for UMAP plots (min.dist = 0.6). R package Monocle2[49–51] was used to build single-cell trajectories with which pseudotime was introduced

for the analysis of fibroblastic connective tissue cell and chondrocyte clusters.

## In situ hybridization

Chromogenic in situ hybridization (ISH) was accomplished using the RNAscope™ 2.5 HD Detection Kit RED (Advanced Cell Diagnostics, PN: 322350) and custom proprietary ISH probes (Advanced Cell Diagnostics, Supplementary Table 2). Samples were baked for 1–2 h at 60 °C, rinsed in PBS, and post-fixed with 10% NBF at 4 °C for 15 min. Slides were then dehydrated in an ethanol gradient and treated with kit hydrogen peroxide solution for 10 min. Slides were rinsed in distilled water and samples were outlined with ImmEdge® hydrophobic pen (Vector Laboratories, PN: H-4000).

Slides were then incubated in RNAscope™ protease III solution (Advanced Cell Diagnostics, PN: 322337) at 40 °C for 30 min and hybridized with custom ISH probes for 2 h at 40 °C. ISH signal was then amplified over six amplification steps with kit amplification reagents, according to the manufacturer's instructions. The signal was detected with 1:50 kit Fast RED-A: Fast RED-B solution for 10 min at room temperature. Sections were then counterstained with 50% Gill's Hematoxylin I (StatLab, PN: HXGHE1LT) for 2.5 min and 0.02% ammonium hydroxide for 1 min at room temperature. Slides were washed with distilled water and then dipped in xylene immediately before mounting in EcoMount mounting media (Biocare, PN: EM897L), and a glass coverslip was placed. Slides were cured overnight at room temperature and stored at room temperature before imaging.

Fluorescent ISH (FISH) was accomplished using the RNAscope™ Multiplex Fluorescent Reagent Kit v2 (Advanced Cell Diagnostics, PN: 323100) and custom proprietary ISH probes (Advanced Cell Diagnostics, Supplementary Table 2). Samples were baked for 1–2 h at 60 °C, rinsed in PBS, and post-fixed with 10% NBF at 4 °C for 15 min. Slides were then dehydrated in an ethanol gradient and treated with kit hydrogen peroxide solution for 10 min. Slides were rinsed in distilled water, and samples were outlined with ImmEdge® hydrophobic barrier pen (Vector Laboratories, PN: H-4000). Slides were then incubated in RNAscope™ protease III solution (Advanced Cell Diagnostics, PN: 322337) at 40 °C for 30 min and hybridized with up to three custom ISH probes for 2 h at 40 °C. ISH signal was then amplified over three amplification steps with kit amplification reagents, according to the manufacturer's instructions.

Kit HRP-C1, C2 or C3 solution, corresponding to channel (C) number 1, 2, or 3 of each ISH probe, was added to slides and incubated at 40 °C for 15 min. Signal was detected with 1:1500 Opal™ 520, 570, or 690 Reagent Packs for GFP, TRITC or Cy5 fluorescence, respectively (Akoya Biosciences, PNs: FP1487001KT, FP1488001KT, FP1497001KT), diluted in RNAscope™ kit TSA dilution buffer, incubated for 30 min at 40 °C. HRP signal was blocked with kit HRP blocker for 15 min at 40 °C. For multiplexed samples, HRP-C# solution, fluorophore solution and HRP blocker steps were repeated with respective probe channel numbers and fluorophores, according to the manufacturer's instructions. Sections were then counterstained with kit DAPI for 30 s at room temperature and immediately mounted in ProLong™ Gold Antifade Mountant (ThermoFisher Scientific, PN: P36930), then a glass coverslip was placed. Slides were cured overnight at room temperature in the dark, and were then sealed with clear nail polish, and stored protected from light at 4 °C before imaging. Positive FISH areas were quantified using Fiji (Image J, NIH v2.9.0), described in detail in Supplementary Methods.

## Immunofluorescence staining

Lizard tissue samples were analyzed by Col2 immunofluorescence (IF) staining (primary antibody: Abcam ab34712, dilution 1:1000) and by Ctsk IF (primary antibody: Abcam ab19027, dilution 1:200) as previously described[14]. Detailed IF protocols and quantification methodology are available in Supplementary Methods.

## Lizard fibroblast isolation

Lizard (L. lugubris) tail and limb tissues were washed three times in 10% povidone-iodine solution and washed once in HBSS supplemented with 100 units/mL penicillin, 100 µg/mL streptomycin, and 250 ng/mL fungizone antimycotic. Washed tissues were incubated in 0.1% EDTA in HBSS for 45 min at room temperature with agitation, and scales/epidermis were peeled from each tissue piece with forceps and discarded. Prepared tissues were then washed extensively in HBSS, minced, and digested in 1 mg/mL trypsin and 1 mg/mL collagenase II for 1 h at 37 °C. Immune, muscle-related, and endothelial cells were depleted by passing cell suspensions through the following MACS® (Miltenyi Biotec) magnetic beads: CD144 (VE-Cadherin) MicroBeads (PN: 130-097-857), Anti-Integrin α−7 MicroBeads (PN: 130-104-26), CD45 MicroBeads (PN: 130-052-301), CD326 (EpCAM) MicroBeads (PN: 130-105-958), according to the manufacturer's instructions. Cell/bead suspensions were loaded onto LD columns (Miltenyi Biotec, PN: 130-042-901) and placed on MidiMACS™ Separator magnets (Miltenyi Biotec, PN: 130-042-301). Enriched fibroblast suspensions were collected in lizard cell culture medium (Dulbecco's Modified Eagle Media (DMEM)/Ham's F12, 2 mM Glutamax, 0.1 µM dexamethasone, 40 µg/mL proline, 50 µg/mL ascorbate, and 10 µg/mL ITS+ supplement).

## Lizard cell labeling and transplantation

DiI and DiO labeling of fibroblast isolation pools were performed using CellTracker™ dyes (ThermoFisher Scientific, PNs: C7001, V22886) according to the manufacturer's instructions. Briefly, cell suspensions were incubated with 1 µM CM-DiI/DiO labeling solutions for 5 min at 37 °C followed by an additional 15-min incubation at 4 °C. Cells were then washed with PBS and resuspended at a density of 5000 cells/µL. Labeled cell suspensions (2.5 million cells/animal) were then injected into tail stumps or blastemas using an insulin syringe and a microinjector system (Sutter Instrument).

## Lizard macrophage and septoclast culture preparation and media conditioning

Bone marrow was isolated via extrusion by crushing femur bones with a mortar and pestle. Bone marrow cells were passed through a 70-µm filter and treated with eBiosciences™ 1X red blood cell lysis buffer (ThermoFisher Scientific, PN: 00-4333-57). To isolate blood vessel cells, caudal arteries and veins were dissected from original lizard tails and digested in vessel dissociation solution (Leibovitz's L15 medium (ThermoFisher Scientific, PN: 11415064) containing 30 units/mL papain, 0.5 mg/mL BSA, 0.24 mg/mL cysteine, 40 µg/mL DNase I type IV, and 1.0 mg/mL trypsin inhibitor) for 90 min at room temperature. Digested vessels were homogenized by repeatedly passing solutions gently through p1000 pipette tips. Digestion was halted with ovomucoid inhibitor (1.0 mg trypsin inhibitor, 0.5 mg/mL BSA, and 40 µg/mL DNase I type IV in Leibovitz's L15 medium). Erythrocytes were depleted using the magnetic bead-based MACSxpress® Erythrocyte Depletion Kit (Miltenyi Biotec, PN:130-098-196) according to the manufacturer's instructions. Endothelial cells were depleted using MACS® CD144 (VE-Cadherin) MicroBeads (Miltenyi Biotec, PN: 130-097-857) magnetic beads and cell separation columns according to the manufacturer's instructions, as described above in fibroblast isolations.

For macrophage and septoclast differentiation, bone marrow cells and tail vessel cells, respectively, were cultured for 1 week in phagocyte selection medium (DMEM, 10% fetal bovine serum, 10% L929 supernatant, 0.1% beta-mercaptoethanol, 100 units/mL penicillin, 100 µg/mL streptomycin, 10 mM non-essential amino acids, and 10 mM HEPES) at 30 °C. Macrophage and septoclast cultures were then used to generate conditioned media (CM); cultures were washed with PBS, and selection medium was replaced with serum-free medium (DMEM, 10 µg/mL ITS+ supplement) for 24 h. Macrophage- and septoclast-CM was collected and concentrated via 3 kD MWCO spin concentrators (Millipore Sigma, PN: UFC5003), and protein content was determined

using BCA protein assays (ThermoFisher Scientific, PN: 23227), according to the manufacturer's instructions. Macrophage and septoclast cell pool purity validation is described in Supplementary Methods.

## Alginate bead preparation and implantation

In all, 1 mg/mL concentrated macrophage- and/or septoclast-CM was added to alginate solution (1.5% w/v alginate, 25 mM HEPES, 118 mM NaCl, 5.6 mM KCl, 2.5 mM MgCl$_2$). Alginate droplets were injected directly into crosslinking solution (100 mM CaCl$_2$, 10 mM HEPES) and cured for 15 min with continuous stirring. Cured beads were extensively rinsed with wash buffer (0.2% CaCl$_2$ in 0.9% saline solution). Beads were immediately inserted into amputated lizard limb and tail stumps.

## Statistical analyses

Statistical analyses were performed using GraphPad Prism 9. Statistical tests utilized in each figure are listed in respective figure legends with corresponding $P$ values or adjusted $P$ values, when applicable. In figures, $P$ values or adjusted $P$ values are represented as follows unless otherwise noted: *$P < 0.05$; **$P < 0.01$; ***$P < 0.001$; ****$P < 0.0001$. All tests were performed with 95% confidence intervals ($\alpha = 0.05$) with $P$ value or adjusted $P$ value < 0.05 deemed to be statistically significant. All values and graphs/error bars are shown as mean +/− standard deviation (SD). All experiments performed with *A. carolinensis* were completed with equal numbers of male and female lizards. No significant differences were observed as a result of sex for all reported results. All statistical tests including test statistic, degrees of freedom, and $P$ value/adjusted $P$ value are summarized in Supplementary Data 1. Source data for all quantification and statistical analyses is available in Supplementary Information.

## Data availability

The raw sequencing data and analysis for the lizard tail regeneration scRNAseq dataset have been deposited in the GEO database under accession code GSE234876. The source data for all quantification and statistical analyses are provided with this paper in the Supplementary Information/Source Data file. All other data that support the findings of this study are available from the corresponding author upon reasonable request.

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

## Acknowledgements

We would like to acknowledge funding from NIH R01GM115444 (TPL), and support from the Molecular Genomics Core at Norris Comprehensive Cancer Center, and their funding support, NCI P30CA014089. Special thanks to Dr. Kate Millette for assistance with statistical analysis, and to Kathleen Vonk and Nash Arbuckle for manuscript editing and continued support.

## Author contributions

A.C.V. contributed to conceptualization, methodology, software, validation, formal analysis, investigation, resources, data curation, writing, including original draft preparation, reviewing, and editing, visualization, supervision, and project administration. X.Z. contributed to methodology, software, validation, formal analysis, investigation, data curation, and visualization. Z.P. contributed to validation, investigation, resources, data curation, and reviewing and editing writing. M.L.H. contributed to validation, investigation, resources, data curation, supervision, and project administration. C.G.O., G.A.L., and S.C.H.-K. contributed to data curation. A.W.C.K. contributed to data curation and reviewing and editing writing. S.B.S. contributed to validation and data curation. D.J.G. contributed to resources and reviewing and editing writing. T.P.L. contributed to conceptualization, methodology, software, validation, formal analysis, investigation, resources, data curation, writing including original draft preparation, reviewing, and editing, visualization, supervision, project administration, and funding acquisition.

## Competing interests

The authors declare no competing interests.
