## [Peer Review File · Nature Communications]

REVIEWER COMMENTS

Reviewer #1 (Remarks to the Author):

Comments on the MS "Single cell analysis....blastema fibroblasts" by Vonk et al.

General comments

The present study utilizes lizards as useful models for analyzing the process of regeneration in an amniote. Specifically, the principal goal of the MS is to find new infos on the process of cartilage differentiation in the regenerating tail. The study is VERY INNOVATIVE and utilizes state-of-art methods to progress in this direction. The results indicate that, after Hh (shh) signaling within the regenerating tail blastema numerous fibroblasts begin to express chondrogenic genes such as spp1, sulf1, sox9 etc. Fibroblasts from the limb do not express some genes that are instead expressed in fibroblasts of the tail, likely under Hh signaling stimulation. The present study appears as a further continuation of previous researches from the same laboratory group, especially on the effect of shh (Hh) on the induction of cartilage in the regenerating tail of lizard. The study indeed brings further clues on how lizards use some key genes for forming a new axial skeleton in the regenerating tail. UNFORTUNATELY the present MS NEGLECTS (?) many studies on lizard regeneration, indicating poor knowledge or/and consideration of the vast past and present literature of the cell biology and immunology in lizard regeneration. New scientific discoveries are ALSO built on the comparison with research carried out from previous/present researchers, and the scientific progress in any field is based primarily on the confrontation on what is already known in order to plan new experiments through the use of modern technologies (see specific comments). This is the way SCIENCE progressed and SHOULD STILL PROGRESS: at least intellectual progress through literature consultation. I invite the present authors to CONSIDER this suggestion that should be obvious for any modern researcher and can only IMPROVE a MS that is already VERY VALUABLE. My comments and remarks are mainly indicated IN CAPITALS below and can be localized throughout the MS by a "computer sentence search" after: "the sentence:.....XYZ". Finally, I compliment with the AUTHORS for progressing with this important research using the lizard model for understanding amniote regeneration.

Specific comments

Abstract

- PLEASE briefly explain (here or in the introduction) the term "pseudotime trajectory analysis"
- please SPELL OUT spp1 and sulf1
-and indicate POSSIBLE/POTENTIAL actionable targets.....

Introduction

-the sentence:.....non-mammalian vertebrates MAY/MIGHT provide clues.....

-the sentence:.....mouse tip regeneration...: RECENT STUDIES HAVE INDICATED THAT THIS IS A “regengrow” NOT A REAL REGENERATION (see Alibardi L (2021) Regeneration in anamniotes was replaced by regengrow ... biological cycles. Dev Dynam, Special Issue 251, 1404-1413.

-the sentence:.....lack true tendons 1-7. NUMEROUS RECENT AND PAST EXTENSIVE STUDIES ON LIZARD REGENERATION ARE NOT CITED here LEAVING THE READER UNINFORMED ON WHAT IS PRESENTLY KNOWN ON THE TOPIC also in other lizards, from anatomy, histology, cell biology to immunology. PLEASE at least CONSIDER ALSO OTHER few STUDIES, FOR EXAMPLE :

- Fisher, R. E., Geiger, L. A., et al. (2012). A histology of the original and regenerating tail

in the green anole, *Anolis carolinensis*. The Anatomical Record, 295, 1609–1619.

-Alibardi L 2014 Alibardi, L. (2014). Histochemical, biochemical and cell biological aspects of ...tissue regeneration. Progress in Histochemistry and Cytochemistry, 48, 143–244.

-Jacyniak, K et al. (2017). Tail regeneration and other phenomena of wound healing ... lizards. Journal of Experimental Biology, 220, 2858–2869.

-the sentence:.....Macrophages act in the.....: THE SENTENCES SHOULD BE UP-DATED. IN FACT, DETAILED STUDIES ON M1, M2 macrophages and TREGs –uncited here- IN LIZARD BLASTEMAS ARE REPORTED IN:

-Alibardi L. 2016. Immunolocalization of 5BrdU long retainingafter limb amputation. Tiss Cell 48: 197-207.

-Alibardi L. 2020. Autoradiography and immunolabeling suggests ...support tail regeneration. Ann Anat 231, 151549.

-Alibardi L. 2022. Immunoreactivity for Dab2 and Foxp3 ... of lizard. Acta Zool 103, 389-401.

Results

AS A GENERAL REMARK THE AUTHORS SHOULD INITIALLY -HERE OR IN MAT&METHODS- MAKE A LIST OF THE MAIN GENES SEARCHED FOR (SPP1, SULF1, CDH11, PLTP, SALL1 ETC.) EXPLAINING THAT THEY WERE IN PARTICULAR INTERESTED IN THOSE RELATED TO CARTILAGE DIFFERENTIATION. THIS WOULD HELP THE READER TO GRASP THE MAIN EXPERIMENTAL TARGETS OF THE PRESENT AUTHORS.

-on page 4, the sentence:.....inflammatory stage (DAY ?), blastema stage (DAY ?), AND regenerating homeostasis (DAY ?). PLEASE INDICATE DAYS/range of DAYS IN EACH STAGE.

-pag 5, the sentence:....., cd8a+ and cd8b+.....:PLEASE spell out CD8a,B (ARE THEY CYTOTOXIC T-CELLS?)

-pag. 5, the sentence:.....through gene expression analysis: PLEASE BRIEFLY EXPLAIN THIS TYPE OF IDENTIFICATION

-Pseudotime trajectory analysis: WHAT DOES IT REVEAL ? THIS IS IMPORTANT TO UNDERSTAND THE FOLLOWING DESCRIPTIONS.

-spp1: please SPELL OUT THE NAME OF THIS GENE, AS WELL AS SULF1

-CDH11 is what?

-Lizard tail blastemas are...FCTCs”: WHICH key genes for “fibroblast activation” were here evaluated?

- pag. 6, the sentence:.....During blastema formation AT 14 DPA....

- pag. 7, the sentence:.....During tail elongation AT 21 DPA....

-Lizard blastema FCTCs...signaling, pag 7

-the sentence:.....Lizard tail and limbs were collected 28 DPA from ANIMALS treated with...

-the sentence:.....septoclast population: CAN THE AUTHORS BRIEFLY MENTION WHAT A SEPTOCLAST IS for the benefit of readers? (Osteoclasts derived from pericytes?)

-In summary, the results described above.....: ARE THE PRESENT AUTHORS AWARE THAT FGFs (1 OR 2) ADMINISTRATION TO THE LIMB STUMP IN LIZARDS PRODUCES RELATIVELY LARGE AMOUNT OF CARTILAGE (femur, tibia and fibula)? HAVE THEY TESTED/WILL THEY BE ABLE TO TEST FGFs IN LIMBS FOR-perhaps- STIMULATING EXPRESSION OF THEIR tail CHONDROCYTE MARKERS also in THE LIMB treated with FGF? SEE: Alibardi L. 2017. FGFs treatments on amputated limbs stimulate the regeneration of long bones, opening new avenues for limb regeneration in amniotes. A morphological study. J Morphol Phys Kinesiol 2, 25.

Discussion

IN GENERAL THIS PART OF THE MS CAN BE sensibly SHORTENED ELIMINATING REPETITIONS AND KNOWN PROCESSES. THE FIRS 4-5 SENTENCES REPEAT THE INTRODUCTION. ...AND (not mandatory however) THE ORGANIZATION OF THE DISCUSSION INTO SUB-SECTIONS WOULD MADE THE TEXT MORE READABLE as the text is exscessively LONG and unstructured (eg. one subheading on the cartilage influence ...; and another on theImmunological implications).

-the sentence:.....all species tested regrow near perfect copies..... THIS STATEMENT IS NOT REALLY TRUE AS SEVERAL URODELES, ESPECIALLY TERRESTRIAL-ADAPTED (SALAMANDERS) DO NOT REGENERATE (forming SCARring outgrowths) OR only SHORTER AND HETEROMORPHIC LIMBS (Scadding, S.R. (1977). Phylogenetic distribution of limb regeneration capacity in adult Amphibians. Journal of Experimental Zoology 202, 57-68.

Scadding, SR. (1981). Limb regeneration in adult amphibians. *Canadian Journal of Zoology* 59, 34-46.

-the sentence:.....the first MOLECULAR interrogation into the heterogeneity of.....: HOWEVER

IT IS KNOWN, BASED ON MORPHOLOGICAL AND LABELING STUDIES (ultrastructure, autoradiography and immunolabeling), FROM LONG TIME that the lizard blastema AND THE ORIGIN OF ITS CELLS IS HETEROGENOUS, CELLS DERIVED FROM different tissues and others from dedifferentiation, as it is reported in the literature.

-the sentence:.....as salamander cells although this may be attributed to cell sampling and accessibility differences???: PREVIOUS STUDIES INDICATED THAT AFTER AUTOTOMY ALONG THE AUTOTOMOUS PLANES MAINLY FIBROBLASTS OF THESE SEPTA REMAIN AND THAT THE BLASTEMA IS ESSENTIALLY DERIVED FROM EITHER THE DE-DIFFERENTIATION FORM THESE CELLS BUT MOST LIKELY FROM THEIR STEM-LIKE NATURE (refs in the literature, see Simpson SB 1968 *Amer Zool* etc.).

-the sentence:....but these were limited to FCTC activation of several genes.... THIS IS INDEED AN IMPORTANT NEW FINDING OF THE PRESENT STUDY SINCE LIZARD REGENERATION LIKELY HAS EVOLVED A MECHANISM OF HETEROMORPHIC REGENERATION BY THE Re-USE OF A LIMITED NUMBER OF GENES THAT WERE INITIALLY USED DURING TAIL EMBRYOGENESIS (see Alibardi, L. (2020). Appendage regeneration in anamniotes utilizes genes active ... studying lizard tail regeneration. *Journal of Morphology* 281, 1358-1381.

-YES, TEM studies showed that lizard blastema cells NEVER become ultrastructurally as simple as the mesenchymal cells of newts or the axolotl.

-THE PRESENT AUTHORS MAY ALSO CONSIDER IN THE FUTURE TO EVALUATE FOR GENES INVOLVED IN SOMITOGENESIS, expressed in lizard embryos BUT NOT In the regenerating tail (see Eckalbar et al. (2012). Somitogenesis in the anole lizard and alligator reveals evolutionary convergence and divergence in the amniote segmentation clock. *Developmental Biology* 363, 308-319.

-the sentence:.....for example, comparing tail blastema versus amputated limb wound healing: BUT THIS STUDY HAS BEEN DONE, AND NUMEROUS GENES WERE DETECTED AS DIFFERENT FROM TAIL VS LIMB, INCLUDING CONNECTIVE tissue GENES: SEE

- Vitulo N, et al. 2017. Transcriptome analysis of the regenerating tail versus the scarring limb ... organ regeneration in amniotes. *Dev Dyn* 246: 116-134.

- Vitulo N, Dalla Valle L, Skobo T, Valle G, Alibardi L. 2017. Down-regulation of lizard immuno-genes in the regenerating tail and myo-genes ...immuno-privileged organ. *Protoplasma*. 254: 2127-2141.

-Patel, et al. (2022). De novo transcriptome sequencing and analysis of differential gene expression ...in *Hemidactylus flaviviridis*. *Journal of Developmental Biology* 10, 24.

-THE DISCUSSION ON THE Hh INFLUENCE IS one of the the MAIN FOCUS OF THE MS BUT ALSO ANOTHER STUDY ON SHH SECRETION FROM LIZARD ependyma can be found consulting the EXISTING LITERATURE ON THE TOPIC (download the MS and provided to the present AUTHORS):

Alibardi L 2017. Immunolocalization of Shh and Patched....epidermal differentiation. *BAOJ Dermatol* 1, 004.

-In their histochemical study, the activation of sox9 for cartilage regeneration in lizard is reported in Delorme et al, 2012. Scar-free wound healing and regeneration... *Eublepharis macularius*. *Anat. Rec.* 295, 1575-1595.

Materials and Methods

As a GENERAL REMARK, the numerous AKRONYSMS UTILIZED (SAG, NBF,

-pag. 21, the sentence:....therapeutics39, which have been shown to impact tail regeneration (ANY CITATION HERE ? or PERSONAL OBSERVATIONS?)

-Lizard tail amputations:.....Lizard tails were anesthetized BY EXPOSING THE ANIMALS TO A SPRAY OF ETHYL CHLORIDE FOR 10 SECONDS and amputated.....

-the sentence:....and 28 days after initial amputation OR DPA (DAYS POST-AMPUTATION) for histology.....

-DATA AVAILABILITY

the sentence:....dataset are available at GSEXXXXXX.....: PLEASE WHAT DOES IT MEAN ?

Reviewer #2 (Remarks to the Author):

Key results

The authors utilized scRNA-seq with pseudotime trajectory analyses on regenerating *A. carolinensis* tails to explore the source of lizard tail blastemal cells and chondrogenesis. *Spp1* was identified as a marker of fibroblastic connective tissue cell (FCTC) activation, and a *sulf1+* subpopulation of *spp1+* FCTCs were

identified as responsive to Hedgehog signaling and subsequently capable of chondrogenesis. Macrophage depletion via clodronate treatment, which the authors have previously shown inhibits blastema development, decreases *sulf1* and *spp1* expression. While macrophage-conditioned media was able to induce FCTC activation marked by *spp1* expression, only tails also supplied with septoclast-conditioned media induced FCTC activation marked by *sulf1* expression, which they suggest is blastema-specific. Taken together, the authors propose that a hierarchical FCTC activation via septoclast and macrophage secreted molecules promotes FCTC responsiveness to Hedgehog signaling, allowing for blastema formation and chondrogenesis in lizard tails. Lizard limbs contain *sulf1*- FCTCs and are therefore unresponsive to chondrogenic promotion via Hedgehog signaling, perhaps explaining the inability of lizards to regenerate their limbs while their tails retain imperfect regenerative capacity. Importantly, lizard limb FCTCs appear to be capable of *sulf1* expression when transplanted early into the regenerating tail.

Validity

While as a whole the study's findings are intriguing and exciting, there are several concerns which should be addressed (see suggested improvements) before the manuscript is ready for publication:

☐ It seems strange to include 1DPA and 3DPA as timepoints in an “inflammatory stage” for scRNA-seq, but then to effectively ignore these timepoints in any later experiments.

☐ Fig. 1b raises the question of why 28DPA is considered a regenerated homeostasis, and light-microscope images of tissues at each of these timepoints would serve to validate this. My concern is due to the relatively low difference in cellular makeup of the “regenerated homeostasis” and “blastema stage” and the relatively high difference between the “regenerated homeostasis” and “original tail” stages. Though it is known that lizard tail regeneration is imperfect, it seems that 28DPA may not thoroughly represent a full regenerate, such as with the high number of actively cycling cells.

☐ It is not entirely clear from the pseudotime trajectory that *sulf1* cells reliably become *soxa9+* and *col2a1+*. Could *sulf1+* cells in the other branch contribute to the regeneration process, and if so, how?

☐ With regards to Fig. 2 and related findings, *cdh11+* FCTCs are referred to being in a “resting” state, but are highly active in the “blastema” and “regeneration” stages.

Significance

Methodological and validation concerns aside, the findings of this manuscript are interesting. Given previous work on the roles of macrophages and fibroblasts in salamander regeneration (Godwin et al, 2013; Gerber et al, 2018), this manuscript serves to supplement this work in a new model with intriguing implications.

Perhaps the biggest addition which would strengthen this paper's significance would be the inclusion of comparisons between pre-amputation and post-regeneration tail tissue. Because lizard tail regeneration is of considerably lower fidelity than that of something like salamander limb regeneration, as stated several times with regards to both skeletal and nervous elements, a better understanding of what the

exact cellular and molecular differences in makeups are would greatly expand the possible targets of future study beyond what the authors have proposed here.

Two points, however, might need to be qualified in terms of significance (both from the discussion):

☐ It may be a stretch to claim that the study identified “a majority of fibroblast activation markers”. There is no way to be sure of this, nor was a novel molecule discovered, such as an equivalent to prrx1 in salamanders.

☐ It is relatedly a stretch to suggest sulf1 is “a master regulator” for blastema formation.

☐ Based on what is currently shown, it may not be appropriate to say cdh11 is repurposed to fill the role of embryonic cdh2, as the experiments did not explore this and the lizard’s regeneration is imperfect – if it is true, it’s not a perfect recapitulation.

Data and methodology

The conducted experiments build logically on each of their findings, covering a swath of aspects of the regenerative process as the authors intended. However, several of the methods utilized in this study muddy the conclusions’ validity, which should be justified in a later version of the manuscript or addressed with the suggested improvements:

☐ For scRNA-seq, the combination of 1DPA, 3DPA, and 7DPA cells into one “inflammatory stage” sample is concerning without proper context. 1DPA and 7DPA may represent vastly different cell states, with 1DPA more of a wound healing stage. Subsequent ISH and other analyses utilize only 7DPA samples of this stage, where certain results would be strengthened by the inclusion of earlier timepoints. Otherwise, justification for the inclusion of just 7DPA ISH should be provided.

☐ The combination of 14DPA and 21DPA samples into one “blastema stage” is similarly concerning, especially given that later in the manuscript (Fig. 2) 21DPA is instead referred to as “regeneration stage”, whereas in the methods and scRNA-seq the 28DPA timepoint is stated as a homeostatic “regeneration stage”. In other models of regeneration, these again represent vastly different cell states. Though both of these DPAs are utilized in later analyses, for the purposes of scRNA-seq it’s unclear why they are combined.

☐ In Fig. 4 and related findings – Is it possible that 14DPA could be too late of a blastema stage to activate limb fibroblasts; is 14DPA considered early blastema, or mid in this context?

☐ Overall, the resolution of the images was low, making it difficult to interpret much of the ISH clearly - although this is more reflective of the submission portal than the images themselves.

Analytical approach

In general, the analytical approach is sound, but there are a few gaps:

☐ There is a lack of description of their FISH protocol and related imaging modalities and processing. More description of their methods for quantification in FIJI would make this more replicable.

☒ Include the meaning of the hashtag in Fig. 5.

☒ In Figs. 4, 5, and 7, it's uncertain whether two-way ANOVA or a nonparametric test would be more appropriate. It doesn't appear, visually, that there is equal variance between groups, nor can we assume a normal data distribution.

Suggested improvements

Per the above comments, there are a number of improvements which could be made before publication.

☒ As *pltp* and *sall1* are indicated important blastema markers with regards to single-cell sequencing, a pseudotime trajectory and clustering analysis of these two genes included in Fig. 1 would be useful reference.

☒ Brief exploration with similar clustering methods of if or how macrophages and other immune cells such as septoclasts evolve through regeneration might provide interesting insight.

☒ Exploring differences in the inflammatory stage and between this and the blastema stage would be useful, especially with regards to the claim of macrophage infiltration before FCTC activation. Showing for certain that macrophages peak at 7DPA, for example compared to 1 and/or 3DPA, would strengthen this claim.

☒ Including data for 28DPA samples would support several experiments, such as in Fig. 2 and 5. The final model in Fig. 10 is of 0, 7, 14, and 28DPA, but 28DPA is largely absent. Otherwise, justify why 28DPA does not need to be presented. Is it very similar to 0 DPA or is it just too similar to 21 DPA to include?

☒ In Fig. 6, it would support the model of a *sulf1+* *spp1* sub-population of FCTCs to include at least a 7DPA timepoint for co-localization. Or can this be deduced by the scRNAseq data?

☒ Panel numbering and in-text references of Figs 8. and 9 could be neater.

☒ With the experiments associated with Fig. 7, it would also make sense to transplant fibroblasts into tails at early regeneration timepoints. It could very well be that 14DPA, when the blastema is already formed, could be far too late to rescue fibroblast activation. Otherwise, adding this as a caveat to the study is sufficient.

☒ In several instances, seemingly important validation and findings are relegated to supplementary figures:

- o All figure references in the results section "Lizard blastema FCTCs undergo chondrogenesis in response to Hedgehog signaling" are to supplementary figures. If this is an important crux to their argument, they should be made main figures.

- o All references to limb regeneration, though importance is placed on comparing and contrasting with tail regeneration, are also placed in supplementary materials, but appear to be important for overall significance.

- o

Clarity and context

As a whole there is sufficient context to justify the experiments as well as previous findings by this and other groups to support methods. The following points mostly concern the logical flow of information, which would make the text more accessible to a reader:

☒ In Fig. 2 and related results, *mdk*, *tenascin*, etc., though in supplementary materials, are discussed before indicating why the genes in the figure are shown. It would be useful to provide context with regards to “condensing chondrogenic mesenchyme” as a justification for the genes they chose to show in the main figure.

☒ For those not acquainted with anole regeneration, more context would be useful on the use of the second lizard species to support experimental validity – given that this research heavily involves immune system recruitment, it seems important given the point about their not needing immunosuppressants.

☒ Some areas of the discussion feel better suited for the introduction, such as the composition of the regenerated tail when compared to salamanders.

☒ The introduction mentions zebrafish regeneration in reference to macrophages, but otherwise does not discuss their utility in consideration of other models.

☒ The discussion of the Gerber/Tanaka axolotl scRNA dataset may be more effective when discussing the results shown in Fig. 1.

☒ The information given about *sulf1* in the discussion may be much more effective in the intro or relevant results sections, as otherwise there is little reason given for their focus on this gene.

References

The manuscript fairly cites references from a broad range of fields and models to support its claims. Previous work by the research group demonstrates expertise in this niche, but does not dominate the references. A few small points:

☒ The claim that lizard blastema cells do not become more embryonic-like, as salamander blastema cells do, would be best supported by a reference to previously published embryonic lizard datasets or studies on lizard de-differentiation potential, such as Eckalbar et al., 2011. Also, has dedifferentiation of tail cells been shown in axolotls? I can only think of examples in limb regeneration.

☒ It may be useful to reference deer antler regeneration, an example of mammalian organ regeneration that is not established to be blastema-dependent (Li, Zhao, Liu & McMahon, 2014).

☒ In the sentence “Fibroblastic connective tissue cells (FCTCs) have been previously identified as the most abundant cell type, found in the blastema of regenerating axolotl limbs”, a reference to Muneoka et al, 1986 is needed.

My expertise

I am not an expert on the immune system, so I don't know whether there might be other cells to consider beyond macrophages and septoclasts as shown here - or whether these are the appropriate markers. Also, I do not work on lizards so may not have the specifics of lizard regeneration, although I have experience in appendage regeneration in general.

Reviewer #3 (Remarks to the Author):

Vonk et al. aimed to reveal the cellular heterogeneity of lizard tail regeneration and establish a mechanism that can explain blastema and chondrogenesis formation, specifically through interactions between fibroblasts, immune system cells, and Shh signaling. The authors used scRNA-Seq, ISH, cell transplantation, clodrosome-based phagocyte depletion, and drug treatment. Certainly, the lizards offer exciting advantages to address questions related to regeneration. Likewise, the authors aim to integrate interaction between immune cells, chondrogenesis, and fibroblasts is an exciting direction. However, although the current manuscript contains many different experiments, the extent to which these experiments reflect the proposed cellular mechanism is extremely questionable.

The central problem is that the large majority of conclusions (as well as discussions) throughout the text are not supported by the provided data or the literature. Most of the problematic conclusions stem from confusion between what is a gene expression and a cell type. The authors do not perform any highly specific cell type manipulation experiment in this work, but most of their conclusions and the proposed model would require such methodologies. It is clear that the authors have a limited understanding of the limitations of scRNA-seq, which led them to propose unsupported conclusions and influence their discussion. Finally, both the text and figures could benefit from better explanations. Currently, the manuscript is jumping from one topic to another and needs to be easier to follow.

Although I find the aim, model organism, and methodologies interesting, the manuscript has multiple significant problems. As a result, the authors either have to include many new analyses and experiments (some of which are probably not accessible to their model organism), or re-interpret results and re-write the manuscript.

Major:

1- In Supp Fig 1A, the authors used DEG analysis and ISH to determine potential cell types. Although this approach could be valid, to this reviewer, these marker genes are not necessarily associated with

proposed cell types. A more common approach of indicating established markers would be helpful (e.g. Pax7 – muscle stem cell)

2- Authors use scRNA-seq to estimate cell type proportions. Although some groups may use such an approach, especially with in vivo complex tissues, this analysis alone cannot account for proportions as their samples vary in size and composition, and dissociation cannot be considered the same for all these populations. On this point, the authors' data already have problems. In figure 1B the authors show that FCTCs are more abundant in the blastema stage, but in Figure 2 all the FCTC markers are more highly expressed in the regeneration stage compared to the blastema stage (I might be wrong here because the figure quality is hard to see). Similarly, In figure 1B the authors show that at day 0 (prior to injury, intact tail) the largest population of cells recognized by single-cell sequencing are immune cells. Authors can perform alternative validation experiments to back up the proportion analysis, or soften their conclusions on this point.

3- Fig 1C: chondrocyte populations look separated from each other. Is there any explanation for this phenotype?

4- Fig 1D: it is tough to see where are the Day 0 samples. Can the authors show separate mapping for each stage? Otherwise, it is hard to appreciate how much similarity is present between the original tail and the “homeostasis” sample

5- Authors made many conclusions based on pseudo timing analysis (related to fig 1 and 2): these include the hierarchy of activation, fibroblast activation, resting cells, and differentiation trajectory. However, pseudotiming alone is not sufficient for these claims. The pseudotiming approach can be “suggestive” but not definitive and similar analyses have created problems before (please see e.g. PMID: 31974159). With current data, it is impossible to distinguish if the existing fibroblasts expand or if new cells gain these gene expressions. For this, authors need to use a lineage tracing approach combined with staining or other means. Authors could consider applying the *prx1* enhancer reporter as it is reported to be active in salamanders to mammals, or some different approach. Overall, conclusions based on scRNA-seq reach far beyond their capacities. Authors should either soften their conclusion as “suggest” with a note on limitations, or perform lineage tracing to support their conclusion. For example, “FCTC activation” and similar statements should be changed throughout the text.

6- Authors state “Furthermore, SAG treatment induced chondrogenesis in blastema regions that, under native conditions, would have differentiated into other tissue types, such as muscle and fat. These results suggested that Hh signaling is necessary and sufficient for inducing chondrogenesis in blastema FCTCs and that a large portion of blastema cells are capable of cartilage differentiation.”. There are several suggestions/conclusions not supported by the data related to these statements. Firstly, what is the evidence for the muscle and fat statement – no figure or literature has been referred. Likewise, what is the evidence that a large portion of blastema cells are capable of cartilage differentiation? For such a claim, authors need to analyze individual fibroblasts and their cartilage differentiation. These changes would influence the rest of the manuscript.

7- Related to SFig4: authors state SAG treatment did not induce chondrogenesis in the limb. Is there evidence showing cyclophosphamide or SAG treatment worked in the limb? Maybe limbs require a higher dosage to see an effect?

8- SFig 5: The authors used limbs as a generic healing control. Limbs and tails differ in their composition and proportion of cell types (limbs do not have spinal cord etc.), and one cannot be used to infer a healing mechanism for the other. If the authors wish to test the nonspecific healing response, they could damage the tail without amputating and check their phenotype.

9- In figure 3 and SFig5, authors conclude how certain gene expressions changed. However, these data are based on ISH results, which is not a quantitative approach. Moreover, these phenotypes' morphologies and cell numbers are changing, and we only see a section here. In figure 3i the authors claimed that "SAG treatment resulted in increased and expanded sulf1+ blastema areas" however, in the figure it doesn't look exactly the same as in the control animal in figure 2r. Was there any statistical analysis performed? If not, it's wrong to claim that sulf+ increased. I am not convinced about the conclusions related to these figures. Quantitative methods (e.g. qRT-PCR) with adequate normalization methods and replicates are necessary. These also influence Sfig6 and the rest of the manuscript conclusion.

10- Related to Fig 4 and Supp 7: The authors do not isolate pure FCTCs or fibroblasts, but their method is very crude tissue removal and grafting. This is one of the central problems related to grafting-based conclusions. None of these experiments have only FCTCs, and potentially they also include other cell types, which would confound the findings and conclusions. If the authors believe they only have FCTC, they can provide additional data, by doing scRNA-seq to isolated populations, staining, or any other means. Still, it is doubtful that they have a pure population.

11- In Fig 5 and SFig9, the authors compare specific immune system-related cell abundances for tails and limbs. Is it possible that these populations show different markers for these two models? Hence the limb may also contain these populations but lack the suggested markers.

12- Fig 7: was it previously established clodronate works in *L. lugubris*? If so, please add the reference, or related data.

13- Fig 7: In figure 7 the authors transplanted fibroblasts from clodronate or PBS-treated animals. However, they don't show the phenotype of these fibroblasts. Clodronate can be very toxic and induce cellular accumulation and myofibroblast activation. Without knowing the phenotype of fibroblast, you can't really compare it with that collected from PBS-treated animals. Perhaps a better experiment here will be to collect fibroblasts from regenerating animals and from the intact animals without amputation

and transplant them into a recipient animal. So, experiments seem to need more controls. Do the authors see the same phenotype when they do not mix PBS fibroblast and clodronate fibroblasts? Current results do not omit the possibility that their phenotype is affected by the mixing populations. Also, the recipient lizards in transplantation studies were all treated with SAG, no control animal recipients?

14- Fig 8: The authors perform conditioned media experiments. This is a very exciting experiment. However, there is no control media for the same purpose, and observed effects might be due to the media's synergistic effect and the cells cultured inside. Thus, I cannot comment on the findings from this part.

15- Authors claim to isolate macrophages and septoclast-like cells. How pure these populations are needed to be clarified.

16- Fig 10. Model fails in describing the phenomenon happening. Arrows represent direct interactions between populations, but these connections still need to be established in this study, as discussed above, mainly due to a lack of cell type-specific manipulations and experimentation.

17- In the discussion, the authors have several misleading statements related to single-cell analysis and comparison to salamanders. These statements show that authors need to familiarize themselves with the single-cell analysis and result interpretation, and they all require significant re-write. For example, the authors state salamander dataset has cycling cells, and theirs do not ???. It is generally hard to see cycling cells as a separate cluster in heterogeneous datasets. It doesn't mean salamanders have a dedicated cycling cell type?

Another example is "Comparing our lizard data sets with published salamander results, both scRNAseq methods captured similar populations of immune cells, epidermal, endothelial, muscle, blood cells, and connective tissue cells in lizards and salamanders. Focusing on the connective tissue cells, both salamander and lizard connective tissue cells expressed the markers ptx3, lum, col3a1, and dpt, and we can conclude that both salamander and lizard blastemas begin with similar cell populations." This statement is based on the annotation of different groups, not transcriptomes. If the authors want to discuss such a possibility, they can compare their scRNA-seq results to published axolotl datasets and make an unbiased comparison. Otherwise, this statement is just misleading. There are more similar instances throughout the discussion.

18- Authors state: "unlike salamander FCTCs, lizard blastema cells did not achieve a more embryonic cell state." What is the data indicating this? For this, authors need to compare their results to the developing tail, if possible via scRNA-seq, and then compare.

19- Currently, most protocols need to be more suitable for replication. Please provide the catalog number for the materials. Likewise, please provide a more detailed protocol for ISH and IF. E.g., what is the blocking agent during IF, staining duration etc.

20- Authors wrote n numbers for animals and tails used for analysis, but how many independent experiments were performed is not stated. Are these experiments done only once with these animals? If so, how valid are the statistics choices?

21- In the intro, the authors state this is the first study to perform scRNA-Seq on lizards. This statement is wrong and confusing at the same time. The same group already published a paper with lizard scRNA-seq, which is not cited here (PMID: 32337387). Additionally, another group published lizard scRNA-seq: PMID: 34873160

22- Tails treated with Cycloamine in figure S3 and figure S4 look completely different. In figure S3 the tail regenerated without bone by 28dpa, but in S4 it did not! Is there any explanation for this? Which one is the phenotype?

Minor:

1- I assume there was a problem during submission, as the figure quality is terrible and needs an update. Staining results throughout the text is tough to assess in this state.

2- The authors refer to their data collection time points as “original tail (day 0), inflammatory stage, blastema stage, or regenerated homeostasis “. Please state the days instead of using terminologies throughout the main text, as every group has different definitions for these. Indeed, In their previous paper, this group named blastema stage at 7dpa, but here the same time point is referred to as the inflammatory stage and blastema are at 14 dpa. Please explain the difference. This simple change would enhance the readability of the manuscript.

3- Authors state “mouse digit tip regeneration is lineage-restricted, and mouse blastema cells are unable to differentiate into cartilage.“, This is not completely accurate. Neonatal mouse digit tip amputations induce blastema to differentiate into cartilage, and adult mouse amputations result in differentiation into bones. ADD REFERENCE

4- Literature referencing is weak. Some statements do not have any reference, and others are under-referenced. I am listing some of these instances from the text, but the authors need to go through the whole manuscript as there are more instances.

- a. “Additionally, lizards have more complex and adaptive immune systems, more like that of mammals, contrasting the rudimentary immune systems exhibited in amphibians lacking efficient adaptive immunity. “ Please add references
- b. “Specifically, the recruitment of macrophages to injury sites has been shown to be crucial for appendage regeneration in several model organisms.” There are more studies done on this point, please add more comprehensive list or state why some others are omitted.
- c. “Among the cells recruited by macrophages during the immune response, fibroblasts have been shown to activate and migrate to areas of tissue loss in a coordinated manner. “ Reference, please.
- d. “In axolotl, these blastema FCTCs mimic embryonic limb bud signatures before redifferentiating into the patterned limb skeleton and connective tissues. “There are more studies done on this point, please add more comprehensive list or state why some others are omitted.
- e. There is a recent paper on axolotl osteoclasts; please cite PMID: 36218256
- f. “Recent interrogations of blastema formation in the neotenic axolotl salamander (*Ambystoma mexicanum*) via single-cell RNA sequencing (scRNAseq) have highlighted the importance of fibroblastic connective tissue cells (FCTCs) in blastema derivation and regenerated tissues.” There are more studies done on this point, dating back to Susan Bryant s initial work – not just recent interrogations, please add a more comprehensive list or state why some others are omitted.

5- The authors state in the abstract, “Lizards do not naturally regenerate limbs but are the closest relatives of mammals capable of epimorphic tail regrowth “. Although this is true to our current knowledge, there is no systematic study where all animal tails were cut and tested for regeneration. So, a better statement would be, “Lizards do not naturally regenerate limbs, but they are the closest relatives of mammals identified so far that are capable of epimorphic tail regrowth “

6- Figure 6 should not be a major figure in the paper, and it should move to supplementary or not shown at all. The authors have already shown that clodronate treatment in lizard tails prevents blastema formation (surprisingly, it was not referenced here, PMID: 32337387), and of course since there is no blastema, there wouldn't be any blastema-related fibroblasts.

7- The discussion almost reads like the discussion of another paper, as authors spent a great deal of time comparing lizard and salamander single-cell data sets without showing any figures or methodologies on how these comparisons were made. A discussion revolving around shown data would be necessary.

Response to Reviewers' Comments

We thank the reviewers for their insightful suggestions and thoughtful feedback. This document addresses reviewers' concerns and comments point-by-point. An additional author, Sasha B Sengelmann, has been added to reflect her contributions to the revised manuscript.

Reviewer #1 (Remarks to the Author):

Abstract

1. **PLEASE briefly explain (here or in the introduction) the term “pseudotime trajectory analysis”**

Thank you for bringing this to our attention. We have added a brief description of pseudotime trajectory analysis to the revised results section at first mention of the technique.

Revised statement: “Pseudotime trajectory analysis, a computational method used to infer biological transitions or cell lineage relationships based on scRNAseq gene expression profiles⁴⁸, was performed on the FCTC/chondrocyte subcluster (Fig. 1e, f) using R package Monocle2^{49–51} and revealed one minor and two major branch points of potential cell fate trajectory change over the course of regeneration.”

2. **please SPELL OUT *spp1* and *sulf1***

We appreciate the suggestion to define these gene symbols for reader clarity. Due to the abstract word limit, we are unable to utilize the long-form name for these genes in this section, however, we have added these descriptions of secreted phospholipid protein 1 (*spp1*) and sulfatase 1 (*sulf1*) to the revised results for clarity, as well as the long-form names for all genes at their first mention.

3. **.....and indicate POSSIBLE/POTENTIAL actionable targets.....**

Thank you for the suggestion. We have revised the abstract to reflect this recommendation.

Introduction

4. **the sentence:.....non-mammalian vertebrates MAY/MIGHT provide clues.....**

Thank you for the suggestion. We have revised the introduction to reflect this recommendation.

5. **the sentence:.....mouse tip regeneration...: RECENT STUDIES HAVE INDICATED THAT THIS IS A “regengrow” NOT A REAL REGENERATION (see Alibardi L (2021) Regeneration in anamniotes was replaced by regengrow ... biological cycles. Dev Dynam, Special Issue 251, 1404-1413.**

Thank you for bringing this to our attention. We have revised our introduction on mouse digit tip regrowth to reflect the difference between this phenomenon and true epimorphic regeneration.

6. **the sentence:.....lack true tendons 1-7. NUMEROUS RECENT AND PAST EXTENSIVE STUDIES ON LIZARD REGENERATION ARE NOT CITED here LEAVING THE READER UNINFORMED ON WHAT IS PRESENTLY KNOWN ON THE TOPIC also in other lizards, from anatomy, histology, cell biology to immunology. PLEASE at least CONSIDER ALSO OTHER few STUDIES, FOR EXAMPLE :**
- Fisher, R. E., Geiger, L. A., et al. (2012). A histology of the original and regenerating tail in the gree anole, *Anolis carolinensis*. *The Anatomical Record*, 295, 1609–1619.
-Alibardi L 2014 Alibardi, L. (2014). Histochemical, biochemical and cell biological aspects of ...tissue regeneration. *Progress in Histochemistry and Cytochemistry*, 48, 143–244.

-Jacyniak, K et al. (2017). Tail regeneration and other phenomena of wound healing ... lizards. Journal of Experimental Biology, 220, 2858–2869.

Thank you for bringing this literature to our attention. We apologize for the oversight and have included the recommended citations in the revised introduction and references.

7. **the sentence:.....Macrophages act in the.....: ThE SENTENCES SHOULD BE UP-DATED. IN FACT, DETAILED STUDIES ON M1, M2 macrophages and TREGs –uncited here- IN LIZARD BLASTEMAS ARE REPORTED IN:**
-Alibardi L. 2016. Immunolocalization of 5BrdU long retainingafter limb amputation. Tiss Cell 48: 197-207.
-Alibardi L. 2020. Autoradiography and immunolabeling suggests ...support tail regeneration. Ann Anat 231, 151549.
-Alibardi L. 2022. Immunoreactivity for Dab2 and Foxp3 ... of lizard. Acta Zool 103, 389-401.

Thank you for bringing this literature to our attention. We apologize for the oversight. We have revised this paragraph to include background information about macrophages and immune cells in lizards, specifically, and have included these citations in the revised introduction and references.

Results

8. **AS A GENERAL REMARK THE AUTHORS SHOULD INITIALLY -HERE OR IN MAT&METHODS- MAKE A LIST OF THE MAIN GENES SEARCHED FOR (SPP1, SULF1, CDH11, PLTP, SALL1 ETC.) EXPLAINING THAT THEY WERE IN PARTICULAR INTERESTED IN THOSE RELATED TO CARTILAGE DIFFERENTIATION. THIS WOULD HELP THE READER TO GRASP THE MAIN EXPERIMENTAL TARGETS OF THE PRESENT AUTHORS.**

Thank you for this suggestion. Our scRNAseq analysis helped us to identify these markers for resting, injury-state, and blastema fibroblasts as they progress through regeneration into cartilage differentiation. Although many marker genes we discussed have significance in bone and cartilage development, they were chosen for further investigation primarily due to their high levels of differential gene expression in the fibroblast cluster of our sequencing analysis. *Cdh11*, *col3a1*, *col12a1*, *mdk*, *pltp*, *sall1*, *sox9*, *spp1*, *sparc*, *sulf1*, and *tnl* were chosen for further interrogation due to their high levels of expression in our scRNAseq fibroblast cluster and were validated via ISH to show specificity for fibroblastic connective tissue cells (FCTCs), shown in Figure 2 and Supplementary Figure 5. Follow up analysis was performed on genes that were specifically activated in FCTCs beginning at the blastema stage, *sulf1*, *pltp* and *sall1*. We assessed these genes for Hedgehog responsiveness in tail and limb, now both shown in revised Figure 3, revealing *sulf1* as the only specific FCTC blastema marker expressed in the regenerating tail, but not the non-regenerating limb, that was responsive to Hedgehog signaling, leading to subsequent transplant and conditioned media experiments. We have added Supplementary Table 1 to clarify the stages at which each fibroblast gene is expressed throughout regeneration, shown below.

Supplementary Table 1. Relative fibroblast marker gene expression throughout tail regeneration. Relative gene expression levels of FCTC markers (assessed from ISH in Figure 2 & Supplementary Figure 5). -, no detectable expression; +, low expression; +++, high expression.

Marker gene	Homeostatic Fibroblasts (0 DPA)	Injury Fibroblasts (7 DPA)	Blastema Fibroblasts (14 DPA)	Chondrocytes (28 DPA)
cdh11	+	+	+++	-
col3a1	+++	+++	+++	+++
coll2a1	-	+	+++	+++
mdk	-	+	+++	+
pltp	-	-	+++	+
sall1	-	-	+++	+
sox9	-	-	+++	+++
sparc	-	+	+++	+++
spp1	-	+++	+++	-
sulf1	-	+	+++	+
tnl	-	+++	+++	-

9. on page 4, the sentence:.....inflammatory stage (DAY ?), blastema stage (DAY ?), AND regenerating homeostasis (DAY ?). PLEASE INDICATE DAYS/range of DAYS IN EACH STAGE.

Thank you for this point of clarity. We have added the recommended details to this statement in the revised results.

10. pag 5, the sentence:....., *cd8a*⁺ and *cd8b*⁺.....:PLEASE spell out CD8a,B (ARE THEY CYTOTOXIC T-CELLS?)

We have defined *cd8a* and *cd8b* in the revised results section. While *cd8a* and *cd8b* expression mark cytotoxic T cells in mammals, our previous work with lizard blood cells has shown that some blood cell markers do not directly correlate to mammalian marker genes. For example, when comparing scRNAseq gene expression of mouse (*Mus musculus*) and green anole (*Anolis carolinensis*) peripheral blood, mouse T lymphocytes expressed *Cd4*, *Cd8b*, *Cd247*, and *Cd226*, while corresponding lizard T cells expressed *cd247* and *cd226*, but not *cd4* or *cd8b*. Further, lizard B cells expressed *cd8b* while mouse B cells did not. Thus, we are unable to confidently identify the cell populations within the *cd8a*⁺ and *cd8b*⁺ clusters at this time (PMID: 32337387). Further interrogation of these and other immune cell clusters within this dataset will be investigated in future research.

11. pag. 5, the sentence:.....through gene expression analysis: PLEASE BRIEFLY EXPLAIN THIS TYPE OF IDENTIFICATION

We have elaborated on this methodology in the revised results section and included the top differentially expressed genes used to mark specific cell populations within each validated cluster.

12. Pseudotime trajectory analysis: WHAT DOES IT REVEAL ? THIS IS IMPORTANT TO UNDERSTAND THE FOLLOWING DESCRIPTIONS.

Thank you for bringing this omission to our attention. We have added a brief description of pseudotime trajectory analysis to clarify the purpose of this analysis and its findings.

Revised statement: “Pseudotime trajectory analysis, a computational method used to infer biological transitions or cell lineage relationships based on scRNAseq gene expression profiles, was performed on the FCTC/Chondrocyte subcluster (Fig. 1d) using R package Monocle2²⁵⁻²⁷ and revealed one minor and two major branch points of cell fate trajectory change over the course of regeneration.”

13. *spp1*: please SPELL OUT THE NAME OF THIS GENE, AS WELL AS SULF1, CDH11 is what?

For clarity, we have added the long-form gene names for each *spp1* (secreted phosphoprotein 1), *sulf1* (sulfatase 1) and *cdh11* (cadherin 11) to their first use in the revised results section.

14. Lizard tail blastemas are...FCTCs”: WHICH key genes for “fibroblast activation” were here evaluated?

We apologize for the lack of clarity here. We have revised the terminology used when referring to FCTCs to avoid ambiguity regarding “activated” and “resting” cell states considering the diverse gene expression activity at each regeneration phase. Instead, we have renamed our FCTC groups based on tail location to distinguish differential FCTC gene expression over the course of regeneration. Thus, the FCTCs at 0 DPA are referred to as homeostatic fibroblasts, 7 DPA are injury fibroblasts and 14 DPA are blastema fibroblasts. The listed fibroblast genes were evaluated due to their top differential expression within the fibroblast cluster of the regenerating tail scRNAseq dataset and were validated via ISH and histology over regeneration stages to characterize FCTCs in anole and highlight the sequential addition each FCTC marker gene expression at given stages, as discussed previously in response to comment #8.

15. pag. 6, the sentence:.....During blastema formation AT 14 DPA....

We have amended this sentence to reflect the suggestion above.

16. pag. 7, the sentence:.....During tail elongation AT 21 DPA....

This sentence has been amended to include relevant DPA designations as they correspond to revised Figure 2 and Supplementary Figure 5.

17. Lizard blastema FCTCs...signaling, pag 7

We have amended this sentence to reflect the suggestion above.

18. the sentence:.....Lizard tail and limbs were collected 28 DPA from ANIMALS treated with...

We have amended this sentence to reflect the suggestions above.

19. the sentence:.....septoclast population: CAN THE AUTHORS BRIEFLY MENTION WHAT A SEPTOCLAST IS for the benefit of readers? (Osteoclasts derived from pericytes?)

Thank you for this suggestion. We have included a brief description of septoclasts (phagocytic cells from a non-myeloid lineage) and related reference (PMID 35091558) to elaborate on this statement.

20. In summary, the results described above.....: ARE THE PRESENT AUTHORS AWARE THAT FGFs (1 OR 2) ADMINISTRATION TO THE LIMB STUMP IN LIZARDS PRODUCES RELATIVELY LARGE AMOUNT OF CARTILAGE (femur, tibia and fibula)? HAVE THEY TESTED/WILL THEY BE ABLE TO TEST FGFs IN LIMBS FOR-perhaps- STIMULATING EXPRESSION OF THEIR tail CHONDROCYTE MARKERS also in THE LIMB treated with FGF? SEE: Alibardi L. 2017. FGFs treatments on amputated limbs stimulate the regeneration of long bones, opening new avenues for limb regeneration in amniotes. A morphological study. *J Morphol Phys Kinesiol* 2, 25.

We thank the reviewer for this inquiry. Toward addressing this comment, we have completed studies investigating the roles of FGF signaling in lizard limb cartilage regeneration, and we provide the figure below, summarizing some of our findings. In these studies, the effects of FGF-2-soaked beads on amputated lizard (*A. carolinensis*) limb healing were tested. Amputated lizard limbs were treated with FGF-2 or control beads and analyzed by histology 28 DPA. n = 5 lizards per treatment condition, representative images shown. *, location of implanted bead; cc, cartilage callus. Bar = 100 μ m. FGF-2 bead-treated limbs exhibited larger cartilage calluses (**a-d**) containing significantly more EdU⁺ cells compared to controls (**g**). Unpaired two-way Student's *t*-test was used. **, $p < 0.01$. We have previously shown that callus cartilage is distinct from blastema-derived cartilage (PMID: 27387871); callus cartilage is derived from periosteal cells and undergoes endochondral ossification, while blastema fibroblast-derived cartilage resists ossification for the lifetimes of regenerates. Here, we validated that cartilage detected in FGF-2 bead-treated amputated lizard limbs was callus cartilage and not blastema-derived cartilage. Limb samples from our experiments involving treatment with beads soaked in septoclast-like cell (herein, septoclasts) culture media (CM) were used as positive controls of ossification-resistant, blastema fibroblast-derived cartilage. All *col2a1*⁺ cartilage in FGF bead-implanted limbs expressed terminal hypertrophy marker *col10a1* via ISH, indicating endochondral ossification (**c, d**). Conversely, cartilage detected in septoclast cell CM-treated limbs did not (**e, f**). These results agreed with those presented in the reference provided by the reviewer and confirmed that FGF beads stimulated production of callus cartilage and bone formation in amputated limbs. However, FGF did not induce blastema fibroblast-derived cartilage or *sulf1* expression in limbs. Thus, these studies distinguish callus cartilage induced by FGF treatment from Hh-responsive chondrogenesis induced by septoclast-CM. We are in the process of preparing the FGF/cartilage/endochondral ossification stories for publication, but they are beyond the scope of the current publication.

Discussion

21. IN GENERAL THIS PART OF THE MS CAN BE sensibly SHORTENED ELIMINATING REPETITIONS AND KNOWN PROCESSES. THE FIRS 4-5 SENTENCES REPEAT THE INTRODUCTION. ...AND (not mandatory however) THE ORGANIZATION OF THE DISCUSSION INTO SUB-SECTIONS WOULD MADE THE TEXT MORE READABLE as the text is exsessively LONG and unstructured (eg. one subheading on the cartilage influence ...; and another on theImmunological implications).

Thank you for your input. We have extensively revised the discussion section and added subheadings to provide a clearer and more concise context in which to discuss our results.

22. the sentence:.....all species tested regrow near perfect copies..... THIS STATEMENT IS NOT REALLY TRUE AS SEVERAL URODELES, ESPECIALLY TERRESTRIAL-ADAPTED (SALAMANDERS) DO NOT REGENERATE (forming SCARring outgrowths) OR only SHORTER AND HETEROMORPHIC LIMBS (Scadding, S.R. (1977). Phylogenetic distribution of limb regeneration capacity in adult Amphibians. *Journal of Experimental Zoology* 202, 57-68. Scadding, SR. (1981). Limb regeneration in adult amphibians. *Canadian Journal of Zoology* 59, 34-46.

This is an excellent point about the diversity of urodeles and an oversight we missed. We have revised this statement to reflect this information.

23. the sentence:....the first MOLECULAR interrogation into the heterogeneity of....: HOWEVER IT IS KNOWN, BASED ON MORPHOLOGICAL AND LABELING STUDIES (ultrastructure, autoradiography and immunolabeling), FROM LONG TIME that the lizard blastema AND THE ORIGIN OF ITS CELLS IS HETEROGENOUS, CELLS DERIVED FROM different tissues and others from dedifferentiation, as it is reported in the literature.

This is an excellent point. We have amended this sentence to reflect the suggestion above.

24. the sentence:.....as salamander cells although this may be attributed to cell sampling and accessibility differences???: PREVIOUS STUDIES INDICATED THAT AFTER AUTOTOMY ALONG THE AUTOTOMOUS PLANES MAINLY FIBROBLASTS OF THESE SEPTA REMAIN AND THAT THE BLASTEMA IS ESSENTIALLY DERIVED FROM EITHER THE DE-DIFFERENTIATION FORM THESE CELLS BUT MOST LIKELY FROM THEIR STEM-LIKE NATURE (refs in the literature, see Simpson SB 1968 Amer Zool etc.).

Thank you for bringing this to our attention. We have removed this section from the revised discussion but have included the aforementioned reference in the manuscript, given its significance in lizard tail regeneration.

25. the sentence:....but these were limited to FCTC activation of several genes.... THIS IS INDEED AN IMPORTANT NEW FINDING OF THE PRESENT STUDY SINCE LIZARD REGENERATION LIKELY HAS EVOLVED A MECHANISM OF HETEROMORPHIC REGENERATION BY THE Re-USE OF A LIMITED NUMBER OF GENES THAT WERE INITIALLY USED DURING TAIL EMBRYOGENESIS (see Alibardi, L. (2020). Appendage regeneration in anamniotes utilizes genes active ... studying lizard tail regeneration. Journal of Morphology 281, 1358-1381.

Thank you for highlighting the significance of this distinction. We have significantly revised the discussion section, excluding the statement above, but we have highlighted this difference between lizards and anamniotes, and included the suggested reference.

26. YES, TEM studies showed that lizard blastema cells NEVER become ultrastructurally as simple as the mesenchymal cells of newts or the axolotl.

Thank you for bringing this to our attention. We have included the following statement with reference (PMID: 29885003) in our revised discussion.

Revised statement: “Previous studies have also identified structural differences in urodele and lizard blastema cells that may reflect differences in de-differentiation and cell states, which could influence plasticity during regeneration⁸⁶.”

27. THE PRESENT AUTHORS MAY ALSO CONSIDER IN THE FUTURE TO EVALUATE FOR GENES INVOLVED IN SOMITOGENESIS, expressed in lizard embryos BUT NOT In the regenerating tail (see Eckalbar et al. (2012). Somitogenesis in the anole lizard and alligator reveals evolutionary convergence and divergence in the amniote segmentation clock. Developmental Biology 363, 308-319.

Thank you for this suggestion. Segmentation and somitogenesis will be a topic of future study but is beyond the scope of the current manuscript.

28. the sentence:.....for example, comparing tail blastema versus amputated limb wound healing: BUT THIS STUDY HAS BEEN DONE, AND NUMEROUS GENES WERE DETECTED AS

DIFFERENT FROM TAIL VS LIMB, INCLUDING CONNECTIVE tissue GENES: SEE- Vitulo N, et al. 2017. Transcriptome analysis of the regenerating tail versus the scarring limb ... organ regeneration in amniotes. Dev Dyn 246: 116-134.
- Vitulo N, Dalla Valle L, Skobo T, Valle G, Alibardi L. 2017. Down-regulation of lizard immunogenes in the regenerating tail and myo-genes ...immuno-privileged organ. Protoplasma. 254: 2127-2141.

-Patel, et al. (2022). De novo transcriptome sequencing and analysis of differential gene expression ...in Hemidactylus flaviviridis. Journal of Developmental Biology 10, 24.

We apologize for this unclear statement and appreciate the associated literature suggestions. This statement has been revised to reflect its intended emphasis on the value of scRNAseq studies specifically in these cases.

29. In their histochemical study, the activation of *sox9* for cartilage regeneration in lizard is reported in Delorme et al, 2012. Scar-free wound healing and regeneration... Eublepharis macularius. Anat. Rec. 295, 1575-1595.

We have added this reference to our first mention of *sox9*⁺ chondrogenesis in our revised results section.

Materials and Methods

30. As a GENERAL REMARK, the numerous ACRONYMS UTILIZED (SAG, NBF,

Thank you for bringing this to our attention. We have searched the manuscript and defined all acronyms and symbols for gene names at first mention. For example, Hedgehog smoothed agonist (SAG); neutral-buffered formalin (NBF).

31. pag. 21, the sentence:....therapeutics39, which have been shown to impact tail regeneration (ANY CITATION HERE ? or PERSONAL OBSERVATIONS?)

Thank you for highlighting this oversight. We have added the following references to this statement to highlight the impact of immunosuppressant drugs on wound healing and regeneration in various organisms: PMIDs 23690624, 11074878, 6855226. Additionally, we have observed similar reduction in regenerative response for tail injuries in green anoles (*A. carolinensis*), depicted below. Lizards were treated with 1 µg/g rapamycin via intraperitoneal (IP) injection, or 0.5 µg/g cyclosporine via IP injection, every 72 hours for 3 weeks. Tail lengths were measured and compared to controls. Both rapamycin and cyclosporine, commonly used immunosuppressant drugs in transplantation studies, significantly inhibited tail regeneration. n = 5 animals per treatment condition. Unpaired two-way Student's *t*-tests were used. ****, *p* < 0.0001.

32. Lizard tail amputations:.....Lizard tails were anesthetized BY EXPOSING THE ANIMALS TO A SPRAY OF ETHYL CHLORIDE FOR 10 SECONDS and amputated.....

We have amended this sentence to reflect the suggestions above.

33. the sentence:....and 28 days after initial amputation OR DPA (DAYS POST-AMPUTATION) for histology.....

We have amended this sentence to reflect the suggestions above.

34. DATA AVAILABILITY the sentence:....dataset are available at GSEXXXXXX.....: PLEASE WHAT DOES IT MEAN ?

For all next-generation sequencing studies, the journal and United States National Institutes of Health (NIH)-funding sources ask that raw sequencing data and gene expression analysis be publicly available to readers at the time of publication. Nature Communications recommends utilizing the National Center for Biotechnology Information (NCBI) Gene Expression Omnibus (GEO) repository for scRNAseq processed expression data, raw sequencing data and metadata. When a dataset is submitted to NCBI GEO, the repository will process the data and create a unique GEO Series entry (GSE) that includes the data and a summary of the study. Currently, we are preparing our data for NCBI GEO submission and will provide an updated GSE number (shown currently in the text with placeholder “GSEXXXXXX”) when the record has been generated, as well as a link to this GEO entry for the online publication. This GSE number and link will be available at the time of publication. Additional information on the NCBI GEO repository can be found here: <https://www.ncbi.nlm.nih.gov/geo/info/faq.html>

Reviewer #2 (Remarks to the Author):

Validity

- 1. It seems strange to include 1DPA and 3DPA as timepoints in an “inflammatory stage” for scRNA-seq, but then to effectively ignore these timepoints in any later experiments.**

Thank you for bringing this to our attention. Given that there is minimal tail growth in the early days post amputation (DPA), including 1, 3 and 7 DPA, inflammation-related time points presented a challenge for single-cell RNA sequencing (scRNAseq) sample collection due to limited cell number and high levels of debris in cell dissociations. Additionally, the inflammatory stage of regeneration, in particular, seemed to show high levels of variability between samples in terms of morphology (i.e. wound closure rates, histolysis activity at amputation sites, scab formation, scab loss rates, etc.), so we included additional time points prior to 7 DPA, to cover the full range of immune activity during tail regeneration and variability between samples.

We have included Supplementary Figure 14 in the revised manuscript, inserted below for reference, to show the representative fluorescent *in situ* hybridization (FISH) and quantification of relative percentage area of immune cells (*aoah*) and phagocytic macrophages and osteoclasts (*ctsb*) within the regenerating tail at time points 0-14 DPA (**a-f**). 0 DPA shows relatively low amounts of immune and phagocytic cells, while 1, 3, and 7 DPA show large percent increases in immune and macrophage cell populations. Although *aoah*⁺ immune cells are relatively less abundant at 7 DPA, compared to 1 and 3 DPA, *ctsb*⁺ macrophages and osteoclasts percentage area peaked at 7 DPA. Bar = 500 μm. Given the manuscript focus of phagocytic cell interactions with fibroblasts in tail regeneration and the still relatively large proportion of immune cells present at 7 DPA, we opted to assess tails at this time point for figures referencing the inflammatory stage, as described in the revised methods section.

2. Fig. 1b raises the question of why 28DPA is considered a regenerated homeostasis, and light-microscope images of tissues at each of these timepoints would serve to validate this. My concern is due to the relatively low difference in cellular makeup of the “regenerated homeostasis” and “blastema stage” and the relatively high difference between the “regenerated homeostasis” and “original tail” stages. Though it is known that lizard tail regeneration is imperfect, it seems that 28DPA may not thoroughly represent a full regenerate, such as with the high number of actively cycling cells.

Thank you for bringing this ambiguity to our attention. While tails may continue to elongate after 28 days post-amputation (DPA), as evidenced by abundance of cycling cells, we have utilized this time point as it represents the stage in which the majority of major cell and tissue types (i.e. regenerated muscle, cartilage tube, regenerated spinal cord, adipose tissue, peripheral nerves, blood vessels, interstitium/connective tissues, etc.) of the regenerated lizard tail can be detected in their terminally differentiated form and are typically easier to process for sectioning and staining analysis. Compared to 28 DPA samples in revised Figure 2 and Supplementary Figure 5, lizard tails 84 DPA, shown below, exhibit similar FCTC gene expression patterns. *ve*, vertebrae; *ct*, cartilage tube. Bar = 100 μ m. Additionally, in response to this and other reviewers' comments on efficacy of total cell population quantification from scRNAseq data, we have chosen to add an additional figure, Supplementary Figure 2, to depict FISH quantification of proportional cell type contributions of selected scRNAseq clusters to each regenerating tail stage time point.

col3a1

pltp

sall1

sox9

spp1

sulf1

- 3. It is not entirely clear from the pseudotime trajectory that *sulf1* cells reliably become *soxa9*⁺ and *col2a1*⁺. Could *sulf1*⁺ cells in the other branch contribute to the regeneration process, and if so, how?**

We intended to utilize pseudotime trajectory analysis as a supportive model to our overarching hypotheses on sequential FCTC gene expression acquisition throughout regeneration, rather than a definitive method to determine lineage relationships and cell state changes in fibroblasts and chondrocytes through scRNAseq data alone. Thus, we have revised the text to soften our results and discussion claims for this analysis, instead emphasizing its utility in forming our hypotheses about fibroblast gene expression pattern changes over the course of regeneration and to use as a basis for subsequent experiments testing these theories later in the manuscript.

We later show *sulf1*⁺ fibroblasts are specifically Hedgehog (Hh) responsive, and previous work has shown that Hh signaling is required for chondrogenesis in tail regeneration (PMID: 27387871). Subsequent fibroblast transplantation experiments revealed only *sulf1*⁺ fibroblasts were capable of forming cartilage while *sulf1*⁻ were unable to activate chondrogenic programming, confirming the pseudotime trajectory model in which some *sulf1*⁺ fibroblasts, those given the appropriate Hh signals, express *sox9* to activate chondrogenesis and eventually form *col2a1*⁺ cartilage.

It is certainly possible that *sulf1*⁺ fibroblasts are contributing to the regeneration of other tissues in the tail, given that not all *sulf1*-expressing fibroblasts receive Hh signaling if they are distal to the regenerating spinal cord and thus, will not form cartilage. Given the diversity of pathways *sulf1* has been shown to interact with and the vast functional implications of sulfatase activity in other tissues and organisms, other *sulf1*⁺ fibroblasts could easily be aiding in the regeneration of other tissues, such as connective, muscle or adipose tissues, in the regenerating tail, but further investigation into *sulf1*⁺ fibroblasts that did not form cartilage was beyond the scope of this manuscript.

The body of the revised manuscript reflects the points discussed above.

- 4. With regards to Fig. 2 and related findings, *cdh11*⁺ FCTCs are referred to being in a “resting” state, but are highly active in the “blastema” and “regeneration” stages.**

Thank you for pointing out this confusing terminology. We have revised the terminology used when referring to FCTCs to avoid ambiguity regarding “activated” and “resting” cell states. Instead, we have renamed our FCTC groups based on tail location to distinguish differential FCTC gene expression over the course of regeneration. Thus, the FCTCs at 0 DPA are referred to as homeostatic fibroblasts, 7 DPA are injury fibroblasts and 14 DPA are blastema fibroblasts. We have also emphasized sequential addition of FCTC marker gene expression throughout each regeneration stage in the text and newly added Supplementary Table 1 to clarify specific gene expression patterns of identified FCTC markers in fibroblasts throughout the course of tail regeneration.

Significance

- 5. It may be a stretch to claim that the study identified “a majority of fibroblast activation markers”. There is no way to be sure of this, nor was a novel molecule discovered, such as an equivalent to *prrx1* in salamanders.**

We agree that this may have been an overstatement and have revised this discussion claim accordingly.

- 6. It is relatedly a stretch to suggest *sulf1* is “a master regulator” for blastema formation.**

We agree that this may have been an overstatement and have softened this discussion claim accordingly.

7. **Based on what is currently shown, it may not be appropriate to say *cdh11* is repurposed to fill the role of embryonic *cdh2*, as the experiments did not explore this and the lizard's regeneration is imperfect – if it is true, it's not a perfect recapitulation.**

Thank you for pointing out this overstep. We have removed this section from the discussion as it is not backed by sufficient experimental results from this study.

Data and methodology

8. **For scRNA-seq, the combination of 1DPA, 3DPA, and 7DPA cells into one “inflammatory stage” sample is concerning without proper context. 1DPA and 7DPA may represent vastly different cell states, with 1DPA more of a wound healing stage. Subsequent ISH and other analyses utilize only 7DPA samples of this stage, where certain results would be strengthened by the inclusion of earlier timepoints. Otherwise, justification for the inclusion of just 7DPA ISH should be provided.**

Thank you for bringing this to our attention. Given that there is minimal tail growth in the early days post amputation (DPA), including 1, 3 and 7 DPA, inflammation-related time points presented a challenge for single-cell RNA sequencing (scRNAseq) sample collection due to limited cell number and high levels of debris in cell dissociations. Additionally, the inflammatory stage of regeneration, in particular, seemed to show high levels of variability between samples in terms of morphology (i.e. wound closure rates, histolysis activity at amputation sites, scab formation, scab loss rates, etc.), so we included additional time points prior to 7 DPA, to cover the full range of immune activity during tail regeneration and variability between samples.

We have included Supplementary Figure 14 in the revised manuscript, inserted below for reference, to show the representative fluorescent *in situ* hybridization (FISH) and quantification of relative percentage area of immune cells (*aoah*) and phagocytic macrophages and osteoclasts (*ctsb*) within the regenerating tail at time points 0-14 DPA (**a-f**). 0 DPA shows relatively low amounts of immune and phagocytic cells, while 1, 3, and 7 DPA show large percent increases in immune and macrophage cell populations. Although *aoah*⁺ immune cells are relatively less abundant at 7 DPA, compared to 1 and 3 DPA, *ctsb*⁺ macrophages and osteoclasts percentage area peaked at 7 DPA. Bar = 500 μ m. Given the manuscript focus of phagocytic cell interactions with fibroblasts in tail regeneration and the still relatively large proportion of immune cells present at 7 DPA, we opted to assess tails at this time point for figures referencing the inflammatory stage, as described in the revised methods section.

9. The combination of 14DPA and 21DPA samples into one “blastema stage” is similarly concerning, especially given that later in the manuscript (Fig. 2) 21DPA is instead referred to as “regeneration stage”, whereas in the methods and scRNA-seq the 28DPA timepoint is stated as a homeostatic “regeneration stage”. In other models of regeneration, these again represent vastly different cell states. Though both of these DPAs are utilized in later analyses, for the purposes of scRNA-seq it’s unclear why they are combined.

Thank you for highlighting this point. We have revised Figure 2 and Supplementary Figure 5 to include 28 DPA tails as the “Regenerated Homeostasis” stage samples.

There is inherent variability in tail growth and regeneration between individual, field-collected lizards, so matching sample groups based on DPA alone is often not sufficient to assess reproducible morphological and phenotypical states of tail regrowth. As shown in other lizard species (PMID: 21846350), regeneration follows a reliable pattern, but the timeline of that pattern can vary significantly between individuals. Regeneration rates may vary due to a plethora of factors, including habitat temperature, photoperiod exposure timing, amputation location on the tail, sex, age, diet, and overall lizard health (PMID:34940500). To ensure consistency among sample groups, in addition to tracking sample DPA, we assessed samples for morphology and characteristic regeneration stage phenotypes, with inflammatory stage tails ranging from the closing of the wound epidermis to scab accumulation, blastema stage tails ranging from the loss of the amputation site scab to tail cone shape formation and homeostatic regeneration stage began after the formation of tail cones through elongation and full growth. 14 and 21 DPA samples

were both morphologically assessed as blastema stage samples and thus were combined during scRNAseq analysis. This distinction has been made in the revised results and methods sections of the manuscript.

10. In Fig. 4 and related findings – Is it possible that 14DPA could be too late of a blastema stage to activate limb fibroblasts; is 14DPA considered early blastema, or mid in this context?

Blastema stage recipient lizards at 14 DPA in transplantation experiments were in the mid-blastema stage according to morphological progression, as these samples had lost their injury site scab and formed a rounded shape due to accumulation of blastema cells at the amputation site, but tails had not yet begun to form a cone shape indicative of later blastema stage.

As shown in revised Figure 3, native limb fibroblasts do not express *sulf1*, regardless of Hedgehog (Hh) signaling modulation at 14 DPA. Below, we have included limb samples collected 7 DPA from lizards treated with Hh inhibitor cyclopamine and Hh smoothed agonist SAG and analyzed for *sulf1* expression via ISH. tb, tibia. Bar = 500 μ m. Regardless of treatment condition, limb fibroblasts 7 DPA do not express *sulf1*. Furthermore, in Supplementary Figure 11, both limb and blastema fibroblasts transplanted into new tail amputation sites at 0 DPA are capable of expressing *sulf1* by 14 DPA and forming Col2⁺ cartilage by 28 DPA, suggesting the wound healing/inflammatory environment of the tail immediately after tail amputation through 14 DPA is playing a role in the induction of fibroblast *sulf1* expression and subsequent chondrogenic potential. This contrasts with the figure mentioned above, now revised Figure 5, in which limb fibroblasts are transplanted into already formed blastema recipients at 14 DPA, resulting in total lack of transplanted limb fibroblast cartilage contribution. Taken together, it does not appear that limb fibroblasts express *sulf1* at a different time in the healing process compared to tail, since no *sulf1* expression is exhibited in native limb fibroblasts at 7 or 14 DPA, regardless of Hh stimulation. Limb fibroblasts can become *sulf1*⁺, but only when exposed to the tail environment immediately after amputation, whereupon limb fibroblasts mirror native tail fibroblast expression patterns and chondrogenic capacities. This suggests that transplantation into any point of the blastema stage would be too late to activate limb fibroblasts and that the critical environment/factors needed to activate limb fibroblast *sulf1* expression happen prior to the blastema stage, in the inflammatory stage.

11. Overall, the resolution of the images was low, making it difficult to interpret much of the ISH clearly - although this is more reflective of the submission portal than the images themselves.

We apologize for this inconvenience. Upon resubmission, we have asked the editors to provide figures at higher quality if requested.

Analytical approach

12. There is a lack of description of their FISH protocol and related imaging modalities and processing. More description of their methods for quantification in FIJI would make this more replicable.

We have revised the methods section to include more detailed protocols for FISH and related imaging and quantification methodologies to improve reproducibility.

13. Include the meaning of the hashtag in Fig. 5.

We apologize for this oversight. We have reevaluated the statistical analyses performed for this figure and adjusted the tests performed accordingly in figure legends and the revised methods section. The revised figure legend now includes descriptive text for all symbols utilized in the figure.

14. In Figs. 4, 5, and 7, it's uncertain whether two-way ANOVA or a nonparametric test would be more appropriate. It doesn't appear, visually, that there is equal variance between groups, nor can we assume a normal data distribution.

Thank you for bringing this to our attention. We have reevaluated all statistical analyses performed for this manuscript and adjusted the tests performed accordingly in figure legends and the revised methods section.

Suggested improvements

15. As *pltp* and *sall1* are indicated important blastema markers with regards to single-cell sequencing, a pseudotime trajectory and clustering analysis of these two genes included in Fig. 1 would be useful reference.

Thank you for the insightful suggestion. We have included this subclustering analysis and pseudotime trajectory data in Supplementary Figure 4, inserted below, for reference. *Sall1* and *pltp* subclustering and pseudotime trajectory analysis did not reveal distinct expression patterns correlating pseudotime with specific regeneration stages, and thus, were less of a focus for analysis in this manuscript.

16. Brief exploration with similar clustering methods of if or how macrophages and other immune cells such as septoclasts evolve through regeneration might provide interesting insight.

This is an excellent idea, and we have mentioned this potential work in the revised discussion section. Briefly, this topic will be the subject of a future paper. We are currently using the method of probe sequencing (PMID: 31815670) to resolve specific phagocyte populations through the tail regeneration process. We are also including bone marrow and peripheral blood myeloid progenitor cells in our analysis to identify starting points for macrophage and osteoclast differentiation. We are excited by this work, but we believe these studies are best presented in a separate paper as they are beyond the scope of the current manuscript.

17. Exploring differences in the inflammatory stage and between this and the blastema stage would be useful, especially with regards to the claim of macrophage infiltration before FCTC activation.

Showing for certain that macrophages peak at 7DPA, for example compared to 1 and/or 3DPA, would strengthen this claim.

This is an excellent point. We have included Supplementary Figure 14, inserted below for reference, to depict phagocytic macrophage and osteoclast and immune cell infiltration via FISH at 0, 1, 3, 7, and 14 DPA, spanning the inflammatory and blastema stages. Bar = 500 μ m. *Aoah*⁺ immune cells are present primarily from 1-7 DPA, validating our description of these time points as representative of an inflammatory response. *Ctsb*⁺ macrophages and osteoclasts appear at 1 DPA and remain at some level through the 14 DPA blastema stage, but peak at 7 DPA in the inflammatory stage.

18. Including data for 28DPA samples would support several experiments, such as in Fig. 2 and 5. The final model in Fig. 10 is of 0, 7, 14, and 28DPA, but 28DPA is largely absent. Otherwise, justify why 28DPA does not need to be presented. Is it very similar to 0 DPA or is it just too similar to 21 DPA to include?

Thank you for bringing this discrepancy to our attention. We have removed the 21 DPA samples from Figure 2 and revised Supplementary Figure 5 and replaced this panel with 28 DPA homeostatic regeneration tail samples, as 21 DPA samples often span late-stage cone-shaped blastema and early regenerated homeostasis stage elongating tails, as previously described in our morphological description of each regeneration stage. Since 21 DPA often spans two regeneration phases, we have largely excluded this time point from the remainder of analysis, to avoid conflicting staging information with the corresponding DPA.

In the phagocytic cell analysis, now shown in revised Figure 6, 28 DPA is excluded from analysis because phagocytic cell activity does not appear to be critical to tail regrowth following the blastema stage beyond 14 DPA. We completed a time course of clodronate liposome treatment windows, which selectively induces apoptosis in phagocytic cells, and treatment effect on tail regeneration length, shown in the figure below. A typical timeline of a regenerating tail and average regrowth (**a, top**) is compared to a time course of clodronate liposome treatment injections (**a, bottom**), delivered intraperitoneally, 0.125 mg per gram lizard weight every 48 hours. Red bars indicate periods of clodronate liposome treatment beside corresponding gross tail morphology at 21 DPA. Resulting tail length quantification is presented in cm, $n = 5$ lizards per treatment condition (**b**). Consistent clodronate liposome treatment from 0-21 DPA resulted in extremely limited tail regrowth compared to PBS liposome-treated controls. Treatment spanning 3-9 DPA resulted in similarly limited tail regrowth to 0-21 DPA treatment. Lizards treated with clodronate treatment from 7-13 DPA showed stunted, but moderate tail regrowth, while those treated with clodronate from 14-20 DPA showed no significant difference in regenerated tail length compared to controls. One-way Welch's ANOVA for unequal variances with Dunnett's T3 multiple comparisons tests was used. **, adjusted $p < 0.01$; ns, not significant, compared to PBS control (Dunnett's T3). This indicated 3-9 DPA as the critical window for phagocyte involvement in tail regeneration, while 7-13 DPA appeared to contribute to tail regrowth, but with a less dramatic loss of regenerative capacity in response to phagocyte ablation. Since loss of phagocytes from 14-20 DPA did not appear to impact tail regeneration, assessment of phagocytes beyond 21 DPA in limb and tail amputation samples did not seem applicable to subsequent phagocytic cell experiments.

19. In Fig. 6, it would support the model of a *sulf1*⁺ *spp1* sub-population of FCTCs to include at least a 7DPA timepoint for co-localization. Or can this be deduced by the scRNAseq data?

Thank you for this suggestion. We have added Supplementary Figure 16, inserted below for reference, to support our model of *spp1*⁺ *sulf1*⁻ fibroblasts at 7 DPA (**a**), followed by fibroblasts co-expressing *spp1*

and *sulfl* at 14 DPA in the distal blastema (**b**), analyzed via FISH. Bar = 500 μ m. Please note that previous Figure 6 has been moved to the supplementary material as Supplementary Figure 15, per suggestion from other reviewers' comments.

20. Panel numbering and in-text references of Figs 8. And 9 could be neater.

Thank you for bringing this to our attention. We have revised the in-text references to these figures for simplicity and clarity.

21. With the experiments associated with Fig. 7, it would also make sense to transplant fibroblasts into tails at early regeneration timepoints. It could very well be that 14DPA, when the blastema is already formed, could be far too late to rescue fibroblast activation. Otherwise, adding this as a caveat to the study is sufficient.

This is an excellent point, and we now include these experiments in the revised manuscript. Supplementary Figure 11, inserted below for reference, shows isolated and labeled limb and tail fibroblasts transplanted into 0 DPA tails (**a**) can rescue *sulfl* expression by 14 DPA (**b-e, j**) and Col2 expression and cartilage formation by 28 DPA in both transplanted populations (**f-i, k**). Green arrowheads denote DiO⁺ cells and red arrowheads mark DiI⁺ cells. Bar = 50 μ m. (**k**) Unpaired two-way *t*-tests with Welch's comparison for unequal variances were used. ns, not significant. This confirms that the blastema itself does not induce fibroblast expression of *sulfl*, but instead earlier tail environmental factors prior to 14 DPA are responsible for supporting *sulfl* gene expression. Later experiments with phagocytic cell-conditioned media bead implantation suggest macrophage- and septoclast-like cell-secreted factors and signals are critical for supporting *spp1* and *sulfl* expression in these fibroblast populations.

22. In several instances, seemingly important validation and findings are relegated to supplementary figures:

- o All figure references in the results section “Lizard blastema FTCs undergo chondrogenesis in response to Hedgehog signaling” are to supplementary figures. If this is an important crux to their argument, they should be made main figures.
- o All references to limb regeneration, though importance is placed on comparing and contrasting with tail regeneration, are also placed in supplementary materials, but appear to be important for overall significance.

Thank you for bringing this to our attention. We have rearranged the aforementioned supplementary figures to be included within the main figures of the manuscript and ensured that each subsection of our revised results references at least one main figure each.

Clarity and context

23. In Fig. 2 and related results, *mdk*, *tenascin*, etc., though in supplementary materials, are discussed before indicating why the genes in the figure are shown. It would be useful to provide context with

regards to “condensing chondrogenic mesenchyme” as a justification for the genes they chose to show in the main figure.

Thank you for this suggestion. Our scRNAseq analysis helped us to identify these markers for resting, injury-state, and blastema fibroblasts as they progress through regeneration into cartilage differentiation. Although many marker genes we discussed have significance in condensing mesenchyme, bone, and cartilage development, they were chosen for further investigation primarily due to their high levels of differential gene expression in the fibroblast cluster of our sequencing analysis. *Cdh11*, *col3a1*, *coll2a1*, *mdk*, *pltp*, *sall1*, *sox9*, *spp1*, *sparc*, *sulfl*, and *tnl* were chosen for further interrogation due to their high expression in our scRNAseq fibroblast cluster and were validated via ISH to show specificity for fibroblastic connective tissue cells (FCTCs), shown in Figure 2 and revised Supplementary Figure 5. Follow up analysis was performed on genes that were specifically activated in FCTCs beginning at the blastema stage, *sulfl*, *pltp* and *sall1*. We have clarified this in the relevant results section and have included background context and references related to these validated genes.

24. For those not acquainted with anole regeneration, more context would be useful on the use of the second lizard species to support experimental validity – given that this research heavily involves immune system recruitment, it seems important given the point about their not needing immunosuppressants.

Thank you for this suggestion. We have included additional context and references for our use of mourning geckos (*Lepidodactylus lugubris*) in transplantation experiments in the revised results and methods sections, including validation of phagocyte ablation via clodronate treatment in this species compared to previously validated green anole (*Anolis carolinensis*), shown in revised Supplementary Figure 17.

25. Some areas of the discussion feel better suited for the introduction, such as the composition of the regenerated tail when compared to salamanders.

We appreciate this suggestion. We have significantly revised the discussion of the paper and have relocated recommended information to the revised introduction section.

26. The introduction mentions zebrafish regeneration in reference to macrophages, but otherwise does not discuss their utility in consideration of other models.

Thank you for bringing this to our attention. We have added additional references and information about macrophages in regeneration of other model organisms, such as the African spiny mouse, as well as background information on macrophage and phagocytic cells in lizards during tail regeneration.

27. The discussion of the Gerber/Tanaka axolotl scRNA dataset may be more effective when discussing the results shown in Fig. 1.

We appreciate this suggestion. We have revised the results section to make this comparison and removed much of this information from the revised discussion.

28. The information given about *sulfl* in the discussion may be much more effective in the intro or relevant results sections, as otherwise there is little reason given for their focus on this gene.

Thank you for this recommendation. We have included information about *sulfl* and *spp1* in the revised results sections as suggested and edited the discussion accordingly.

References

29. The claim that lizard blastema cells do not become more embryonic-like, as salamander blastema cells do, would be best supported by a reference to previously published embryonic lizard datasets or studies on lizard de-differentiation potential, such as Eckalbar et al., 2011. Also, has dedifferentiation of tail cells been shown in axolotls? I can only think of examples in limb regeneration.

In response to other reviewers' suggestions and comments, we have revised the discussion to put less emphasis on direct comparison between salamander and lizard blastema cells. Rather, these comparisons will be addressed in future, more focused studies.

30. It may be useful to reference deer antler regeneration, an example of mammalian organ regeneration that is not established to be blastema-dependent (Li, Zhao, Liu & McMahon, 2014).

Thank you for highlighting this distinction and the suggested references. We have added information related to non-epimorphic regeneration in mammals to the revised introduction section.

31. In the sentence “ Fibroblastic connective tissue cells (FCTCs) have been previously identified as the most abundant cell type, found in the blastema of regenerating axolotl limbs”, a reference to Muneoka et al, 1986 is needed.

Thank you for bringing this oversight to our attention. We have added this reference to the revised discussion section.

Reviewer #3 (Remarks to the Author):

Major:

1. In Supp Fig 1A, the authors used DEG analysis and ISH to determine potential cell types. Although this approach could be valid, to this reviewer, these marker genes are not necessarily associated with proposed cell types. A more common approach of indicating established markers would be helpful (e.g. Pax7 – muscle stem cell)

Thank you for this suggestion. We did attempt to align DEG analysis with established markers in literature when available for cluster identification. However, due to lack of sufficient gene marker data in reptiles, known and unknown discrepancies between lizard and mammalian cell marker genes, and lack of comprehensive annotation in the *Anolis carolinensis* genome, we chose to combine ISH validation with known marker gene expression identification from literature. For example, *pax7* is not currently annotated in the green anole genome, so we utilized *ckm* for muscle-related cell cluster identification, a known marker expressed in reptile muscle (PMID: 29164111). We also are aware of several cell populations with divergent marker expression between lizards and mammals. For example, mammalian cells with high expression of *Cd8a* and *Cd8b* could be identified as cytotoxic T cells, however, green anole lizards have shown high levels of *cd8b* expression within B cell populations and lack *cd8a* and *cd8b* expression in T cells (PMID: 32337387), thus cell clusters characterized by high *cd8a* and *cd8b* expression within the presented scRNAseq data set were purposely unnamed.

For clarity, we have listed marker genes used to identify each cluster in the revised results section and explained instances where alternative marker identification strategies were employed as opposed to established marker identification in literature.

2. Authors use scrnseq to estimate cell type proportions. Although some groups may use such an approach, especially with in vivo complex tissues, this analysis alone cannot account for proportions as their samples vary in size and composition, and dissociation cannot be considered the same for

all these populations. On this point, the authors' data already have problems. In figure 1B the authors show that FCTCs are more abundant in the blastema stage, but in Figure 2 all the FCTC markers are more highly expressed in the regeneration stage compared to the blastema stage (I might be wrong here because the figure quality is hard to see).

Thank you for pointing out this shortcoming. To validate Figure 1B, we have added Supplementary Figure 2, inserted below for reference, in which validated cell marker genes for major clusters were assessed via fluorescent *in situ* hybridization (FISH) quantification at each regeneration stage to determine the respective percentage area each cell type contributed to the regenerated portion of the tail. *Col3a1*⁺ FCTCs percentage contribution peaks at the blastema stage at 14 DPA, confirming results in Figure 1B, and this expression level is maintained through regenerated homeostasis at 28 DPA. While marker expression of FCTCs 28 DPA appears to be greater in Figure 2 and revised Supplementary Figure 5 because the regenerated portions of tails are longer at this stage, there appears to be no significant difference of overall proportional contribution of FCTCs to total regenerated tail area between the blastema and regenerated homeostasis stages. One-way Welch's ANOVA with Dunnett's T3 multiple comparisons tests were used per cell type across regeneration stages. *, adjusted $p < 0.05$; **, adjusted $p < 0.01$; ***, adjusted $p < 0.001$; ns, not significant, compared to respective cluster Day 0, unless otherwise noted (Dunnett's T3 multiple comparisons tests).

Additionally, we have revised the results and discussion sections related to FCTC marker expression to emphasize the sequential addition of FCTC marker gene expression over the course of regeneration, where several FCTC genes, such as *col3a1*, *sall1*, *pltp*, *spp1*, and *sulfl*, are induced at specific phases of tail regeneration and remain highly expressed in fibroblasts throughout the remainders of the regeneration process. This model explains why FCTC gene expression is maintained through the regenerated homeostasis stage and represents an important distinction between original and regenerated tail fibroblast tissues. Supplementary Table 1 has been added to clarify FCTC expression patterns throughout each tail regeneration stage.

3. Fig 1C: chondrocyte populations look separated from each other. Is there any explanation for this phenotype?

The regenerating cartilage tube is inherently heterogeneous, thus some scattering of chondrocytes into FCTC clusters may reflect different subtypes of the chondrocyte populations, such as perichondrium, hypertrophic chondrocytes, calcified cartilage, callus cartilage derived from the terminal vertebrae and blastema-derived hyaline-like cartilage in the distal tail. These subpopulations of cartilage cells have differing gene expression profiles that may be more or less similar to FCTCs depending on developmental stage and tail position, potentially explaining this phenotype in subclustering chondrocytes (See PMIDs 25596336 and 27387871 for additional information on cartilage diversity within the regenerating tail).

- 4. Fig 1D: it is tough to see where are the Day 0 samples. Can the authors show separate mapping for each stage? Otherwise, it is hard to appreciate how much similarity is present between the original tail and the “homeostasis” sample.**

We have added Supplementary Figure 3, inserted below for reference, depicting separate plots mapping each regeneration stage over the course of pseudotime trajectory and corresponding subcluster TSNE plots.

- 5. Authors made many conclusions based on pseudo timing analysis (related to fig 1 and 2): these include the hierarchy of activation, fibroblast activation, resting cells, and differentiation trajectory. However, pseudotiming alone is not sufficient for these claims. The pseudotiming approach can be “suggestive” but not definitive and similar analyses have created problems before (please see e.g. PMID: 31974159). With current data, it is impossible to distinguish if the existing fibroblasts expand or if new cells gain these gene expressions. For this, authors need to use a lineage tracing approach combined with staining or other means. Authors could consider applying the *prrx1* enhancer reporter as it is reported to be active in salamanders to mammals, or some different approach. Overall, conclusions based on scRNA-seq reach far beyond their capacities. Authors should either soften their conclusion as “suggest” with a note on limitations, or perform lineage tracing to support their conclusion. For example, “FCTC activation” and similar statements should be changed throughout the text.**

Thank you for bringing these issues to our attention and for the supporting literature suggestion. Unfortunately, the current state of lizard transgenesis precludes lineage tracing due to challenges presented by the biology of reptile reproduction. Further, *prrx1* has not been annotated in lizard genomes, barring us from utilizing its FCTC specificity for analysis. As a result, we utilized our clonal population of *Lepidodactylus lugubris*, isolated and labeled specific fibroblast populations, then transplanted them into genetically identical geckos, allowing us to fluorescently trace those cell populations *in vivo*.

We have previously shown that blastema cells are capable of differentiating into chondrocytes (PMID: 35225965), thus, the intent of pseudotime analysis was to investigate whether a specific subpopulation of FCTC could exclusively differentiate into cartilage, and if so, which genes could be involved in that cell fate change. We agree that claims of hierarchy and activation when discussing pseudotime analysis alone are premature, and we have revised the results accordingly to soften these claims before introducing subsequent experimental evidence. We have also revised the terminology used when referring to FCTCs to avoid ambiguity regarding “activated” and “resting” cell states. Instead, we have renamed our FCTC groups based on tail location to distinguish differential FCTC gene expression over the course of regeneration. We have also revised the discussion section to highlight limitations of pseudotime analysis.

- 6. Authors state “Furthermore, SAG treatment induced chondrogenesis in blastema regions that, under native conditions, would have differentiated into other tissue types, such as muscle and fat. These results suggested that Hh signaling is necessary and sufficient for inducing chondrogenesis in blastema FCTCs and that a large portion of blastema cells are capable of cartilage differentiation.”. There are several suggestions/conclusions not supported by the data related to these statements. Firstly, what is the evidence for the muscle and fat statement – no figure or literature has been referred. Likewise, what is the evidence that a large portion of blastema cells are capable of cartilage differentiation? For such a claim, authors need to analyze individual fibroblasts and their cartilage differentiation. These changes would influence the rest of the manuscript.**

Thank you for pointing out this oversight. We have removed the above claims regarding other potential cell lineage fates from the revised results section, given the lack of supporting experimental evidence. We have also reserved comments related to fibroblast cartilage differentiation potential for later in the results, when transplantation experiments and FCTC chondrogenic potential are assessed.

- 7. Related to SFig4: authors state SAG treatment did not induce chondrogenesis in the limb. Is there evidence showing cyclopamine or SAG treatment worked in the limb? Maybe limbs require a higher dosage to see an effect?**

This is an excellent point. To validate the efficacy of SAG and cyclopamine treatments in limb and tail, we analyzed samples via ISH for *gli1*, a downstream reporter of the Hedgehog (Hh) signaling pathway

and an established readout of Hh pathway activation (PMID: 10433919). As shown in revised Supplementary Figure 7, inserted below for reference, tail samples express *gli1* adjacent to the ependymal tube, the Hh signaling center in the native tail (**c**). *Gli1* is not detected in cyclopamine-treated tails (**i**) and is highly expressed in SAG-treated tails (**o**). Amputated limb samples do not express *gli1* under control conditions and cyclopamine treatment (**f**, **l**), as expected, considering the lack of Hh signaling center in the limb and inhibitor treatment, respectively. SAG treatment resulted in high levels of *gli1* expression in the limb (**r**), indicating sufficient Hh signaling activation in the limb to induce chondrogenesis under the proper conditions. n = 5 lizards per treatment condition; representative images shown. tb, tibia; ve, vertebra. Bar = 500 μ m. This is further confirmed in conditioned media (CM) experiments in which septoclast-like cell-CM induces ectopic cartilage formation in SAG-treated lizard limbs (Figure 9) but does not form cartilage in control limbs, lacking Hh stimulation, receiving the same CM treatment (Supplementary Figure 22).

8. SFig 5: The authors used limbs as a generic healing control. Limbs and tails differ in their composition and proportion of cell types (limbs do not have spinal cord etc.), and one cannot be used to infer a healing mechanism for the other. If the authors wish to test the nonspecific healing response, they could damage the tail without amputating and check their phenotype.

Several classical studies in reptile regeneration have highlighted the regenerative capacity of limbs mimicking that of tails following spinal cord/ependyma transplantation. These studies indicate that, when given the proper signaling from a regenerating spinal cord, a limb amputation can also result in a comparable, tail-like regenerative response (PMIDs: 5712416, 5426257, 737883, 4415899). We have performed similar experiments, summarized in the figure below. Briefly, regenerated spinal cord (RSC) pieces were isolated from donor gecko (*L. lugubris*) blastemas 14 DPA and subsequently implanted into freshly amputated limbs (0 DPA) of recipient geckos (**a**). Limbs were analyzed 28 days post transplantation (DPT) and analyzed via gross morphology, histology and modified Russell-Movat pentachrome staining. Compared to control, untreated limbs that exhibited fibrotic tissue deposition and scarring (**b, d, f**), amputated limbs receiving ectopic RSC implants (RSCI) formed blastema cones around RSCIs and regenerated tail-like appendage structures (**c, e, g**). Given the previous studies and our personal observations, we concluded the limb possesses the necessary composition and cell types to induce a comparable appendage regenerative response but lacks natural anti-fibrotic and pro-regenerative signals. When supplied with the proper signaling in the form of RSCIs or combinations of exogenous Hedgehog signaling via SAG injections and phagocytic cell-conditioned media, as shown in Figure 9, lizard limbs resist scar formation and establish blastemas. We have also considered non-amputation tail injuries performed using biopsy punches, but these injury models were difficult to relate to tail amputations. For example, tail biopsy punch protocols did not result in bone and bone marrow cavity injuries that are characteristic of limb and tail amputations. Thus, amputated lizard limbs provide the most direct healing controls for our tail amputation experiments.

9. In figure 3 and SFig5, authors conclude how certain gene expressions changed. However, these data are based on ISH results, which is not a quantitative approach. Moreover, these phenotypes' morphologies and cell numbers are changing, and we only see a section here. In figure 3i the authors claimed that “SAG treatment resulted in increased and expanded *sulf1*+ blastema areas” however, in the figure it doesn't look exactly the same as in the control animal in figure 2r. Was there any statistical analysis performed? If not, it's wrong to claim that *sulf1*+ increased. I am not convinced about the conclusions related to these figures. Quantitative methods (e.g. qRT-PCR) with adequate normalization methods and replicates are necessary. These also influence Sfig6 and the rest of the manuscript conclusion.

Thank you for bringing up this point. We have added Supplementary Figure 8, inserted below for reference, to validate these claims using real-time polymerase chain reaction (RT-PCR). We isolated total RNA from blastema stage lizard (*A. carolinensis*) tail samples 14 DPA treated with Hh inhibitor cyclopamine, Hh agonist SAG, or PBS vehicle control and performed RT-PCR to quantify relative gene expression between treatments. $n = 5$ tails per drug treatment condition assessed across 5 RT-PCR runs. Unpaired two-way *t*-tests with Welch's correction for unequal variances were used to compare to vehicle control for each gene. *, $p < 0.05$; ***, $p < 0.001$; ****, $p < 0.0001$. *Sulf1* expression was significantly upregulated in response to SAG treatment and significantly downregulated in response to cyclopamine Hh inhibition compared to controls. *Spp1* and *pltp* showed no significant differences in relative gene expression levels between Hh signaling modulation treatments and controls, while *sall1* showed a mild increase in expression due to SAG treatment ($p = 0.03667$) but was unaffected by cyclopamine treatment. These results support our limb and tail staining results, consolidated now in revised Figure 3.

10. Related to Fig 4 and Supp 7: The authors do not isolate pure FCTCs or fibroblasts, but their method is very crude tissue removal and grafting. This is one of the central problems related to grafting-based conclusions. None of these experiments have only FCTCs, and potentially they also include other cell types, which would confound the findings and conclusions. If the authors believe they only have FCTC, they can provide additional data, by doing scRNA-seq to isolated populations, staining, or any other means. Still, it is doubtful that they have a pure population.

To validate fibroblast isolation methods, we have added Supplementary Figure 9 to the manuscript, inserted below for reference, to depict sample purity. We isolated fibroblasts from *L. lugubris* donor limb and tail tissues using enzymatic and physical cell digestion protocols and enriched for fibroblasts using MACS[®] beads, as detailed in the revised methods section. Briefly, incubated digested limb and tail cells were incubated with anti-CD144 (VE-Cadherin), anti-CD45, anti-CD326 (EpCAM) and anti-integrin α -7 magnetic MACS[®] beads, and bead/cell solutions were passed through LD columns on MidiMACS[™] Separator magnets. This allowed us to specifically deplete immune, endothelial and muscle cells from our digestion pools and enrich our samples for fibroblasts. Fibroblast enrichment was validated by analyzing fibroblast preparations before and after MACS[®] depletion via FISH for fibroblast marker *col3a1*, muscle marker *ckm*, endothelial cell marker *vwf* and immune cell marker *aoah* (a-f). FISH analysis of post-MACS[®] depletion revealed purified fibroblast populations, almost entirely devoid of endothelial, muscle and immune cell populations (d-f). Bar = 25 μ m. We quantified these images (g), n = 50 images quantified, 10 images each from 5 total cells digestion pools, revealing significantly purified FCTC preparations following MACS[®] bead treatments and significant reduction in all contaminating cell types assessed. Paired two-way *t*-tests were used between respective cell types. ****, *p* < 0.0001.

11. In Fig 5 and SFig9, the authors compare specific immune system-related cell abundances for tails and limbs. Is it possible that these populations show different markers for these two models? Hence the limb may also contain these populations but lack the suggested markers.

In Figure 9, we show that implantation of septoclast-like cell (herein, septoclasts) conditioned media-soaked beads (CM) into limb amputations is sufficient for enabling *sulf1* expression in native limb fibroblast with the addition of exogenous Hh signaling via SAG injections. If these critical phagocytic septoclasts were present in the limb, but with different marker expression, a regenerative response and

similar biological outcome (Hh-responsive fibroblasts activation of *sulfl* expression and subsequent chondrogenesis) would be expected following limb amputation and SAG injection alone. Given septoclast-CM is necessary to rescue fibroblast *sulfl* expression and ectopic cartilage formation in amputated limbs in addition to SAG, we can conclude no equivalent septoclast population exists in the limb, regardless of marker gene expression.

12. Fig 7: was it previously established clodronate works in *L. lugubris*? If so, please add the reference, or related data.

We have previously validated clodronate liposome treatment in *A. carolinensis* and have included this reference in the revised results section (PMID: 32337387). Further, we have added Supplementary Figure 17, inserted below for reference, to depict the validation of phagocytic cell depletion via clodronate liposome treatment in *L. lugubris* (**e-h**) compared to previously validated *A. carolinensis* (**a-d**) and related quantification comparisons of regenerated tail length (**i**) and phagocytic cell density (**j**) between clodronate liposome treatment and PBS liposome controls in each species. Both species form typical elongated tails at 21 DPA under control conditions and have significantly reduced tail length/regeneration as a result of clodronate liposome treatment. Phagocytes were labeled with DiI, following phagocytosis of injected Fluoroliposome® DiI. Control PBS liposome-treated tails of both species exhibited DiI-labeled phagocytic cells (**c, g**), while clodronate treatment significantly depleted phagocyte cell densities (**d, h, j**). $n = 10$ lizards quantified per treatment condition per species. (**a-h**) Bar = 1 mm. (**i, j**) Unpaired two-way *t*-tests with Welch's correction for unequal variances were used. ***, $p < 0.0001$.

13. Fig 7: In figure 7 the authors transplanted fibroblasts from clodronate or PBS-treated animals. However, they don't show the phenotype of these fibroblasts. Clodronate can be very toxic and induce cellular accumulation and myofibroblast activation. Without knowing the phenotype of fibroblast, you can't really compare it with that collected from PBS-treated animals. Perhaps a better experiment here will be to collect fibroblasts from regenerating animals and from the intact animals without amputation and transplant them into a recipient animal. So, experiments seem to need more controls. Do the authors see the same phenotype when they do not mix PBS fibroblast and clodronate fibroblasts? Current results do not omit the possibility that their phenotype is affected by the mixing populations. Also, the recipient lizards in transplantation studies were all treated with SAG, no control animal recipients?

Thank you for bringing up this point. To address these comments and better define the fibroblast phenotype, we have included a new experiment involving fibroblasts collected from regenerating animals and original tails that are then transplanted into recipient animals. Results from these experiments are shown in revised Figure 4 and inserted below for reference. Fibroblasts isolated from original tails (0 DPA), tail blastemas (14 DPA) and regenerated tails (28 DPA) were separately labeled with DiI and injected into blastemas of recipient lizards. Transplanted fibroblasts were assessed 14 days post transplantation (DPT)/28 DPA for chondrogenic potential via Col2 IF **(a)**. Original tail fibroblasts **(b-d)** and regenerated homeostatic tail fibroblasts **(h-j)** did not form cartilage following transplantation and did not co-express Col2. Only transplanted blastema fibroblasts expressed cartilage marker Col2 **(e-g)**, with significantly higher proportions of transplanted blastema fibroblasts expressing Col2 compared to original and regenerated tail fibroblasts. Green arrowheads denote DiI-labeled fibroblasts. Bar = 50 μ m. **(k)**. n = 50 cell counts measured from 5 images among 10 different animals/tail samples for each condition. One-way Welch's ANOVA for unequal variances and Dunnett's T3 multiple comparisons tests was used. ****, adjusted $p < 0.0001$ (Dunnett's T3). These results indicated some fundamental difference between blastema fibroblasts and homeostatic fibroblasts that specifically allow blastema fibroblasts to form cartilage in response to Hh signaling. These results also demonstrated the phenotype of unmixed, untreated tail fibroblasts in transplantation experiments.

We also added Supplementary Figures 12 and 18 to provide controls for revised Figures 5 and 7, respectively. As a control for limb and tail fibroblast transplants in revised Figure 5, Supplementary Figure 12, shown below, assesses Dil-labeled tail and limb fibroblasts incorporation into recipient tails separately in both PBS control- and SAG-injected lizards, to ensure phenotypes observed in revised Figure 5 are not simply the result of mixing fibroblast populations. Tail fibroblasts injected into 14 DPA recipient blastemas without exogenous Hh signaling do not incorporate into Col2⁺ cartilage (**a-c, m**), while significantly more tail fibroblasts express Col2 and form cartilage in recipient blastema treated with SAG (**d-f, m**). Limb fibroblasts transplanted into control (**g-i**) or SAG-treated (**j-l**) recipient blastema both showed virtually no Col2 expression and cartilage formation. Dil-labeled fibroblasts. Bar = 50 μ m. Significantly more tail fibroblasts formed cartilage in response to Hh signaling in SAG-treated recipient

lizards and expressed Col2 than limb fibroblasts under similar conditions. **(m)** $n = 50$ cell counts measured from 5 images among 10 different animals/tails for each condition. Two-way ANOVA with pairwise Tukey's adjustment for multiple comparisons was used. ****, adjusted $p < 0.0001$; ns, not significant (Tukey's).

As a control for PBS liposome- and clodronate liposome-treated tail fibroblast transplants in Figure 7, Supplementary Figure 18, inserted below, assesses DiI-labeled fibroblast incorporation into recipient tails separately in both PBS control and SAG-injected recipient lizards, to ensure phenotypes observed in Figure 7 are not simply the result of mixing fibroblast populations or liposome treatments. Similar results are observed compared to Supplementary Figure 12 above, showing PBS liposome-treated fibroblasts injected into 14 DPA recipient blastema without exogenous Hh signaling did not incorporate into Col2⁺ cartilage **(a-c, m)**, while significantly more PBS liposome control-transplanted fibroblasts expressed Col2 and formed cartilage in recipient blastema treated with SAG **(d-f, m)**. Clodronate liposome-treated fibroblasts transplanted into control **(g-i)** or SAG-treated **(j-l)** recipient blastema both showed virtually no Col2 expression and cartilage formation. Green arrowheads denote DiI⁺ fibroblasts. Bar = 50 μm. Significantly more PBS liposome-transplanted fibroblasts formed cartilage in response to Hh signaling in SAG-treated recipient lizards and expressed Col2 than clodronate liposome-treated fibroblasts under similar conditions **(m)**. $n = 50$ cell counts measured from 5 images among 10 different animals/tails for each condition. Two-way ANOVA with pairwise Tukey's adjustment for multiple comparisons was used. ****, adjusted $p < 0.0001$; ns, not significant (Tukey's).

Further, no significant difference was observed in Col2⁺ DiI-labeled cell percentages 28 DPT between PBS liposome-treated fibroblasts and untreated tail fibroblasts transplantations into PBS control- or SAG-treated blastema tail (14 DPA) recipients, shown below, indicating liposome treatment did not confound transplantation study analyses. Unpaired two-way *t*-tests with Welch's correction for unequal variances were used. ns, not significant.

Potential clodronate toxicity does not seem to be altering fibroblast phenotypes. Isolated fibroblasts were not affected by clodronate liposomes phenotypes *in vitro*. Furthermore, as evidenced in Figure 8, clodronate liposomes did not interfere with fibroblast biology *in vivo*. In this study, amputated tails of lizards treated with clodronate liposomes were implanted with beads soaked in conditioned media from macrophage and/or septoclast cultures. Native fibroblasts were capable of activating *spp1* expression in response to macrophage-CM, and *spp1* and *sulfl* expression was rescued by a combination of macrophage- and septoclast-CM treatment, indicating fibroblasts treated with clodronate liposomes were still capable of gaining FCTC sequential marker expression of *spp1* and *sulfl*. Thus, clodronate liposome treatment did not interfere with rescue of fibroblast expression patterns and chondrogenic programming in regeneration.

- 14. Fig 8: The authors perform conditioned media experiments. This is a very exciting experiment. However, there is no control media for the same purpose, and observed effects might be due to the media's synergistic effect and the cells cultured inside. Thus, I cannot comment on the findings from this part.**

We apologize for this oversight. We have included Supplementary Figure 21, inserted below for reference. This new figure provides unconditioned media controls for conditioned media (CM) experiments in clodronate liposome-treated lizard tails (Figure 8) and untreated limbs (Figure 9). Beads were soaked in unconditioned lizard phagocytic cell culture media and implanted into clodronate liposome-treated lizard tails immediately following amputation (0 DPA). These recipients were also treated with PBS vehicle control or Hh agonist SAG treatment and assessed for FCTC marker gene expression via ISH (**a-d**). Unconditioned media treatments did not result in *spp1* or *sulfl* expression in both SAG-treated and PBS control-injected recipient animals by 14 DPA. Similarly prepared beads were implanted into lizard limbs 7 DPA in both PBS- and SAG-treated recipients. Limbs were assessed 28 DPA via ISH for FCTC and chondrogenic markers (**e-j**). Limb samples did not express *sulfl*, *spp1* or *sox9* in response to unconditioned media treatments in both control and SAG-treated lizards. n = 8 lizards/samples per treatment condition. *, location of implanted bead; tb, tibia; ve, vertebra. Bar = 50 μ m. With these controls, Figures 8 and 9 results indicate CM treatments directly resulted in observed *spp1* and *sulfl* expression in clodronate treated tails, and *spp1*, *sulfl* and *sox9* expression, as well as ectopic cartilage formation, in CM-treated limbs.

15. Authors claim to isolate macrophages and septoclast-like cells. How pure these populations are needed to be clarified.

We have revised our macrophage and septoclast validation figure, Supplementary Figure 19, inserted below for reference, and our supplementary methods, to reflect the assessment of respective cell population purity. In addition to pHrodo green bioparticle phagocytic cell uptake and representative staining FISH/IF of isolated macrophage and septoclast cell populations, we have added flow cytometry plots and quantification of isolated cell populations, $n = 5$ isolated cell pools per derivation method, as evidence to the purity of each isolated phagocytic cell population. Bone marrow-derived and blood vessel-derived phagocytes were treated with antibodies anti-Ctsb-Alexa Fluor™ 647 and anti-Ctsk-Alexa Fluor™ 488 and analyzed via flow cytometry (**k-p**), revealing $ctsb^+ ctsk^-$ macrophages represented an average of 94.5% of all cells within phagocytic bone marrow isolation samples, and more than 98% of blood vessel-derived cell pools were $ctsb^- ctsk^+$ septoclasts. (**m, p**) Cell populations within quadrant are indicated as percentages. One-way Welch's ANOVA for unequal variances with Dunnett's T3 multiple comparisons tests were used. ****, adjusted $p < 0.0001$, compared to all other quadrants (Dunnett's T3).

16. Fig 10. Model fails in describing the phenomenon happening. Arrows represent direct interactions between populations, but these connections still need to be established in this study, as discussed above, mainly due to a lack of cell type-specific manipulations and experimentation.

Through the addition of several experiments and supplementary control data, we now believe we have provided sufficient evidence to support the proposed model in Figure 10. We identified *col3a1* from scRNAseq data and ISH/histology validation as a pan-fibroblast marker gene that is expressed by homeostatic FCTCs and is maintained throughout the regeneration process (Figs. 1 & 2, Supplementary Figs 1. & 2). *Spp1* expression appeared in the inflammatory stage FCTCs 7 DPA and was maintained by fibroblasts throughout tail regeneration (Figs 1 & 2), but its expression was observed in both limb and tail amputations and was not responsive to Hh signal modulations (Fig. 3, Supplementary Figs. 7 & 8). We identified and validated *sulfl* as a FCTC marker gene that begins expression at the blastema stage 14 DPA (Figs. 1 & 2), is specific to tail blastema fibroblasts, but not amputated limb fibroblasts, (Fig. 3) and is responsive to modulations in Hh signaling (Fig. 3, Supplementary Figs. 7 & 8), the critical signaling pathway for the induction of cartilage formation in the regenerating tail (Supplementary Fig. 6). We specifically isolated and enriched for fibroblasts (Supplementary Fig. 9) in genetically identical *L. lugubris* geckos from original tail samples 0 DPA, blastema tails 14 DPA, regenerated tails 28 DPA, limb samples 14 DPA, and clodronate-treated tail samples lacking phagocytic cells 14 DPA. Transplantation of these different fibroblast populations into recipient geckos revealed blastema tail fibroblasts specifically expressed *sulfl* and were able to form cartilage in response to Hh signaling (Fig. 4), while homeostatic original and regenerated tail fibroblasts (Fig. 4), limb fibroblasts (Fig. 5, Supplementary Figs. 11 & 12) and clodronate liposome-treated fibroblasts (Fig. 7, Supplementary Fig. 18) did not.

Phagocytic *ctsb⁺ ctsk⁻* macrophages and *ctsb⁺ ctsk⁺* osteoclasts were identified in both tail and limb following amputation (Fig. 6, Supplementary Fig. 13), with phagocytic macrophage and osteoclast populations peaking at 7 DPA (Fig. 6, Supplementary Fig. 14). *Ctsb⁻ ctsk⁺ col4a1⁺* phagocytic septoclasts were specific to amputation sites in tails, but not limbs, (Fig. 6) and were distinctively localized to *sulfl⁺* areas of the distal blastema at 14 DPA (Supplementary Fig. 15). Clodronate liposome treatment depleted phagocytic cell population in both *A. carolinensis* and *L. lugubris* (Supplementary Figs. 15 & 17).

Isolation and purification of macrophages from bone marrow and septoclasts from tail blood vessel (Supplementary Fig. 19) allowed for cell-type specific conditioned media (CM) experimental treatments to assess the impact of secreted macrophage and septoclast cell factors and signals on FCTCs, regeneration and chondrogenic potential. Macrophage-CM alone and co-treated with septoclast-CM rescued *spp1* expression in tail fibroblasts of recipient lizards without native phagocytic cell populations (Fig. 8, Supplementary Fig. 20). Co-treatment of macrophage- and septoclast-CM also rescued *sulfl* expression in clodronate- and SAG-treated tail fibroblasts and induced an accumulation of cells similar to a typical tail blastema. However, separate macrophage- or septoclast-CM treatments alone were insufficient for *sulfl* expression rescue and regenerative response (Fig. 8). Clodronate-treated tail fibroblasts without exogenous Hh signaling were also unable to rescue *sulfl* expression with any macrophage- or septoclast-CM treatment combinations (Supplementary Fig. 20) and unconditioned phagocytic cell media treatments did not result in *spp1* or *sulfl* expression, regardless of Hh signaling status (Supplementary Fig. 21). Thus, factors in macrophage-CM specifically resulted in *spp1* expression in recipient fibroblasts, but macrophage- and septoclast-CM co-treatment and Hh signaling was required to rescue *sulfl* expression. These results suggested a sequential addition of *spp1* expression in fibroblasts induced via macrophages, whose population peaked during the inflammatory stage 7 DPA, followed by subsequent *sulfl* expression proximal to septoclast-CM beads, mimicking *in vivo* septoclast localization to *sulfl*⁺ areas in the blastema stage 14 DPA.

SAG-treated limb fibroblasts received the same phagocytic cell-CM treatments at 7 DPA and were analyzed 28 DPA with native limb macrophages and osteoclasts intact. Exogenous macrophage-CM treatment could not maintain *spp1* or induce blastema FCTC marker *sulfl* or chondrogenic marker *sox9* (Fig. 9). Only the introduction of septoclast-CM, a cell population absent from the native limb, to the Hh stimulated fibroblasts resulted in *spp1*, *sulfl*, and *sox9* expression in FCTCs, indicating the activation of chondrogenic programming, and the formation of ectopic cartilage. Without SAG treatment, septoclast-CM treatment in limb resulted in *spp1*⁺ *sulfl*⁻ fibroblasts (Supplementary Fig. 22), indicating *sulfl* expression is only maintained in response to both Hh signaling AND septoclast cell factors.

Taken together, our evidence supports the following *in vivo* model, shown in revised Fig. 10. Original, homeostatic fibroblasts in both tail and limb express *col3a1*, and lack *spp1*, *sulfl*, and *sox9* expression. Macrophages, present in both limb and tail, induce *spp1* expression in *col3a1*-expressing fibroblasts following injury. *Col3a1* expression is maintained, established as a pan-fibroblast marker in lizards, as *spp1* expression is added to the gene expression profile of injury state fibroblasts 7 DPA. Tails specifically contain septoclast cell populations, while limbs do not, and these factors from these septoclasts induce the addition of *sulfl* to the *col3a1*- and *spp1*-expressing tail fibroblasts expression profile at the blastema stage 14 DPA. Septoclasts are absent in the limb, and thus, *sulfl* expression is not induced in limb fibroblasts, *spp1* expression is not maintained, and eventually fibrotic scars form. *Col3a1*⁺ *spp1*⁺ *sulfl*⁺ blastema stage tail fibroblasts are able to respond to Hh signals from ependymal tubes and begin to express *sox9*, marking the tail fibroblasts initiation of chondrogenic lineage programming.

17. In the discussion, the authors have several misleading statements related to single-cell analysis and comparison to salamanders. These statements show that authors need to familiarize themselves with the single-cell analysis and result interpretation, and they all require significant re-write. For example, the authors state salamander dataset has cycling cells, and theirs do not ?? . It is generally hard to see cycling cells as a separate cluster in heterogeneous datasets. It doesn't mean salamanders have a dedicated cycling cell type? Another example is "Comparing our lizard data sets with published salamander results, both scRNAseq methods captured similar populations of immune cells, epidermal, endothelial, muscle, blood cells, and connective tissue cells in lizards and salamanders. Focusing on the connective tissue cells, both salamander and lizard connective tissue cells expressed the markers ptx3, lum, col3a1, and dpt, and we can conclude that both salamander and lizard blastemas begin with similar cell populations." This statement is based on the annotation of different groups, not transcriptomes. If the authors want to discuss such a possibility, they can

compare their scRNA-seq results to published axolotl datasets and make an unbiased comparison. Otherwise, this statement is just misleading. There are more similar instances throughout the discussion.

We apologize for the lack of clarity regarding our scRNAseq analysis. We have extensively revised the discussion section to eliminate the aforementioned misleading statements considering a direct comparison of lizard and axolotl scRNAseq datasets was not presented in the manuscript, and to clarify the results interpretations regarding presented data.

- 18. Authors state: “unlike salamander FCTCs, lizard blastema cells did not achieve a more embryonic cell state.” What is the data indicating this? For this, authors need to compare their results to the developing tail, if possible via scRNA-seq, and then compare.**

Thank you for bringing this to our attention. We have removed this specific section in the revised discussion and have discussed the value of direct unbiased comparison of lizard and axolotl tail regeneration scRNAseq datasets in future studies.

- 19. Currently, most protocols need to be more suitable for replication. Please provide the catalog number for the materials. Likewise, please provide a more detailed protocol for ISH and IF. E.g., what is the blocking agent during IF, staining duration etc.**

Thank you for bringing this to our attention. We have elaborated on the ISH and IF protocols in the revised methods and supplementary methods sections for ease of replication, as well as elaborated on imaging and quantification modalities. Further, we have included all catalog numbers and supplier information for reagents critical to experimental outcomes in the revised methods, supplementary methods, and Supplementary Table 2, according to journal formatting guidelines.

- 20. Authors wrote n numbers for animals and tails used for analysis, but how many independent experiments were performed is not stated. Are these experiments done only once with these animals? If so, how valid are the statistics choices?**

We apologize for excluding this information. Each experiment was performed a minimum of three times with listed n numbers reflecting the total number of animals used across all experimental replicates. Additionally, the green anole lizards used in conditioned media experiments and histology are field-collected, so all anole experiments represent both biological and technical replicates. We have consulted an expert to discuss the proper statistical tests for each experiment and have updated our analyses accordingly, where applicable.

- 21. In the intro, the authors state this is the first study to perform scRNA-Seq on lizards. This statement is wrong and confusing at the same time. The same group already published a paper with lizard scRNA-seq, which is not cited here (PMID: 32337387). Additionally, another group published lizard scRNA-seq: PMID: 34873160**

We apologize for the ambiguity of this statement. Our intention was to highlight this study as the first scRNAseq dataset to investigate the entire tail regeneration time course in lizards. We have revised this statement for clarity and to conform to journal formatting guidelines regarding claims of “novel” or “first” experiments.

Revised statement: “Here, we present single-cell RNA sequencing performed in the green anole lizard, *Anolis carolinensis*, characterizing regeneration of the tail and an investigation of the role of FCTCs and phagocytes in regeneration and chondrogenesis.”

22. Tails treated with Cyclopamine in figure S3 and figure S4 look completely different. In figure S3 the tail regenerated without bone by 28dpa, but in S4 it did not! Is there any explanation for this? Which one is the phenotype?

Thank you for pointing out this discrepancy. There is slight inherent variability in tail growth between individual animals and lizard species during any appendage regeneration, but following cyclopamine treatment, this variability tends to be more pronounced due to the lack of skeletal structure and rigidity in resulting regenerates. Generally, cyclopamine-treated lizards will regenerate typical tails, except that these tails do not contain a cartilage tube. Additionally, the absence of skeletal elements and reduced rigidity often complicates sample manipulation and analysis, leading to lower quality tail sections and staining, especially in longer tail samples. We have revised Supplementary Figure 7, in the manuscript and inserted below for reference, to display a tail of comparable length and morphology to matched controls in revised Supplementary Figs. 6 and 7.

Minor:

- 1. I assume there was a problem during submission, as the figure quality is terrible and needs an update. Staining results throughout the text is tough to assess in this state.**

We apologize for this inconvenience. Upon resubmission, we have asked the editors to provide figures at higher quality if requested.

- 2. The authors refer to their data collection time points as “original tail (day 0), inflammatory stage, blastema stage, or regenerated homeostasis “. Please state the days instead of using terminologies throughout the main text, as every group has different definitions for these. Indeed, In their previous paper, this group named blastema stage at 7dpa, but here the same time point is referred to as the inflammatory stage and blastema are at 14 dpa. Please explain the difference. This simple change would enhance the readability of the manuscript.**

Thank you for bringing up this discrepancy. As shown in other lizard species (PMID: 21846350), tail regeneration follows a reliable pattern, but the timeline of that pattern can vary significantly between individuals. Regeneration rates may vary due to habitat temperature, photoperiod exposure timing, amputation location on the tail, sex, age, diet, overall lizard health and a variety of other factors (PMID:34940500). There is also inherent variability in tail growth and regeneration between individual lizards, especially given our green anoles are field-collected by suppliers, rather than bred in captivity, so matching sample groups using DPA alone is often not sufficient to assess matching morphological and phenotypical states of tail regrowth. Previous research mentioned above was performed at a different animal facility/university with higher habitat ambient temperature (29-30°C) and thus, the tail regeneration process was accelerated, with green anoles reaching blastema stage by 7 DPA. Our current animal facility housing the lizard species utilized in this study were maintained a lower ambient temperature during the day (24-26°C) and an even lower temperature at night (18.5-21°C), to accommodate diversity of lizard species held in the same habitat, resulting in slower tail regeneration, with anoles reaching blastema stage at 14 DPA.

To ensure consistency among sample groups, in addition to tracking sample DPA, we assessed samples for morphology, with inflammatory stage tails ranging from the closing of the wound epidermis to scab accumulation, blastema stage tails ranging from the loss of the amputation site scab to tail cone shape formation and homeostatic regeneration stage following tail cones through elongation until full growth. This distinction has been made in the revised results and methods section of the manuscript.

- 3. Authors state “mouse digit tip regeneration is lineage-restricted, and mouse blastema cells are unable to differentiate into cartilage.“, This is not completely accurate. Neonatal mouse digit tip amputations induce blastema to differentiate into cartilage, and adult mouse amputations result in differentiation into bones. ADD REFERENCE**

Thank you for bringing this detail to our attention. As our work focuses on appendage regeneration in adult models, we have revised the statement to reflect this distinction.

- 4. Literature referencing is weak. Some statements do not have any reference, and others are under-referenced. I am listing some of these instances from the text, but the authors need to go through the whole manuscript as there are more instances.**
 - a. “Additionally, lizards have more complex and adaptive immune systems, more like that of mammals, contrasting the rudimentary immune systems exhibited in amphibians lacking efficient adaptive immunity. “ Please add references**
 - b. “Specifically, the recruitment of macrophages to injury sites has been shown to be crucial for**

appendage regeneration in several model organisms.“ There are more studies done on this point, please add more comprehensive list or state why some others are omitted.
c. “Among the cells recruited by macrophages during the immune response, fibroblasts have been shown to activate and migrate to areas of tissue loss in a coordinated manner. “ Reference, please.
d. “In axolotl, these blastema FCTCs mimic embryonic limb bud signatures before redifferentiating into the patterned limb skeleton and connective tissues. “There are more studies done on this point, please add more comprehensive list or state why some others are omitted.
e. There is a recent paper on axolotl osteoclasts; please cite PMID: 36218256
f. “Recent interrogations of blastema formation in the neotenic axolotl salamander (*Ambystoma mexicanum*) via single-cell RNA sequencing (scRNAseq) have highlighted the importance of fibroblastic connective tissue cells (FCTCs) in blastema derivation and regenerated tissues.” There are more studies done on this point, dating back to Susan Bryant s initial work – not just recent interrogations, please add a more comprehensive list or state why some others are omitted.

Thank you for highlighting these oversights. We have added the suggested literature reference suggestions and have assessed the revised manuscript for completeness in referencing.

- 5. The authors state in the abstract, “Lizards do not naturally regenerate limbs but are the closest relatives of mammals capable of epimorphic tail regrowth “. Although this is true to our current knowledge, there is no systematic study where all animal tails were cut and tested for regeneration. So, a better statement would be, “Lizards do not naturally regenerate limbs, but they are the closest relatives of mammals identified so far that are capable of epimorphic tail regrowth “**

Thank you for this suggestion. Due to the abstract word limit, we were unable to include your suggestion in its entirety, but we have revised the abstract to reflect this distinction.

Revised statement: “Lizards cannot naturally regenerate limbs but are the closest known relatives of mammals capable of epimorphic tail regrowth.”

- 6. Figure 6 should not be a major figure in the paper, and it should move to supplementary or not shown at all. The authors have already shown that clodronate treatment in lizard tails prevents blastema formation (surprisingly, it was not referenced here, PMID: 32337387), and of course since there is no blastema, there wouldn’t be any blastema-related fibroblasts.**

Thank you for this suggestion, we have moved this figure to the Supplementary Materials and have revised the corresponding results section to emphasize the concentration of macrophages and osteoclasts near the terminal vertebra, distal to *sulfl*⁺ areas of the control blastema, and the position of septoclasts localized in *sulfl*⁺ areas.

- 7. The discussion almost reads like the discussion of another paper, as authors spent a great deal of time comparing lizard and salamander single-cell data sets without showing any figures or methodologies on how these comparisons were made. A discussion revolving around shown data would be necessary.**

We appreciate this feedback. We have significantly revised the discussion section to emphasize manuscript findings in the context of literature, study limitations and future directions.

The revised manuscript and supplementary materials documents are available with tracked changes to reflect all amendments completed during the revision process for reference. Additional changes to the manuscript have been made to adjust for journal formatting guidelines.

REVIEWERS' COMMENTS

Reviewer #1 (Remarks to the Author):

COMMENTS ON THE MS "Single analysis of lizard...fibroblasts" by Vonk et al.

GENERAL REMARKS

The MS has been substantially improved and responded to most requests. It reports a massive and detailed experimental study, well documented from the images provided, that significantly increases present knowledge on this animal model for regenerative studies in relation to the general problem of inducing regeneration in other amniotes, including mammals. The study uniquely reveals that a specific population of fibroblasts, only present in the tail, are at the origin of the axial cartilage that in the regenerating tail replaces the vertebral column, and confirms previous studies from the same lab that indicated shh as the key molecule that induces cartilage formation. The study also reveals the importance of the type of immune cells activated in the healing tissues for the success of regeneration. Although the MS is OK in its present form, the authors MAY make a small further effort to reduce the length of it (presently 51 pages!), for sake of the readership...but this is not mandatory for the acceptance of the MS. Also an attempt to reduce the extensive use of acronyms to the minimum, would be good for easy the reading of the MS. At least reduce the terms that are unfrequently utilized (example DEGs, GEM etc.). HOWEVER, If the authors really retain that the MS cannot be shortened without losing infos, the MS is fine with me.

SPECIFIC REMARKS

Introduction

Can be shortened

Mat Meth

-I only advise the authors to exploit tail autotomy in next experiments (can be done relatively on a precise level of the tail), to avoid stimulation of inflammation using a blade. This is also a natural way for a lizard to regenerate the tail.

Results

Can be shortened

-pag. 12:14 DPA. AT 14 days post-transplant.....: HERE AND IN OTHER SENTENCES START ANY SENTENCE WRITING NUMBERS AS A WORD...so either "AT 14 days post-transplant...OR "Fourteen days post-transplant....

-The drawings really help the reader to grasp and digest the numerous and long sentences of this part of the MS.

Discussion

-this part of the MS is fine.

-pag 23:.....interesting question: Why the amputated lizard tail....: "Why" should be in small letter "...: why the amputated....

Reviewer #2 (Remarks to the Author):

The authors addressed all of our concerns to the best of their ability and to sufficient depth for publication.

Reviewer #3 (Remarks to the Author):

The authors deserve appreciation for conducting numerous additional experiments to address reviewer concerns and carefully revising their text to moderate their initial conclusions. Although I maintain the viewpoint that the main text remains challenging to follow, and the proposed model does not demonstrate a direct step-by-step activation as depicted in Figure 10 (dashed lines would be more appropriate), considering the improvements made, the manuscript can be considered for publication in its current version.